# Combinatorial Pure Exploration of Multi-Armed Bandits

**Shouyuan Chen**[1*]    **Tian Lin**[2]    **Irwin King**[1]    **Michael R. Lyu**[1]    **Wei Chen**[3]
[1]The Chinese University of Hong Kong    [2]Tsinghua University    [3]Microsoft Research Asia
[1]{sychen,king,lyu}@cse.cuhk.edu.hk    [2]lint10@mails.tsinghua.edu.cn    [3]weic@microsoft.com

## Abstract

We study the *combinatorial pure exploration (CPE)* problem in the stochastic multi-armed bandit setting, where a learner explores a set of arms with the objective of identifying the optimal member of a *decision class*, which is a collection of subsets of arms with certain combinatorial structures such as size-$K$ subsets, matchings, spanning trees or paths, etc. The CPE problem represents a rich class of pure exploration tasks which covers not only many existing models but also novel cases where the object of interest has a non-trivial combinatorial structure. In this paper, we provide a series of results for the general CPE problem. We present general learning algorithms which work for all decision classes that admit offline maximization oracles in both fixed confidence and fixed budget settings. We prove problem-dependent upper bounds of our algorithms. Our analysis exploits the combinatorial structures of the decision classes and introduces a new analytic tool. We also establish a general problem-dependent lower bound for the CPE problem. Our results show that the proposed algorithms achieve the optimal sample complexity (within logarithmic factors) for many decision classes. In addition, applying our results back to the problems of top-$K$ arms identification and multiple bandit best arms identification, we recover the best available upper bounds up to constant factors and partially resolve a conjecture on the lower bounds.

## 1    Introduction

Multi-armed bandit (MAB) is a predominant model for characterizing the tradeoff between exploration and exploitation in decision-making problems. Although this is an intrinsic tradeoff in many tasks, some application domains prefer a dedicated exploration procedure in which the goal is to identify an optimal object among a collection of candidates and the reward or loss incurred during exploration is irrelevant. In light of these applications, the related learning problem, called pure exploration in MABs, has received much attention. Recent advances in pure exploration MABs have found potential applications in many domains including crowdsourcing, communication network and online advertising.

In many of these application domains, a recurring problem is to identify the optimal object with certain *combinatorial structure*. For example, a crowdsourcing application may want to find the best assignment from workers to tasks such that overall productivity of workers is maximized. A network routing system during the initialization phase may try to build a spanning tree that minimizes the delay of links, or attempts to identify the shortest path between two sites. An online advertising system may be interested in finding the best matching between ads and display slots. The literature of pure exploration MAB problems lacks a framework that encompasses these kinds of problems where the object of interest has a non-trivial combinatorial structure. Our paper contributes such a framework which accounts for general combinatorial structures, and develops a series of results, including algorithms, upper bounds and lower bounds for the framework.

In this paper, we formulate the *combinatorial pure exploration (CPE)* problem for stochastic multi-armed bandits. In the CPE problem, a learner has a fixed set of arms and each arm is associated with an unknown reward distribution. The learner is also given a collection of sets of arms called *decision class*, which corresponds to a collection of certain combinatorial structures. During the exploration period, in each round the learner chooses an arm to play and observes a random reward sampled from

the associated distribution. The objective is when the exploration period ends, the learner outputs a member of the decision class that she believes to be optimal, in the sense that the sum of expected rewards of all arms in the output set is maximized among all members in the decision class.

The CPE framework represents a rich class of pure exploration problems. The conventional pure exploration problem in MAB, whose objective is to find the single best arm, clearly fits into this framework, in which the decision class is the collection of all singletons. This framework also naturally encompasses several recent extensions, including the problem of finding the top $K$ arms (henceforth TOPK) [18, 19, 8, 20, 31] and the multi-bandit problem of finding the best arms simultaneously from several disjoint sets of arms (henceforth MB) [12, 8]. Further, this framework covers many more interesting cases where the decision classes correspond to collections of non-trivial combinatorial structures. For example, suppose that the arms represent the edges in a graph. Then a decision class could be the set of all paths between two vertices, all spanning trees or all matchings of the graph. And, in these cases, the objectives of CPE become identifying the optimal paths, spanning trees and matchings through bandit explorations, respectively. To our knowledge, there are no results available in the literature for these pure exploration tasks.

The CPE framework raises several interesting challenges to the design and analysis of pure exploration algorithms. One challenge is that, instead of solving each type of CPE task in an ad-hoc way, one requires a unified algorithm and analysis that support different decision classes. Another challenge stems from the combinatorial nature of CPE, namely that the optimal set may contain some arms with very small expected rewards (e.g., it is possible that a maximum matching contains the edge with the smallest weight); hence, arms cannot be eliminated simply based on their own rewards in the learning algorithm or ignored in the analysis. This differs from many existing approach of pure exploration MABs. Therefore, the design and analysis of algorithms for CPE demands novel techniques which take both rewards and combinatorial structures into account.

**Our results.** In this paper, we propose two novel learning algorithms for general CPE problem: one for the fixed confidence setting and one for the fixed budget setting. Both algorithms support a wide range of decision classes in a unified way. In the fixed confidence setting, we present Combinatorial Lower-Upper Confidence Bound (CLUCB) algorithm. The CLUCB algorithm does not need to know the definition of the decision class, as long as it has access to the decision class through a maximization oracle. We upper bound the number of samples used by CLUCB. This sample complexity bound depends on both the expected rewards and the structure of decision class. Our analysis relies on a novel combinatorial construction called *exchange class*, which may be of independent interest for other combinatorial optimization problems. Specializing our result to TOPK and MB, we recover the best available sample complexity bounds [19, 13, 20] up to constant factors. While for other decision classes in general, our result establishes the first sample complexity upper bound. We further show that CLUCB can be easily extended to the fixed budget setting and PAC learning setting and we provide related theoretical guarantees in the supplementary material.

Moreover, we establish a problem-dependent sample complexity lower bound for the CPE problem. Our lower bound shows that the sample complexity of the proposed CLUCB algorithm is optimal (to within logarithmic factors) for many decision classes, including TOPK, MB and the decision classes derived from matroids (e.g., spanning tree). Therefore our upper and lower bounds provide a nearly full characterization of the sample complexity of these CPE problems. For more general decision classes, our results show that the upper and lower bounds are within a relatively benign factor. To the best of our knowledge, there are no problem-dependent lower bounds known for pure exploration MABs besides the case of identifying the single best arm [24, 1]. We also notice that our result resolves the conjecture of Bubeck et al. [8] on the problem-dependent sample complexity lower bounds of TOPK and MB problems, for the cases of Gaussian reward distributions.

In the fixed budget setting, we present a parameter-free algorithm called Combinatorial Successive Accept Reject (CSAR) algorithm. We prove a probability of error bound of the CSAR algorithm. This bound can be shown to be equivalent to the sample complexity bound of CLUCB within logarithmic factors, although the two algorithms are based on quite different techniques. Our analysis of CSAR re-uses exchange classes as tools. This suggests that exchange classes may be useful for analyzing similar problems. In addition, when applying the algorithm to back TOPK and MB, our bound recovers the best known result in the fixed budget setting due to Bubeck et al. [8] up to constant factors.

## 2   Problem Formulation

In this section, we formally define the CPE problem. Suppose that there are $n$ arms and the arms are numbered $1, 2, \ldots, n$. Assume that each arm $e \in [n]$ is associated with a reward distribution $\varphi_e$. Let $\boldsymbol{w} = \big(w(1), \ldots, w(n)\big)^T$ denote the vector of expected rewards, where each entry $w(e) = \mathbb{E}_{X \sim \varphi_e}[X]$ denotes the expected reward of arm $e$. Following standard assumptions of stochastic MABs, we assume that all reward distributions have $R$-sub-Gaussian tails for some known constant $R > 0$. Formally, if $X$ is a random variable drawn from $\varphi_e$ for some $e \in [n]$, then, for all $t \in \mathbb{R}$, one has $\mathbb{E}\big[\exp(tX - t\mathbb{E}[X])\big] \leq \exp(R^2 t^2 / 2)$. It is known that the family of $R$-sub-Gaussian tail distributions encompasses all distributions that are supported on $[0, R]$ as well as many unbounded distributions such as Gaussian distributions with variance $R^2$ (see e.g., [27, 28]).

We define a *decision class* $\mathcal{M} \subseteq 2^{[n]}$ as a collection of sets of arms. Let $M_* = \arg\max_{M \in \mathcal{M}} w(M)$ denote the optimal member of the decision class $\mathcal{M}$ which maximizes the sum of expected rewards[1]. A learner's objective is to identify $M_*$ from $\mathcal{M}$ by playing the following game with the stochastic environment. At the beginning of the game, the decision class $\mathcal{M}$ is revealed to the learner while the reward distributions $\{\varphi_e\}_{e \in [n]}$ are unknown to her. Then, the learner plays the game over a sequence of rounds; in each round $t$, she pulls an arm $p_t \in [n]$ and observes a reward sampled from the associated reward distribution $\varphi_{p_t}$. The game continues until certain stopping condition is satisfied. After the game finishes, the learner need to output a set $\mathsf{Out} \in \mathcal{M}$.

We consider two different stopping conditions of the game, which are known as *fixed confidence* setting and *fixed budget* setting in the literature. In the fixed confidence setting, the learner can stop the game at any round. She need to guarantee that $\Pr[\mathsf{Out} = M_*] \geq 1 - \delta$ for a given confidence parameter $\delta$. The learner's performance is evaluated by her *sample complexity*, i.e., the number of pulls used by the learner. In the fixed budget setting, the game stops after a fixed number $T$ of rounds, where $T$ is given before the game starts. The learner tries to minimize the *probability of error*, which is formally $\Pr[\mathsf{Out} \neq M_*]$, within $T$ rounds. In this setting, her performance is measured by the probability of error.

## 3   Algorithm, Exchange Class and Sample Complexity

In this section, we present Combinatorial Lower-Upper Confidence Bound (CLUCB) algorithm, a learning algorithm for the CPE problem in the fixed confidence setting, and analyze its sample complexity. En route to our sample complexity bound, we introduce the notions of exchange classes and the widths of decision classes, which play an important role in the analysis and sample complexity bound. Furthermore, the CLUCB algorithm can be extended to the fixed budget and PAC learning settings, the discussion of which is included in the supplementary material (Appendix B).

**Oracle.** We allow the CLUCB algorithm to access a *maximization oracle*. A maximization oracle takes a weight vector $\boldsymbol{v} \in \mathbb{R}^n$ as input and finds an optimal set from a given decision class $\mathcal{M}$ with respect to the weight vector $\boldsymbol{v}$. Formally, we call a function Oracle: $\mathbb{R}^n \to \mathcal{M}$ a maximization oracle for $\mathcal{M}$ if, for all $\boldsymbol{v} \in \mathbb{R}^n$, we have $\mathrm{Oracle}(\boldsymbol{v}) \in \arg\max_{M \in \mathcal{M}} v(M)$. It is clear that a wide range of decision classes admit such maximization oracles, including decision classes corresponding to collections of matchings, paths or bases of matroids (see later for concrete examples). Besides the access to the oracle, CLUCB does not need *any* additional knowledge of the decision class $\mathcal{M}$.

**Algorithm.** Now we describe the details of CLUCB, as shown in Algorithm 1. During its execution, the CLUCB algorithm maintains empirical mean $\bar{w}_t(e)$ and confidence radius $\mathrm{rad}_t(e)$ for each arm $e \in [n]$ and each round $t$. The construction of confidence radius ensures that $|w(e) - \bar{w}_t(e)| \leq \mathrm{rad}_t(e)$ holds with high probability for each arm $e \in [n]$ and each round $t > 0$. CLUCB begins with an initialization phase in which each arm is pulled once. Then, at round $t \geq n$, CLUCB uses the following procedure to choose an arm to play. First, CLUCB calls the oracle which finds the set $M_t = \mathrm{Oracle}(\bar{\boldsymbol{w}}_t)$. The set $M_t$ is the "best" set with respect to the empirical means $\bar{\boldsymbol{w}}_t$. Then, CLUCB explores possible refinements of $M_t$. In particular, CLUCB uses the confidence radius to compute an adjusted expectation vector $\tilde{\boldsymbol{w}}_t$ in the following way: for each arm $e \in M_t$, $\tilde{w}_t(e)$ is equal to to the lower confidence bound $\tilde{w}_t(e) = \bar{w}_t(e) - \mathrm{rad}_t(e)$; and for each arm $e \notin M_t$, $\tilde{w}_t(e)$ is equal to the upper confidence bound $\tilde{w}_t(e) = \bar{w}_t(e) + \mathrm{rad}_t(e)$. Intuitively, the adjusted expectation vector $\tilde{\boldsymbol{w}}_t$ penalizes arms belonging to the current set $M_t$ and encourages exploring arms out of

**Algorithm 1** CLUCB: Combinatorial Lower-Upper Confidence Bound

---
**Require:** Confidence $\delta \in (0,1)$; Maximization oracle: Oracle$(\cdot) : \mathbb{R}^n \to \mathcal{M}$
    **Initialize:** Play each arm $e \in [n]$ once. Initialize empirical means $\bar{\boldsymbol{w}}_n$ and set $T_n(e) \leftarrow 1$ for all $e$.
 1: **for** $t = n, n+1, \dots$ **do**
 2:     $M_t \leftarrow$ Oracle$(\bar{\boldsymbol{w}}_t)$
 3:     Compute confidence radius $\mathrm{rad}_t(e)$ for all $e \in [n]$            $\triangleright$ $\mathrm{rad}_t(e)$ is defined later in Theorem 1
 4:     **for** $e = 1, \dots, n$ **do**
 5:         **if** $e \in M_t$ **then** $\tilde{w}_t(e) \leftarrow \bar{w}_t(e) - \mathrm{rad}_t(e)$
 6:         **else** $\tilde{w}_t(e) \leftarrow \bar{w}_t(e) + \mathrm{rad}_t(e)$
 7:     $\tilde{M}_t \leftarrow$ Oracle$(\tilde{\boldsymbol{w}}_t)$
 8:     **if** $\tilde{w}_t(\tilde{M}_t) = \tilde{w}_t(M_t)$ **then**
 9:         Out $\leftarrow M_t$
10:         **return** Out
11:     $p_t \leftarrow \arg\max_{e \in (\tilde{M}_t \setminus M_t) \cup (M_t \setminus \tilde{M}_t)} \mathrm{rad}_t(e)$            $\triangleright$ break ties arbitrarily
12:     Pull arm $p_t$ and observe the reward
13:     Update empirical means $\bar{\boldsymbol{w}}_{t+1}$ using the observed reward
14:     Update number of pulls: $T_{t+1}(p_t) \leftarrow T_t(p_t) + 1$ and $T_{t+1}(e) \leftarrow T_t(e)$ for all $e \neq p_t$

---

$M_t$. CLUCB then calls the oracle using the adjusted expectation vector $\tilde{\boldsymbol{w}}_t$ as input to compute a refined set $\tilde{M}_t = \mathrm{Oracle}(\tilde{\boldsymbol{w}}_t)$. If $\tilde{w}_t(\tilde{M}_t) = \tilde{w}_t(M_t)$ then CLUCB stops and returns Out $= M_t$. Otherwise, CLUCB pulls the arm that belongs to the symmetric difference between $M_t$ and $\tilde{M}_t$ and has the largest confidence radius (intuitively the largest uncertainty). This ends the $t$-th round of CLUCB. We note that CLUCB generalizes and unifies the ideas of several different fixed confidence algorithms dedicated to the TOPK and MB problems in the literature [19, 13, 20].

### 3.1 Sample complexity

Now we establish a problem-dependent sample complexity bound of the CLUCB algorithm. To formally state our result, we need to introduce several notions.

**Gap.** We begin with defining a natural hardness measure of the CPE problem. For each arm $e \in [n]$, we define its gap $\Delta_e$ as

$$\Delta_e = \begin{cases} w(M_*) - \max_{M \in \mathcal{M}: e \in M} w(M) & \text{if } e \notin M_*, \\ w(M_*) - \max_{M \in \mathcal{M}: e \notin M} w(M) & \text{if } e \in M_*, \end{cases} \tag{1}$$

where we adopt the convention that the maximum value of an empty set is $-\infty$. We also define the hardness $\mathbf{H}$ as the sum of inverse squared gaps

$$\mathbf{H} = \sum_{e \in [n]} \Delta_e^{-2}. \tag{2}$$

We see that, for each arm $e \notin M_*$, the gap $\Delta_e$ represents the sub-optimality of the best set that includes arm $e$; and, for each arm $e \in M_*$, the gap $\Delta_e$ is the sub-optimality of the best set that does not include arm $e$. This naturally generalizes and unifies previous definitions of gaps [1, 12, 18, 8].

**Exchange class and the width of a decision class.** A notable challenge of our analysis stems from the generality of CLUCB which, as we have seen, supports a wide range of decision classes $\mathcal{M}$. Indeed, previous algorithms for special cases including TOPK and MB require a separate analysis for each individual type of problem. Such strategy is intractable for our setting and we need a unified analysis for all decision classes. Our solution to this challenge is a novel combinatorial construction called *exchange class*, which is used as a proxy for the structure of the decision class. Intuitively, an exchange class $\mathcal{B}$ for a decision class $\mathcal{M}$ can be seen as a collection of "patches" (borrowing concepts from source code management) such that, for any two different sets $M, M' \in \mathcal{M}$, one can transform $M$ to $M'$ by applying a series of patches of $\mathcal{B}$; and each application of a patch yields a valid member of $\mathcal{M}$. These patches are later used by our analysis to build gadgets that interpolate between different members of the decision class and serve to bridge key quantities. Furthermore, the maximum patch size of $\mathcal{B}$ will play an important role in our sample complexity bound.

Now we formally define the exchange class. We begin with the definition of exchange sets, which formalize the aforementioned "patches". We define an exchange set $b$ as an ordered pair of disjoint sets $b = (b_+, b_-)$ where $b_+ \cap b_- = \emptyset$ and $b_+, b_- \subseteq [n]$. Then, we define operator $\oplus$ such that, for any set $M \subseteq [n]$ and any exchange set $b = (b_+, b_-)$, we have $M \oplus b \triangleq M \backslash b_- \cup b_+$. Similarly, we also define operator $\ominus$ such that $M \ominus b \triangleq M \backslash b_+ \cup b_-$.

We call a collection of exchange sets $\mathcal{B}$ an *exchange class for* $\mathcal{M}$ if $\mathcal{B}$ satisfies the following property. For any $M, M' \in \mathcal{M}$ such that $M \neq M'$ and for any $e \in (M \backslash M')$, there exists an exchange set $(b_+, b_-) \in \mathcal{B}$ which satisfies five constraints: **(a)** $e \in b_-$, **(b)** $b_+ \subseteq M' \backslash M$, **(c)** $b_- \subseteq M \backslash M'$, **(d)** $(M \oplus b) \in \mathcal{M}$ and **(e)** $(M' \ominus b) \in \mathcal{M}$.

Intuitively, constraints **(b)** and **(c)** resemble the concept of patches in the sense that $b_+$ contains only the "new" elements from $M'$ and $b_-$ contains only the "old" elements of $M$; constraints **(d)** and **(e)** allow one to transform $M$ one step closer to $M'$ by applying a patch $b \in \mathcal{B}$ to yield $(M \oplus b) \in \mathcal{M}$ (and similarly for $M' \ominus b$). These transformations are the basic building blocks in our analysis. Furthermore, as we will see later in our examples, for many decision classes, there are exchange classes representing natural combinatorial structures, e.g., augmenting paths and cycles of matchings.

In our analysis, the key quantity of exchange class is called *width*, which is defined as the size of the largest exchange set as follows

$$\text{width}(\mathcal{B}) = \max_{(b_+, b_-) \in \mathcal{B}} |b_+| + |b_-|. \tag{3}$$

Let $\text{Exchange}(\mathcal{M})$ denote the family of all possible exchange classes for $\mathcal{M}$. We define the width of a decision class $\mathcal{M}$ as the width of the thinnest exchange class

$$\text{width}(\mathcal{M}) = \min_{\mathcal{B} \in \text{Exchange}(\mathcal{M})} \text{width}(\mathcal{B}). \tag{4}$$

**Sample complexity.** Our main result of this section is a problem-dependent sample complexity bound of the CLUCB algorithm which show that, with high probability, CLUCB returns the optimal set $M_*$ and uses at most $\tilde{O}\big(\text{width}(\mathcal{M})^2 \mathbf{H}\big)$ samples.

**Theorem 1.** *Given any* $\delta \in (0,1)$*, any decision class* $\mathcal{M} \subseteq 2^{[n]}$ *and any expected rewards* $\boldsymbol{w} \in \mathbb{R}^n$*. Assume that the reward distribution* $\varphi_e$ *for each arm* $e \in [n]$ *has mean* $w(e)$ *with an* $R$*-sub-Gaussian tail. Let* $M_* = \arg\max_{M \in \mathcal{M}} w(M)$ *denote the optimal set. Set* $\text{rad}_t(e) = R\sqrt{2\log\left(\frac{4nt^3}{\delta}\right)/T_t(e)}$ *for all* $t > 0$ *and* $e \in [n]$*. Then, with probability at least* $1 - \delta$*, the CLUCB algorithm (Algorithm 1) returns the optimal set* $\text{Out} = M_*$ *and*

$$T \leq O\left(R^2 \text{width}(\mathcal{M})^2 \mathbf{H} \log\left(nR^2\mathbf{H}/\delta\right)\right), \tag{5}$$

*where* $T$ *denotes the number of samples used by Algorithm 1,* $\mathbf{H}$ *is defined in Eq. (2) and* $\text{width}(\mathcal{M})$ *is defined in Eq. (4).*

### 3.2 Examples of decision classes

Now we investigate several concrete types of decision classes, which correspond to different CPE tasks. We analyze the width of these decision classes and apply Theorem 1 to obtain the sample complexity bounds. A detailed analysis and the constructions of exchange classes can be found in the supplementary material (Appendix F). We begin with the problems of top-$K$ arm identification (TOPK) and multi-bandit best arms identification (MB).

**Example 1** (TOPK and MB). *For any* $K \in [n]$*, the problem of finding the top* $K$ *arms with the largest expected reward can be modeled by decision class* $\mathcal{M}_{\text{TOPK}(K)} = \{M \subseteq [n] \mid |M| = K\}$*. Let* $\mathcal{A} = \{A_1, \ldots, A_m\}$ *be a partition of* $[n]$*. The problem of identifying the best arms from each group of arms* $A_1, \ldots, A_m$ *can be modeled by decision class* $\mathcal{M}_{\text{MB}(\mathcal{A})} = \{M \subseteq [n] \mid \forall i \in [m], |M \cap A_i| = 1\}$*. Note that maximization oracles for these two decision classes are trivially the functions of returning the top* $k$ *arms or the best arms of each group.*

*Then we have* $\text{width}(\mathcal{M}_{\text{TOPK}(K)}) \leq 2$ *and* $\text{width}(\mathcal{M}_{\text{MB}(\mathcal{A})}) \leq 2$ *(see Fact 2 and 3 in the supplementary material) and therefore the sample complexity of CLUCB for solving TOPK and MB is* $O\big(\mathbf{H}\log(n\mathbf{H}/\delta)\big)$*, which matches previous results in the fixed confidence setting [19, 13, 20] up to constant factors.*

Next we consider the problem of identifying the maximum matching and the problem of finding the shortest path (by negating the rewards), in a setting where arms correspond to edges. For these problems, Theorem 1 establishes the first known sample complexity bound.

**Example 2** (Matchings and Paths). *Let $G(V, E)$ be a graph with $n$ edges and assume there is a one-to-one mapping between edges $E$ and arms $[n]$. Suppose that $G$ is a bipartite graph. Let $\mathcal{M}_{\mathrm{MATCH}(G)}$ correspond to the set of all matchings in $G$. Then we have $\mathrm{width}(\mathcal{M}_{\mathrm{MATCH}(G)}) \leq |V|$ (In fact, we construct an exchange class corresponding to the collection of augmenting cycles and augmenting paths of $G$; see Fact 4).*

*Next suppose that $G$ is a directed acyclic graph and let $s, t \in V$ be two vertices. Let $\mathcal{M}_{\mathrm{PATH}(G,s,t)}$ correspond to the set of all paths from $s$ to $t$. Then we have $\mathrm{width}(\mathcal{M}_{\mathrm{PATH}(G,s,t)}) \leq |V|$ (In fact, we construct an exchange class corresponding to the collection of disjoint pairs of paths; see Fact 5). Therefore the sample complexity bounds of CLUCB for decision classes $\mathcal{M}_{\mathrm{MATCH}(G)}$ and $\mathcal{M}_{\mathrm{PATH}(G,s,t)}$ are $O\big(|V|^2 \mathbf{H} \log(n\mathbf{H}/\delta)\big)$.*

Last, we investigate the general problem of identifying the maximum-weight basis of a matroid. Again, Theorem 1 is the first sample complexity upper bound for this type of pure exploration tasks.

**Example 3** (Matroids). *Let $T = (E, \mathcal{I})$ be a finite matroid, where $E$ is a set of size $n$ (called ground set) and $\mathcal{I}$ is a family of subsets of $E$ (called independent sets) which satisfies the axioms of matroids (see Footnote 3 in Appendix F). Assume that there is a one-to-one mapping between $E$ and $[n]$. Recall that a basis of matroid $T$ is a maximal independent set. Let $\mathcal{M}_{\mathrm{MATROID}(T)}$ correspond to the set of all bases of $T$. Then we have $\mathrm{width}(\mathcal{M}_{\mathrm{MATROID}(T)}) \leq 2$ (derived from strong basis exchange property of matroids; see Fact 1) and the sample complexity of CLUCB for $\mathcal{M}_{\mathrm{MATROID}(T)}$ is $O\big(\mathbf{H} \log(n\mathbf{H}/\delta)\big)$.*

The last example $\mathcal{M}_{\mathrm{MATROID}(T)}$ is a general type of decision class which encompasses many pure exploration tasks including TOPK and MB as special cases, where TOPK corresponds to uniform matroids of rank $K$ and MB corresponds to partition matroids. It is easy to see that $\mathcal{M}_{\mathrm{MATROID}(T)}$ also covers the decision class that contains all spanning trees of a graph. On the other hand, it has been established that matchings and paths cannot be formulated as matroids since they are matroid intersections [26].

## 4   Lower Bound

In this section, we present a problem-dependent lower bound on the sample complexity of the CPE problem. To state our results, we first define the notion of $\delta$-*correct algorithm* as follows. For any $\delta \in (0, 1)$, we call an algorithm $\mathbb{A}$ a $\delta$-correct algorithm if, for any expected reward $\boldsymbol{w} \in \mathbb{R}^n$, the probability of error of $\mathbb{A}$ is at most $\delta$, i.e., $\Pr[M_* \neq \mathsf{Out}] \leq \delta$, where $\mathsf{Out}$ is the output of $\mathbb{A}$.

We show that, for any decision class $\mathcal{M}$ and any expected rewards $\boldsymbol{w}$, a $\delta$-correct algorithm $\mathbb{A}$ must use at least $\Omega\big(\mathbf{H} \log(1/\delta)\big)$ samples in expectation.

**Theorem 2.** *Fix any decision class $\mathcal{M} \subseteq 2^{[n]}$ and any vector $\boldsymbol{w} \in \mathbb{R}^n$. Suppose that, for each arm $e \in [n]$, the reward distribution $\varphi_e$ is given by $\varphi_e = \mathcal{N}(w(e), 1)$, where we let $\mathcal{N}(\mu, \sigma^2)$ denote Gaussian distribution with mean $\mu$ and variance $\sigma^2$. Then, for any $\delta \in (0, e^{-16}/4)$ and any $\delta$-correct algorithm $\mathbb{A}$, we have*

$$\mathbb{E}[T] \geq \frac{1}{16} \mathbf{H} \log \left( \frac{1}{4\delta} \right), \tag{6}$$

*where $T$ denote the number of total samples used by algorithm $\mathbb{A}$ and $\mathbf{H}$ is defined in Eq. (2).*

In Example 1 and Example 3, we have seen that the sample complexity of CLUCB is $O(\mathbf{H} \log(n\mathbf{H}/\delta))$ for pure exploration tasks including TOPK, MB and more generally the CPE tasks with decision classes derived from matroids, i.e., $\mathcal{M}_{\mathrm{MATROID}(T)}$ (including spanning trees). Hence, our upper and lower bound show that the CLUCB algorithm achieves the optimal sample complexity within logarithmic factors for these pure exploration tasks. In addition, we remark that Theorem 2 resolves the conjecture of Bubeck et al. [8] that the lower bounds of sample complexity of TOPK and MB problems are $\Omega\big(\mathbf{H} \log(1/\delta)\big)$, for the cases of Gaussian reward distributions.

On the other hand, for general decision classes with non-constant widths, we see that there is a gap of $\tilde{\Theta}(\mathrm{width}(\mathcal{M})^2)$ between the upper bound Eq. (5) and the lower bound Eq. (6). Notice that we have $\mathrm{width}(\mathcal{M}) \leq n$ for any decision class $\mathcal{M}$ and therefore the gap is relatively benign. Our lower bound also suggests that the dependency on $\mathbf{H}$ of the sample complexity of CLUCB cannot be improved up to logarithmic factors. Furthermore, we conjecture that the sample complexity lower bound might inherently depend on the size of exchange sets. In the supplementary material (Appendix C.2), we

provide evidences on this conjecture which is a lower bound on the sample complexity of exploration of the exchange sets.

## 5 Fixed Budget Algorithm

In this section, we present Combinatorial Successive Accept Reject (CSAR) algorithm, which is a parameter-free learning algorithm for the CPE problem in the fixed budget setting. Then, we upper bound the probability of error CSAR in terms of gaps and $\mathrm{width}(\mathcal{M})$.

**Constrained oracle.** The CSAR algorithm requires access to a *constrained oracle*, which is a function denoted as $\mathrm{COracle} : \mathbb{R}^n \times 2^{[n]} \times 2^{[n]} \to \mathcal{M} \cup \{\perp\}$ and satisfies

$$\mathrm{COracle}(\boldsymbol{v}, A, B) = \begin{cases} \arg\max_{M \in \mathcal{M}_{A,B}} v(M) & \text{if } \mathcal{M}_{A,B} \neq \emptyset \\ \perp & \text{if } \mathcal{M}_{A,B} = \emptyset, \end{cases} \tag{7}$$

where we define $\mathcal{M}_{A,B} = \{M \in \mathcal{M} \mid A \subseteq M, B \cap M = \emptyset\}$ as the collection of feasible sets and $\perp$ is a null symbol. Hence we see that $\mathrm{COracle}(\boldsymbol{v}, A, B)$ returns an optimal set that includes all elements of $A$ while excluding all elements of $B$; and if there are no feasible sets, the constrained oracle $\mathrm{COracle}(\boldsymbol{v}, A, B)$ returns the null symbol $\perp$. In the supplementary material (Appendix G), we show that constrained oracles are equivalent to maximization oracles up to a transformation on the weight vector. In addition, similar to CLUCB, CSAR does not need any additional knowledge of $\mathcal{M}$ other than accesses to a constrained oracle for $\mathcal{M}$.

**Algorithm.** The idea of the CSAR algorithm is as follows. The CSAR algorithm divides the budget of $T$ rounds into $n$ phases. In the end of each phase, CSAR either accepts or rejects a single arm. If an arm is accepted, then it is included into the final output. Conversely, if an arm is rejected, then it is excluded from the final output. The arms that are neither accepted nor rejected are sampled for an equal number of times in the next phase.

Now we describe the procedure of the CSAR algorithm for choosing an arm to accept/reject. Let $A_t$ denote the set of accepted arms before phase $t$ and let $B_t$ denote the set of rejected arms before phase $t$. We call an arm $e$ to be active if $e \notin A_t \cup B_t$. In the beginning of phase $t$, CSAR samples each active arm for $\tilde{T}_t - \tilde{T}_{t-1}$ times, where the definition of $\tilde{T}_t$ is given in Algorithm 2. Next, CSAR calls the constrained oracle to compute an optimal set $M_t$ with respect to the empirical means $\bar{\boldsymbol{w}}_t$, accepted arms $A_t$ and rejected arms $B_t$, i.e., $M_t = \mathrm{COracle}(\bar{\boldsymbol{w}}_t, A_t, B_t)$. It is clear that the output of $\mathrm{COracle}(\bar{\boldsymbol{w}}_t, A_t, B_t)$ is independent from the input $\bar{w}_t(e)$ for any $e \in A_t \cup B_t$. Then, for each active arm $e$, CSAR estimates the "empirical gap" of $e$ in the following way. If $e \in M_t$, then CSAR computes an optimal set $\tilde{M}_{t,e}$ that does not include $e$, i.e., $\tilde{M}_{t,e} = \mathrm{COracle}(\bar{\boldsymbol{w}}_t, A_t, B_t \cup \{e\})$. Conversely, if $e \notin M_t$, then CSAR computes an optimal $\tilde{M}_{t,e}$ which includes $e$, i.e., $\tilde{M}_{t,e} = \mathrm{COracle}(\bar{\boldsymbol{w}}_t, A_t \cup \{e\}, B_t)$. Then, the empirical gap of $e$ is calculated as $\bar{w}_t(M_t) - \bar{w}_t(\tilde{M}_{t,e})$. Finally, CSAR chooses the arm $p_t$ which has the largest empirical gap. If $p_t \in M_t$ then $p_t$ is accepted, otherwise $p_t$ is rejected. The pseudo-code CSAR is shown in Algorithm 2. We note that CSAR can be considered as a generalization of the ideas of the two versions of SAR algorithm due to Bubeck et al. [8], which are designed specifically for the TOPK and MB problems respectively.

### 5.1 Probability of error

In the following theorem, we bound the probability of error of the CSAR algorithm.

**Theorem 3.** *Given any $T > n$, any decision class $\mathcal{M} \subseteq 2^{[n]}$ and any expected rewards $\boldsymbol{w} \in \mathbb{R}^n$. Assume that the reward distribution $\varphi_e$ for each arm $e \in [n]$ has mean $w(e)$ with an $R$-sub-Gaussian tail. Let $\Delta_{(1)}, \ldots, \Delta_{(n)}$ be a permutation of $\Delta_1, \ldots, \Delta_n$ (defined in Eq. (1)) such that $\Delta_{(1)} \leq \ldots \ldots \Delta_{(n)}$. Define $\mathbf{H}_2 \triangleq \max_{i \in [n]} i \Delta_{(i)}^{-2}$. Then, the CSAR algorithm uses at most $T$ samples and outputs a solution $\mathsf{Out} \in \mathcal{M} \cup \{\perp\}$ such that*

$$\Pr[\mathsf{Out} \neq M_*] \leq n^2 \exp\left(-\frac{(T-n)}{18R^2 \tilde{\log}(n)\,\mathrm{width}(\mathcal{M})^2 \mathbf{H}_2}\right), \tag{8}$$

*where $\tilde{\log}(n) \triangleq \sum_{i=1}^n i^{-1}$, $M_* = \arg\max_{M \in \mathcal{M}} w(M)$ and $\mathrm{width}(\mathcal{M})$ is defined in Eq. (4).*

One can verify that $\mathbf{H}_2$ is equivalent to $\mathbf{H}$ up to a logarithmic factor: $\mathbf{H}_2 \leq \mathbf{H} \leq \log(2n)\mathbf{H}_2$ (see [1]). Therefore, by setting the probability of error (the RHS of Eq. (8)) to a constant, one can see that CSAR requires a budget of $T = \tilde{O}(\mathrm{width}(\mathcal{M})^2\mathbf{H})$ samples. This is equivalent to the sample complexity bound of CLUCB up to logarithmic factors. In addition, applying Theorem 3 back to TOPK and MB, our bound matches the previous fixed budget algorithm due to Bubeck et al. [8].

---
**Algorithm 2** CSAR: Combinatorial Successive Accept Reject
---
**Require:** Budget: $T > 0$; Constrained oracle: $\text{COracle} : \mathbb{R}^n \times 2^{[n]} \times 2^{[n]} \to \mathcal{M} \cup \{\bot\}$.
1: Define $\tilde{\log}(n) \triangleq \sum_{i=1}^{n} \frac{1}{i}$
2: $\tilde{T}_0 \leftarrow 0, A_1 \leftarrow \emptyset, B_1 \leftarrow \emptyset$
3: **for** $t = 1, \ldots, n$ **do**
4:　　$\tilde{T}_t \leftarrow \left\lceil \frac{T-n}{\tilde{\log}(n)(n-t+1)} \right\rceil$
5:　　Pull each arm $e \in [n] \backslash (A_t \cup B_t)$ for $\tilde{T}_t - \tilde{T}_{t-1}$ times
6:　　Update the empirical means $\bar{\boldsymbol{w}}_t$ for each arm $e \in [n] \backslash (A_t \cup B_t)$ 　　　　▷ set $\bar{w}_t(e) = 0, \ \forall e \in A_t \cup B_t$
7:　　$M_t \leftarrow \text{COracle}(\bar{\boldsymbol{w}}_t, A_t, B_t)$
8:　　**if** $M_t = \bot$ **then**
9:　　　　**fail:** set $\mathsf{Out} \leftarrow \bot$ and **return** $\mathsf{Out}$
10:　　**for each** $e \in [n] \backslash (A_t \cup B_t)$ **do**
11:　　　　**if** $e \in M_t$ **then** $\tilde{M}_{t,e} \leftarrow \text{COracle}(\bar{\boldsymbol{w}}_t, A_t, B_t \cup \{e\})$
12:　　　　**else** $\tilde{M}_{t,e} \leftarrow \text{COracle}(\bar{\boldsymbol{w}}_t, A_t \cup \{e\}, B_t)$
13:　　$p_t \leftarrow \arg\max_{e \in [n] \backslash (A_t \cup B_t)} \bar{w}_t(M_t) - \bar{w}_t(\tilde{M}_{t,e})$ 　　　　▷ define $\bar{w}_t(\bot) = -\infty$; break ties arbitrarily
14:　　**if** $p_t \in M_t$ **then**
15:　　　　$A_{t+1} \leftarrow A_t \cup \{p_t\}, B_{t+1} \leftarrow B_t$
16:　　**else**
17:　　　　$A_{t+1} \leftarrow A_t, B_{t+1} \leftarrow B_t \cup \{p_t\}$
18: $\mathsf{Out} \leftarrow A_{n+1}$
19: **return** $\mathsf{Out}$
---

## 6 Related Work

The multi-armed bandit problem has been extensively studied in both stochastic and adversarial settings [22, 3, 2]. We refer readers to [5] for a survey on recent advances. Many work in MABs focus on minimizing the cumulative regret, which is an objective known to be fundamentally different from the objective of pure exploration MABs [6]. Among these work, a recent line of research considers a generalized setting called combinatorial bandits in which a set of arms (satisfying certain combinatorial constraints) are played on each round [9, 17, 25, 7, 10, 14, 23, 21]. Note that the objective of these work is to minimize the cumulative regret, which differs from ours.

In the literature of pure exploration MABs, the classical problem of identifying the single best arm has been well-studied in both fixed confidence and fixed budget settings [24, 11, 6, 1, 13, 15, 16]. A flurry of recent work extend this classical problem to TOPK and MB problems and obtain algorithms with upper bounds [18, 12, 13, 19, 8, 20, 31] and worst-case lower bounds of TOPK [19, 31]. Our framework encompasses these two problems as special cases and covers a much larger class of combinatorial pure exploration problems, which have not been addressed in current literature. Applying our results back to TOPK and MB, our upper bounds match best available problem-dependent bounds up to constant factors [13, 19, 8] in both fixed confidence and fixed budget settings; and our lower bound is the first proven problem-dependent lower bound for these two problems, which are conjectured earlier by Bubeck et al. [8].

## 7 Conclusion

In this paper, we proposed a general framework called combinatorial pure exploration (CPE) that can handle pure exploration tasks for many complex bandit problems with combinatorial constraints, and have potential applications in various domains. We have shown a number of results for the framework, including two novel learning algorithms, their related upper bounds and a novel lower bound. The proposed algorithms support a wide range of decision classes in a unifying way and our analysis introduced a novel tool called exchange class, which may be of independent interest. Our upper and lower bounds characterize the complexity of the CPE problem: the sample complexity of our algorithm is optimal (up to a logarithmic factor) for the decision classes derived from matroids (including TOPK and MB), while for general decision classes, our upper and lower bounds are within a relatively benign factor.

**Acknowledgments.** The work described in this paper was partially supported by the National Grand Fundamental Research 973 Program of China (No. 2014CB340401 and No. 2014CB340405), the Research Grants Council of the Hong Kong Special Administrative Region, China (Project No. CUHK 413212 and CUHK 415113), and Microsoft Research Asia Regional Seed Fund in Big Data Research (Grant No. FY13-RES-SPONSOR-036).

## Footnotes

*This work was done when the first two authors were interns at Microsoft Research Asia.

[1] We define $v(S) \triangleq \sum_{i \in S} v(i)$ for any vector $\boldsymbol{v} \in \mathbb{R}^n$ and any set $S \subseteq [n]$. In addition, for convenience, we will assume that $M_*$ is unique.

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
