[Supplementary Material]

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

8: $\quad$ **if** $M_t = \bot$ **then**
9: $\quad\quad$ **fail:** set $\text{Out} \leftarrow \bot$ and **return** Out
10: $\quad$ **for each** $e \in [n] \backslash (A_t \cup B_t)$ **do**
11: $\quad\quad$ **if** $e \in M_t$ **then** $\tilde{M}_{t,e} \leftarrow \text{COracle}(\bar{\boldsymbol{w}}_t, A_t, B_t \cup \{e\})$
12: $\quad\quad$ **else** $\tilde{M}_{t,e} \leftarrow \text{COracle}(\bar{\boldsymbol{w}}_t, A_t \cup \{e\}, B_t)$
13: $\quad p_t \leftarrow \arg\max_{e \in [n] \backslash (A_t \cup B_t)} \bar{w}_t(M_t) - \bar{w}_t(\tilde{M}_{t,e})$ $\quad\quad$ ▷ define $\bar{w}_t(\bot) = -\infty$; break ties arbitrarily
14: $\quad$ **if** $p_t \in M_t$ **then**
15: $\quad\quad A_{t+1} \leftarrow A_t \cup \{p_t\}, B_{t+1} \leftarrow B_t$
16: $\quad$ **else**
17: $\quad\quad A_{t+1} \leftarrow A_t, B_{t+1} \leftarrow B_t \cup \{p_t\}$
18: $\text{Out} \leftarrow A_{n+1}$
19: **return** Out

---

## 6 Related Work

The multi-armed bandit problem has been extensively studied in both stochastic and adversarial settings [22, 3, 2]. We refer readers to [5] for a survey on recent advances. Many work in MABs focus on minimizing the cumulative regret, which is an objective known to be fundamentally different from the objective of pure exploration MABs [6]. Among these work, a recent line of research considers a generalized setting called combinatorial bandits in which a set of arms (satisfying certain combinatorial constraints) are played on each round [9, 17, 25, 7, 10, 14, 23, 21]. Note that the objective of these work is to minimize the cumulative regret, which differs from ours.

In the literature of pure exploration MABs, the classical problem of identifying the single best arm has been well-studied in both fixed confidence and fixed budget settings [24, 11, 6, 1, 13, 15, 16]. A flurry of recent work extend this classical problem to TOPK and MB problems and obtain algorithms with upper bounds [18, 12, 13, 19, 8, 20, 31] and worst-case lower bounds of TOPK [19, 31]. Our framework encompasses these two problems as special cases and covers a much larger class of combinatorial pure exploration problems, which have not been addressed in current literature. Applying our results back to TOPK and MB, our upper bounds match best available problem-dependent bounds up to constant factors [13, 19, 8] in both fixed confidence and fixed budget settings; and our lower bound is the first proven problem-dependent lower bound for these two problems, which are conjectured earlier by Bubeck et al. [8].

## 7 Conclusion

In this paper, we proposed a general framework called combinatorial pure exploration (CPE) that can handle pure exploration tasks for many complex bandit problems with combinatorial constraints, and have potential applications in various domains. We have shown a number of results for the framework, including two novel learning algorithms, their related upper bounds and a novel lower bound. The proposed algorithms support a wide range of decision classes in a unifying way and our analysis introduced a novel tool called exchange class, which may be of independent interest. Our upper and lower bounds characterize the complexity of the CPE problem: the sample complexity of our algorithm is optimal (up to a logarithmic factor) for the decision classes derived from matroids (including TOPK and MB), while for general decision classes, our upper and lower bounds are within a relatively benign factor.

**Acknowledgments.** The work described in this paper was partially supported by the National Grand Fundamental Research 973 Program of China (No. 2014CB340401 and No. 2014CB340405), the Research Grants Council of the Hong Kong Special Administrative Region, China (Project No. CUHK 413212 and CUHK 415113), and Microsoft Research Asia Regional Seed Fund in Big Data Research (Grant No. FY13-RES-SPONSOR-036).

## Footnotes

*This work was done when the first two authors were interns at Microsoft Research Asia.

[1] We define $v(S) \triangleq \sum_{i \in S} v(i)$ for any vector $\boldsymbol{v} \in \mathbb{R}^n$ and any set $S \subseteq [n]$. In addition, for convenience, we will assume that $M_*$ is unique.

[2]We notice that the definition of Zhou et al. [31] allow an $(\epsilon', \delta)$-PAC algorithm to produce an output with *average* sub-optimality of $\epsilon'$. This is equivalent to our definition of $(\epsilon, \delta)$-PAC algorithm with $\epsilon = K\epsilon'$ for the TopK problem. In this paper, we translate their guarantees to our definition of PAC algorithm.

[3] The three axioms of matroid are (1) $\emptyset \in \mathcal{I}$ and $\mathcal{I} \neq \{\emptyset\}$; (2) Every subsets of an independent set are independent (heredity property); (3) For all $A, B \in \mathcal{I}$ such that $|B| = |A| + 1$ there exists an element $e \in B \backslash A$ such that $A \cup \{e\} \in \mathcal{I}$ (augmentation property). We refer interested readers to [26] for a general introduction to the matroid theory.

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

## Organization

This supplementary material is organized as follows. First, we provide the deferred proofs of Theorem 1 (Appendix A). Then, we present two simple extensions of CLUCB: one to the fixed budget setting and one to the PAC learning setting; and provide related analysis (Appendix B). Next, we provide the deferred proof of our lower bound (Theorem 2) in Appendix C. In Appendix D, we provide the deferred proof of Theorem 3. We also analyze the uniform allocation algorithm as a simple benchmark in Appendix E. Afterwards, we provide the deferred discussion and constructions of example decision classes and exchange classes in Appendix F. In Appendix G, we prove the equivalence between the constrained oracles and maximization oracles. Finally, we present some preliminary experimental results in Appendix H.

## A   Analysis of CLUCB (Theorem 1)

In this section, we analyze the sample complexity of CLUCB and prove Theorem 1.

**Notations.** Fix some decision class $\mathcal{M} \subseteq 2^{[n]}$ and fix some expected reward vector $\boldsymbol{w} \in \mathbb{R}^n$. Recall that $M_* = \arg\max_{M \in \mathcal{M}} w(M)$ is the optimal set. Since we assume that $M_*$ is unique, one can verify that, for every $e \in [n]$, the gap defined in Eq. (1) is positive, i.e., $\Delta_e > 0$.

We will also need some additional notations for our analysis. For any set $a \subseteq [n]$, let $\boldsymbol{\chi}_a \in \{0,1\}^n$ denote the incidence vector of set $a \subseteq [n]$, i.e., $\chi_a(e) = 1$ if and only if $e \in a$. For an exchange set $b = (b_+, b_-)$, we define $\boldsymbol{\chi}_b \triangleq \boldsymbol{\chi}_{b_+} - \boldsymbol{\chi}_{b_-}$ as the incidence vector of $b$. We notice that $\boldsymbol{\chi}_b \in \{-1, 0, 1\}^n$.

For each round $t$, we define vector $\mathbf{rad}_t = \left( \mathrm{rad}_t(1), \dots, \mathrm{rad}_t(n) \right)^T$ and recall that $\bar{\boldsymbol{w}}_t \in \mathbb{R}^n$ is the empirical mean rewards of arms up to round $t$.

Let $\boldsymbol{u} \in \mathbb{R}^n$ and $\boldsymbol{v} \in \mathbb{R}^n$ be two vectors. Let $\langle \boldsymbol{u}, \boldsymbol{v} \rangle$ denote the inner product of $\boldsymbol{u}$ and $\boldsymbol{v}$. We define $\boldsymbol{u} \circ \boldsymbol{v} \triangleq \left( u(1) \cdot v(1), \dots, u(n) \cdot v(n) \right)^T$ as the element-wise product of $\boldsymbol{u}$ and $\boldsymbol{v}$. For any $s \in \mathbb{R}$, we also define $\boldsymbol{u}^s \triangleq \left( u(1)^s, \dots, u(n)^s \right)^T$ as the element-wise exponentiation of $\boldsymbol{u}$. We let $|\boldsymbol{u}| = \left( |u(1)|, \dots, |u(n)| \right)^T$ denote the element-wise absolute value of $\boldsymbol{u}$.

Finally, let us recall that for any exchange class $b = (b_+, b_-)$ and any set $M \subseteq [n]$, we have defined $M \oplus b = M \backslash b_- \cup b_+$ and $M \ominus b = M \backslash b_+ \cup b_-$.

### A.1   Preparatory Lemmas

Let us begin with a simple lemma that characterizes the incidence vectors of exchange sets.

**Lemma 1.** *Let $M_1 \subseteq [n]$ be a set. Let $b = (b_+, b_-)$ be an exchange set such that $b_- \subseteq M_1$ and $b_+ \cap M_1 = \emptyset$. Define $M_2 = M_1 \oplus b$. Then, we have*

$$\boldsymbol{\chi}_{M_1} + \boldsymbol{\chi}_b = \boldsymbol{\chi}_{M_2}.$$

*In addition, we have $M_1 = M_2 \ominus b$.*

*Proof.* Recall that $M_2 = M_1 \backslash b_- \cup b_+$ and $b_+ \cap b_- = \emptyset$. Therefore we see that $M_2 \backslash M_1 = b_+$ and $M_1 \backslash M_2 = b_-$. We can decompose $\boldsymbol{\chi}_{M_1}$ as $\boldsymbol{\chi}_{M_1} = \boldsymbol{\chi}_{M_1 \backslash M_2} + \boldsymbol{\chi}_{M_1 \cap M_2}$. Hence, we have

$$\begin{aligned}
\boldsymbol{\chi}_{M_1} + \boldsymbol{\chi}_b &= \boldsymbol{\chi}_{M_1 \backslash M_2} + \boldsymbol{\chi}_{M_1 \cap M_2} + \boldsymbol{\chi}_{b_+} - \boldsymbol{\chi}_{b_-} \\
&= \boldsymbol{\chi}_{M_1 \cap M_2} + \boldsymbol{\chi}_{M_2 \backslash M_1} \\
&= \boldsymbol{\chi}_{M_2}.
\end{aligned}$$

Using the definition of operator $\ominus$, one can verify that $M_1 = M_2 \ominus b$. $\qquad\square$

The next lemma serves as a basic tool derived from exchange classes, which allows us to interpolate between different members of a decision class $\mathcal{M}$. Moreover, it characterizes the relationship between gaps and exchange sets. In Figure 1, we illustrate the intuitions of the interpolations characterized in Lemma 2.

| (a) | (b) |

Figure 1: Illustration of Lemma 2: (a) A Venn diagram for the relationships among $M, M', b_-$ and $b_+$. Note that $e \in b_- \cup b_+$. (b) An illustration for the relationships among $M$, $M'$, $M \oplus b$ and $M' \ominus b$. We recall that $M \oplus b = M \backslash b_- \cup b_+$ and $M' \ominus b = M' \backslash b_+ \cup b_-$. We use dotted line to represent an application of Lemma 2 between two sets.

**Lemma 2** (Interpolation Lemma). *Let $\mathcal{M} \subseteq 2^{[n]}$ and let $\mathcal{B}$ be an exchange class for $\mathcal{M}$. Then, for any two different members $M, M'$ of $\mathcal{M}$ and any $e \in (M \backslash M') \cup (M' \backslash M)$, there exists an exchange set $b = (b_+, b_-) \in \mathcal{B}$ which satisfies five constraints: (a) $e \in (b_+ \cup b_-)$, (b) $b_- \subseteq (M \backslash M')$, (c) $b_+ \subseteq (M' \backslash M)$, (d) $(M \oplus b) \in \mathcal{M}$ and (e) $(M' \ominus b) \in \mathcal{M}$. Moreover, if $M' = M_*$, then we have $\langle w, \chi_b \rangle \geq \Delta_e$, where $\Delta_e$ is the gap defined in Eq. (1).*

*Proof.* We decompose our proof into two cases.

**Case (1):** $e \in M \backslash M'$.

By the definition of exchange class, we know that there exists $b = (b_+, b_-) \in \mathcal{B}$ which satisfies that $e \in b_-$, $b_- \subseteq (M \backslash M')$, $b_+ \subseteq (M' \backslash M)$, $(M \oplus b) \in \mathcal{M}$ and $(M' \ominus b) \in \mathcal{M}$. Therefore the five constraints are satisfied.

Next, if $M' = M_*$, we see that $e \notin M_*$. Let us consider the set $M_1 = \arg\max_{S \in \mathcal{M}: e \in S} w(S)$. Note that, by the definition of gaps (Eq. (1)), one has $w(M_*) - w(M_1) = \Delta_e$. Now we define $M_0 = M_* \ominus b$. Note that we already have $M_* \ominus b \in \mathcal{M}$. By combining this with the fact that $e \in M_0$, we see that $w(M_0) \leq \max_{S \in \mathcal{M}: e \in S} w(S) = w(M_1)$. Therefore, we obtain that $w(M_*) - w(M_0) \geq w(M_*) - w(M_1) = \Delta_e$. Notice that the left-hand side of the former inequality can be rewritten using Lemma 1 as follows

$$w(M_*) - w(M_0) = \langle w, \chi_{M_*} \rangle - \langle w, \chi_{M_0} \rangle = \langle w, \chi_{M_*} - \chi_{M_0} \rangle = \langle w, \chi_b \rangle .$$

Therefore, we obtain $\langle w, \chi_b \rangle \geq \Delta_e$.

**Case (2):** $e \in M' \backslash M$.

Using the definition of exchange class, we see that there exists $c = (c_+, c_-) \in \mathcal{B}$ such that $e \in c_-$, $c_- \subseteq (M' \backslash M)$, $c_+ \subseteq (M \backslash M')$, $(M' \oplus c) \in \mathcal{M}$ and $(M \ominus c) \in \mathcal{M}$.

We construct $b = (b_+, b_-)$ by setting $b_+ = c_-$ and $b_- = c_+$. Notice that, by the construction of $b$, we have $M \oplus b = M \ominus c$ and $M' \ominus b = M' \oplus c$. Therefore, it is clear that $b$ satisfies the five constraints of the lemma.

Now, suppose that $M' = M_*$. In this case, we have $e \in M_*$. Consider the set $M_3 = \arg\max_{S \in \mathcal{M}: e \notin S} w(S)$. By definition of $\Delta_e$, we see that $w(M_*) - w(M_3) = \Delta_e$. Now we define $M_2 = M_* \ominus b$ and notice that $M_2 \in \mathcal{M}$. By combining with the fact that $e \notin M_2$, we obtain that $w(M_2) \leq \max_{S \in \mathcal{M}: e \notin S} = w(M_3)$. Hence, we have $w(M_*) - w(M_2) \geq w(M_*) - w(M_3) = \Delta_e$. Similar to Case (1), applying Lemma 1 again, we have

$$\langle w, \chi_b \rangle = w(M_*) - w(M_2) \geq \Delta_e.$$

$\square$

Next we state two basic lemmas that help us to convert set-theoretical arguments to linear algebraic arguments.

**Lemma 3.** *Let $M, M' \subseteq [n]$ be two sets. Let $\mathbf{rad}_t$ be an $n$-dimensional vector with nonnegative entries. Then, we have*

$$\max_{e \in (M \backslash M') \cup (M' \backslash M)} \mathrm{rad}_t(e) = \left\| \mathbf{rad}_t \circ |\chi_{M'} - \chi_M| \right\|_\infty.$$

*Proof.* Notice that $\boldsymbol{\chi}_{M'} - \boldsymbol{\chi}_M = \boldsymbol{\chi}_{M'\setminus M} - \boldsymbol{\chi}_{M\setminus M'}$. In addition, since $(M'\setminus M) \cap (M\setminus M') = \emptyset$, we have $\boldsymbol{\chi}_{M'\setminus M} \circ \boldsymbol{\chi}_{M\setminus M'} = \mathbf{0}_n$. Also notice that $\boldsymbol{\chi}_{M'\setminus M} - \boldsymbol{\chi}_{M\setminus M'} \in \{-1, 0, 1\}^n$. Therefore, we have

$$
\begin{aligned}
|\boldsymbol{\chi}_{M'\setminus M} - \boldsymbol{\chi}_{M\setminus M'}| &= (\boldsymbol{\chi}_{M'\setminus M} - \boldsymbol{\chi}_{M\setminus M'})^2 \\
&= \boldsymbol{\chi}_{M'\setminus M}^2 + \boldsymbol{\chi}_{M\setminus M'}^2 - 2\boldsymbol{\chi}_{M'\setminus M} \circ \boldsymbol{\chi}_{M\setminus M'} \\
&= \boldsymbol{\chi}_{M'\setminus M} + \boldsymbol{\chi}_{M\setminus M'} \\
&= \boldsymbol{\chi}_{(M'\setminus M)\cup(M\setminus M')},
\end{aligned}
$$

where the third equation follows from the fact that $\boldsymbol{\chi}_{M\setminus M'} \in \{0, 1\}^n$ and $\boldsymbol{\chi}_{M'\setminus M} \in \{0, 1\}^n$. The lemma follows immediately from the fact that $\mathrm{rad}_t(e) \geq 0$ and $\boldsymbol{\chi}_{(M\setminus M')\cup(M'\setminus M)} \in \{0, 1\}^n$. $\square$

**Lemma 4.** *Let $\boldsymbol{a}, \boldsymbol{b}, \boldsymbol{c} \in \mathbb{R}^n$ be three vectors. Then, we have $\langle \boldsymbol{a}, \boldsymbol{b} \circ \boldsymbol{c} \rangle = \langle \boldsymbol{a} \circ \boldsymbol{b}, \boldsymbol{c} \rangle$.*

*Proof.* We have

$$
\langle \boldsymbol{a}, \boldsymbol{b} \circ \boldsymbol{c} \rangle = \sum_{i=1}^n a(i)\big(b(i)c(i)\big) = \sum_{i=1}^n \big(a(i)b(i)\big)c(i) = \langle \boldsymbol{a} \circ \boldsymbol{b}, \boldsymbol{c} \rangle.
$$

$\square$

The next lemma characterizes the property of $\tilde{\boldsymbol{w}}_t$ which is defined in the CLUCB algorithm.

**Lemma 5.** *Let $M_t$, $\tilde{\boldsymbol{w}}_t$ and $\mathbf{rad}_t$ be defined in Algorithm 1 and Theorem 1. Let $M' \in \mathcal{M}$ be an arbitrary member of decision class. We have*

$$
\tilde{w}_t(M') - \tilde{w}_t(M_t) = \langle \tilde{\boldsymbol{w}}_t, \boldsymbol{\chi}_{M'} - \boldsymbol{\chi}_{M_t} \rangle = \langle \bar{\boldsymbol{w}}_t, \boldsymbol{\chi}_{M'} - \boldsymbol{\chi}_{M_t} \rangle + \langle \mathbf{rad}_t, |\boldsymbol{\chi}_{M'} - \boldsymbol{\chi}_{M_t}| \rangle.
$$

*Proof.* We begin with proving the first part. It is easy to verify that $\tilde{\boldsymbol{w}}_t = \bar{\boldsymbol{w}}_t + \mathbf{rad}_t \circ (\mathbf{1}_n - 2\boldsymbol{\chi}_{M_t})$. Then, we have

$$
\begin{aligned}
\langle \tilde{\boldsymbol{w}}_t, \boldsymbol{\chi}_{M'} - \boldsymbol{\chi}_{M_t} \rangle &= \langle \bar{\boldsymbol{w}}_t + \mathbf{rad}_t \circ (\mathbf{1}_n - 2\boldsymbol{\chi}_{M_t}), \ \boldsymbol{\chi}_{M'} - \boldsymbol{\chi}_{M_t} \rangle \\
&= \langle \bar{\boldsymbol{w}}_t, \boldsymbol{\chi}_{M'} - \boldsymbol{\chi}_{M_t} \rangle + \langle \mathbf{rad}_t, (\mathbf{1}_n - 2\boldsymbol{\chi}_{M_t}) \circ (\boldsymbol{\chi}_{M'} - \boldsymbol{\chi}_{M_t}) \rangle && (9) \\
&= \langle \bar{\boldsymbol{w}}_t, \boldsymbol{\chi}_{M'} - \boldsymbol{\chi}_{M_t} \rangle + \langle \mathbf{rad}_t, \boldsymbol{\chi}_{M'} - \boldsymbol{\chi}_{M_t} - 2\boldsymbol{\chi}_{M_t} \circ \boldsymbol{\chi}_{M'} + 2\boldsymbol{\chi}_{M_t}^2 \rangle \\
&= \langle \bar{\boldsymbol{w}}_t, \boldsymbol{\chi}_{M'} - \boldsymbol{\chi}_{M_t} \rangle + \langle \mathbf{rad}_t, \boldsymbol{\chi}_{M'}^2 - \boldsymbol{\chi}_{M_t}^2 - 2\boldsymbol{\chi}_{M_t} \circ \boldsymbol{\chi}_{M'} + 2\boldsymbol{\chi}_{M_t}^2 \rangle && (10) \\
&= \langle \bar{\boldsymbol{w}}_t, \boldsymbol{\chi}_{M'} - \boldsymbol{\chi}_{M_t} \rangle + \langle \mathbf{rad}_t, (\boldsymbol{\chi}_{M'} - \boldsymbol{\chi}_{M_t})^2 \rangle \\
&= \langle \bar{\boldsymbol{w}}_t, \boldsymbol{\chi}_{M'} - \boldsymbol{\chi}_{M_t} \rangle + \langle \mathbf{rad}_t, |\boldsymbol{\chi}_{M'} - \boldsymbol{\chi}_{M_t}| \rangle, && (11)
\end{aligned}
$$

where Eq. (9) follows from Lemma 4; Eq. (10) holds since $\boldsymbol{\chi}_{M'} \in \{0, 1\}^n$ and $\boldsymbol{\chi}_{M_t} \in \{0, 1\}^n$ and therefore $\boldsymbol{\chi}_{M'} = \boldsymbol{\chi}_{M'}^2$ and $\boldsymbol{\chi}_{M_t} = \boldsymbol{\chi}_{M_t}^2$; and Eq. (11) follows since $\boldsymbol{\chi}_{M'} - \boldsymbol{\chi}_{M_t} \in \{-1, 0, 1\}^n$. $\square$

## A.2 Confidence Intervals

First, we recall a standard concentration inequality of sub-Gaussian random variables.

**Lemma 6** (Hoeffding's inequality). *Let $X_1, \ldots, X_n$ be $n$ independent random variables such that, for each $i \in [n]$, random variable $X_i - \mathbb{E}[X_i]$ is $R$-sub-Gaussian distributed, i.e., $\forall t \in \mathbb{R}$, $\mathbb{E}[\exp(tX_i - t\mathbb{E}[X_i])] \leq \exp(R^2 t^2/2)$. Let $\bar{X} = \frac{1}{n} \sum_{i=1}^n X_i$ denote the average of these random variables. Then, for any $\lambda > 0$, we have*

$$
\Pr\left[|\bar{X} - \mathbb{E}[\bar{X}]| \geq \lambda\right] \leq 2\exp\left(-\frac{n\lambda^2}{2R^2}\right).
$$

*Proof.* For all $i \in [n]$, we define $v_i = X_i - \mathbb{E}[X_i]$. We also define $S = \sum_{i=1}^n v_i$ and $\epsilon = n\lambda$. Therefore, for any $t > 0$, we have

$$\Pr[S \geq \epsilon] = \Pr[tS \geq t\epsilon] \overset{(a)}{\leq} \frac{\mathbb{E}[\exp(tS)]}{\exp(t\epsilon)} = \frac{\mathbb{E}[\exp(\sum_{i=1}^n tv_i)]}{\exp(t\epsilon)}$$

$$\overset{(b)}{=} \frac{\prod_{i=1}^n \mathbb{E}[\exp(tv_i)]}{\exp(t\epsilon)} \leq \frac{\prod_{i=1}^n \exp\left(R^2 t^2/2\right)}{\exp(t\epsilon)} = \exp(nR^2t^2/2 - t\epsilon),$$

where (a) follows from Markov's inequality and (b) holds since $v_1, \ldots, v_n$ are independent. Now minimizing over $t > 0$, we get

$$\Pr[S \geq \epsilon] \leq \inf_{t>0} \exp(nR^2t^2/2 - t\epsilon) = \exp(-\epsilon^2/2nR^2) = \exp(-n\lambda^2/2R^2).$$

Similarly, one can show that $\Pr[S \leq -\epsilon] \leq \exp(-n\lambda^2/2R^2)$. Hence, the lemma follows from a union bound. $\qquad\square$

Next, for all $t > 0$, we define random event $\xi_t$ as follows

$$\xi_t = \left\{ \forall i \in [n], \quad |w(i) - \bar{w}_t(i)| < \mathrm{rad}_t(i) \right\}. \tag{12}$$

We notice that random event $\xi_t$ characterizes the event that the confidence bounds of all arms are valid at round $t$.

If the confidence bounds are valid, we can generalize Eq. (12) to inner products as follows.

**Lemma 7.** *Given any $t > 0$, assume that event $\xi_t$ as defined in Eq. (12) occurs. Then, for any vector $\boldsymbol{a} \in \mathbb{R}^n$, we have*

$$\left| \langle \boldsymbol{w}, \boldsymbol{a} \rangle - \langle \bar{\boldsymbol{w}}_t, \boldsymbol{a} \rangle \right| < \langle \mathbf{rad}_t, |\boldsymbol{a}| \rangle.$$

*Proof.* Suppose that $\xi_t$ occurs. Then, we have

$$\left| \langle \boldsymbol{w}, \boldsymbol{a} \rangle - \langle \bar{\boldsymbol{w}}_t, \boldsymbol{a} \rangle \right| = \left| \langle \boldsymbol{w} - \bar{\boldsymbol{w}}_t, \boldsymbol{a} \rangle \right|$$

$$= \left| \sum_{i=1}^n \left( w(i) - \bar{w}_t(i) \right) a(i) \right|$$

$$\leq \sum_{i=1}^n \left| w(i) - \bar{w}_t(i) \right| |a(i)|$$

$$< \sum_{i=1}^n \mathrm{rad}_t(i) \cdot |a(i)| \tag{13}$$

$$= \langle \mathbf{rad}_t, |\boldsymbol{a}| \rangle,$$

where Eq. (13) follows the definition of event $\xi_t$ in Eq. (12) and the assumption that it occurs. $\qquad\square$

Next, we construct the high probability confidence intervals for the fixed confidence setting.

**Lemma 8.** *Suppose that the reward distribution $\varphi_e$ is a $R$-sub-Gaussian distribution for all $e \in [n]$. And if, for all $t > 0$ and all $e \in [n]$, the confidence radius $\mathrm{rad}_t(e)$ is given by*

$$\mathrm{rad}_t(e) = R\sqrt{\frac{2\log\left(\frac{4nt^3}{\delta}\right)}{T_t(e)}},$$

*where $T_t(e)$ is the number of samples of arm $e$ up to round $t$. Then, we have*

$$\Pr\left[\bigcap_{t=1}^{\infty} \xi_t\right] \geq 1 - \delta.$$

*Proof.* Fix any $t > 0$ and $e \in [n]$. Note that $\varphi_e$ is a $R$-sub-Gaussian tail distribution with mean $w(e)$ and $\bar{w}_t(e)$ is the empirical mean of $\varphi_e$ from $T_t(e)$ samples. Then, we have

$$\Pr\left[|\bar{w}_t(e) - w(e)| \geq R\sqrt{\frac{2\log\left(\frac{4nt^3}{\delta}\right)}{T_t(e)}}\right] = \sum_{s=1}^{t-1} \Pr\left[|\bar{w}_t(e) - w(e)| \geq R\sqrt{\frac{2\log\left(\frac{4nt^3}{\delta}\right)}{s}}, T_t(e) = s\right] \tag{14}$$

$$\leq \sum_{s=1}^{t-1} \frac{\delta}{2nt^3} \tag{15}$$

$$\leq \frac{\delta}{2nt^2},$$

where Eq. (14) follows from the fact that $1 \leq T_t(e) \leq t - 1$ and Eq. (15) follows from Hoeffding's inequality (Lemma 6). By a union bound over all $e \in [n]$, we see that $\Pr[\xi_t] \geq 1 - \frac{\delta}{2t^2}$. Using a union bound again over all $t > 0$, we have

$$\Pr\left[\bigcap_{t=1}^{\infty} \xi_t\right] \geq 1 - \sum_{t=1}^{\infty} \Pr[\neg\xi_t]$$

$$\geq 1 - \sum_{t=1}^{\infty} \frac{\delta}{2t^2}$$

$$= 1 - \frac{\pi^2}{12}\delta \geq 1 - \delta.$$

$\square$

## A.3 Main Lemmas

Now we state our key technical lemmas. In these lemmas, we shall use Lemma 2 to construct gadgets that interpolate between different members of a decision class. The first lemma shows that, if the confidence intervals are valid, then CLUCB always returns the correct answer when it stops.

**Lemma 9.** *Given any $t > n$, assume that event $\xi_t$ (defined in Eq. (12)) occurs. Then, if Algorithm 1 terminates at round t, we have $M_t = M_*$.*

*Proof.* Suppose that $M_t \neq M_*$. By the assumption that $M_*$ is the unique optimal set, we have $w(M_*) > w(M_t)$. Rewriting this inequality, we obtain that $\langle w, \chi_{M_*} \rangle > \langle w, \chi_{M_t} \rangle$.

Let $\mathcal{B}$ be an exchange class for $\mathcal{M}$. Applying Lemma 2 by setting $M = M_t$ and $M' = M_*$, we see that there exists $b = (b_+, b_-) \in \mathcal{B}$ such that $(M_t \oplus b) \in \mathcal{M}$ and $\langle w, \chi_b \rangle > 0$.

Now define $M'_t = M_t \oplus b$. Recall that $\tilde{M}_t = \arg\max_{M \in \mathcal{M}} \tilde{w}_t(M)$ and therefore $\tilde{w}_t(\tilde{M}_t) \geq \tilde{w}_t(M'_t)$. Hence, we have

$$\tilde{w}_t(\tilde{M}_t) - \tilde{w}_t(M_t) \geq \tilde{w}_t(M'_t) - \tilde{w}_t(M_t)$$

$$= \left\langle \bar{w}_t, \chi_{M'_t} - \chi_{M_t} \right\rangle + \left\langle \mathbf{rad}_t, |\chi_{M'} - \chi_{M_t}| \right\rangle \tag{16}$$

$$\geq \left\langle w, \chi_{M'_t} - \chi_{M_t} \right\rangle \tag{17}$$

$$= \langle w, \chi_b \rangle > 0, \tag{18}$$

where Eq. (16) follows from Lemma 5; and Eq. (17) follows the assumption that event $\xi_t$ occurs and Lemma 7.

Therefore Eq. (18) shows that $\tilde{w}_t(\tilde{M}_t) > \tilde{w}_t(M_t)$. However, this contradicts to the stopping condition of CLUCB: $\tilde{w}_t(\tilde{M}_t) = \tilde{w}_t(M_t)$ and the assumption that the algorithm terminates on round $t$. $\square$

The next lemma shows that if the confidence interval of an arm is sufficiently small, then this arm will not be played by the algorithm. In the proof, we construct a number of gadgets using Lemma 2. We illustrate the relationships among the gadgets in Figure 2.

(a) Gadgets $M_t'$ and $M_t' \oplus b$ for Case (1)    (b) Gadget $\tilde{M}_t \oplus b$ for Case (2)

Figure 2: An illustration of the relationship among the gadgets used the proof of Lemma 10; We use dotted line to represent an application of Lemma 2 between two sets.

**Lemma 10.** *Given any $t > 0$ and suppose that event $\xi_t$ (defined in Eq. (12)) occurs. For any $e \in [n]$, if $\operatorname{rad}_t(e) < \frac{\Delta_e}{3\operatorname{width}(\mathcal{M})}$, then, arm $e$ will not be pulled on round $t$, i.e., $p_t \neq e$.*

*Proof.* We prove by contradiction. Therefore we shall assume the opposite that $p_t = e$ in the rest of the proof.

First let us fix an exchange class $\mathcal{B} \in \arg\min_{\mathcal{B}' \in \operatorname{Exchange}(\mathcal{M})} \operatorname{width}(\mathcal{B}')$. Note that $\operatorname{width}(\mathcal{B}) = \operatorname{width}(\mathcal{M})$. By Lemma 2, there exists an exchange set $c = (c_+, c_-) \in \mathcal{B}$ such that $e \in (c_+ \cup c_-)$, $c_- \subseteq (M_t \backslash \tilde{M}_t)$, $c_+ \subseteq (\tilde{M}_t \backslash M_t)$, $(M_t \oplus c) \in \mathcal{M}$ and $(\tilde{M}_t \ominus c) \in \mathcal{M}$.

Now, we decompose our proof into two cases.

**Case (1):** $(e \in M_* \wedge e \in c_+) \vee (e \notin M_* \wedge e \in c_-)$.

First we construct a gadget $M_t' = \tilde{M}_t \ominus c$ and recall that $M_t' \in \mathcal{M}$. By the definitions of $\ominus$ and $\oplus$, we see that $\tilde{M}_t = M_t' \oplus c$.

We claim that $M_t' \neq M_*$. The assumption of Case (1) means that either (a) $e \in M_*$ and $e \in c_+$; or (b) $e \notin M_*$ and $e \in c_-$ holds. Suppose that $e \in M_*$ and $e \in c_+$. Then, we see that $e \notin M_t'$ and hence $M_t' \neq M_*$. On the other hand, if $e \notin M_*$ and $e \in c_-$, then $e \in M_t'$ which also means that $M_t' \neq M_*$. Therefore we have $M_t' \neq M_*$ in either cases.

Next, we apply Lemma 2 by setting $M = M_t'$ and $M' = M_*$. We see that there exists an exchange set $b \in \mathcal{B}$ such that, $e \in (b_+ \cup b_-)$, $(M_t' \oplus b) \in \mathcal{M}$ and $\langle \boldsymbol{w}, \boldsymbol{\chi}_b \rangle \geq \Delta_e > 0$. We will also use $M_t' \oplus b$ as a gadget.

Now, we define vectors $\boldsymbol{d} = \boldsymbol{\chi}_{\tilde{M}_t} - \boldsymbol{\chi}_{M_t}$, $\boldsymbol{d}_1 = \boldsymbol{\chi}_{M_t'} - \boldsymbol{\chi}_{M_t}$ and $\boldsymbol{d}_2 = \boldsymbol{\chi}_{M_t' \oplus b} - \boldsymbol{\chi}_{M_t}$. By the definition of $M_t'$ and Lemma 1, we see that $\boldsymbol{d}_1 = \boldsymbol{d} - \boldsymbol{\chi}_c$ and $\boldsymbol{d}_2 = \boldsymbol{d}_1 + \boldsymbol{\chi}_b = \boldsymbol{d} - \boldsymbol{\chi}_c + \boldsymbol{\chi}_b$.

Then, we claim that $\|\mathbf{rad}_t \circ (\boldsymbol{d} - \boldsymbol{\chi}_c)\|_\infty < \frac{\Delta_e}{3\operatorname{width}(\mathcal{B})}$. To prove this claim, we first appeal to standard set-algebraic manipulations. We obtain

$$
\begin{aligned}
M_t \backslash M_t' &= M_t \backslash (\tilde{M}_t \ominus c) \\
&= M_t \backslash (\tilde{M}_t \backslash c_+ \cup c_-) \\
&= M_t \backslash (\tilde{M}_t \backslash c_+) \cap (M_t \backslash c_-) \\
&= (M_t \cap c_+) \cup (M_t \backslash \tilde{M}_t) \cap (M_t \backslash c_-) \\
&= (M_t \backslash \tilde{M}_t) \cap (M_t \backslash c_-) \\
&\subseteq M_t \backslash \tilde{M}_t,
\end{aligned}
$$
(19)
(20)

where Eq. (19) follows from $c_+ \subseteq \tilde{M}_t \backslash M_t$ and therefore $c_+ \cap M_t = \emptyset$. Similarly, we can derive $M_t' \backslash M_t$ as follows

$$
\begin{aligned}
M_t' \backslash M_t &= (\tilde{M}_t \ominus c) \backslash M_t = (\tilde{M}_t \backslash c_+ \cup c_-) \backslash M_t \\
&= \left( (\tilde{M}_t \backslash c_+) \backslash M_t \right) \cup (c_- \backslash M_t) \\
&= \tilde{M}_t \backslash c_+ \backslash M_t
\end{aligned}
$$
(21)

$$\subseteq \tilde{M}_t \backslash M_t, \tag{22}$$

where Eq. (21) follows from $c_- \subseteq M_t \backslash \tilde{M}_t$ and hence $c_- \backslash M_t = \emptyset$. By combining Eq. (20) and Eq. (22), we see that $\big((M_t \backslash M_t') \cup (M_t' \backslash M_t)\big) \subseteq \big((M_t \backslash \tilde{M}_t) \cup (\tilde{M}_t \backslash M_t)\big)$. Then, applying Lemma 3, we obtain

$$
\begin{aligned}
\|\mathbf{rad}_t \circ (\boldsymbol{d} - \boldsymbol{\chi}_c)\|_\infty &= \left\|\mathbf{rad}_t \circ (\boldsymbol{\chi}_{M_t'} - \boldsymbol{\chi}_{M_t})\right\|_\infty \\
&= \max_{i \in (M_t \backslash M_t') \cup (M_t' \backslash M_t)} \mathrm{rad}_t(i) \\
&\leq \max_{i \in (M_t \backslash \tilde{M}_t) \cup (\tilde{M}_t \backslash M_t)} \mathrm{rad}_t(i) \\
&= \mathrm{rad}_t(e) \tag{23} \\
&< \frac{\Delta_e}{3\,\mathrm{width}(\mathcal{B})}, \tag{24}
\end{aligned}
$$

where Eq. (23) follows from the assumption that $p_t = e$.

Next we claim that $\|\mathbf{rad}_t \circ \boldsymbol{\chi}_c\|_\infty < \frac{\Delta_e}{3\,\mathrm{width}(\mathcal{B})}$. Recall that, by the definition of $c$, we have $c_+ \subseteq (\tilde{M}_t \backslash M_t)$ and $c_- \subseteq (M_t \backslash \tilde{M}_t)$. Hence $c_+ \cup c_- \subseteq (\tilde{M}_t \backslash M_t) \cup (M_t \backslash \tilde{M}_t)$. Since $\boldsymbol{\chi}_c \in \{-1, 0, 1\}^n$, we see that

$$
\begin{aligned}
\|\mathbf{rad}_t \circ \boldsymbol{\chi}_c\|_\infty &= \max_{i \in c_+ \cup c_-} \mathrm{rad}_t(i) \\
&\leq \max_{i \in (\tilde{M}_t \backslash M_t) \cup (M_t \backslash \tilde{M}_t)} \mathrm{rad}_t(i) \\
&= \mathrm{rad}_t(e) < \frac{\Delta_e}{3\,\mathrm{width}(\mathcal{B})}. \tag{25}
\end{aligned}
$$

From Eq. (25), we derive

$$
\begin{aligned}
\langle \mathbf{rad}_t, |\boldsymbol{\chi}_c| \rangle &= \langle \mathbf{rad}_t, \boldsymbol{\chi}_c^2 \rangle \tag{26} \\
&= \langle \mathbf{rad}_t \circ \boldsymbol{\chi}_c, \boldsymbol{\chi}_c \rangle \tag{27} \\
&\leq \|\mathbf{rad}_t \circ \boldsymbol{\chi}_c\|_\infty \|\boldsymbol{\chi}_c\|_1 \tag{28} \\
&< \frac{\Delta_e}{3\,\mathrm{width}(\mathcal{B})} \|\boldsymbol{\chi}_c\|_1 \tag{29} \\
&\leq \frac{\Delta_e}{3}, \tag{30}
\end{aligned}
$$

where Eq. (26) hold since $\boldsymbol{\chi}_c \in \{-1, 0, 1\}^n$; Eq. (27) follows form Lemma 4; Eq. (28) follows from Hölder's inequality; Eq. (29) follows from Eq. (25); and Eq. (30) holds since $\|\boldsymbol{\chi}_c\|_1 = |c_+| + |c_-| \leq \mathrm{width}(\mathcal{B})$ where the inequality is due to $c \in \mathcal{B}$ .

Next, we claim that $\boldsymbol{d} \circ \boldsymbol{\chi}_c = |\boldsymbol{\chi}_c|$. Recall that $\boldsymbol{\chi}_c = \boldsymbol{\chi}_{c_+} - \boldsymbol{\chi}_{c_-}$ and $\boldsymbol{d} = \boldsymbol{\chi}_{\tilde{M}_t} - \boldsymbol{\chi}_{M_t} = \boldsymbol{\chi}_{\tilde{M}_t \backslash M_t} - \boldsymbol{\chi}_{M_t \backslash \tilde{M}_t}$. We also notice that $c_+ \subseteq (\tilde{M}_t \backslash M_t)$ and $c_- \subseteq (M_t \backslash \tilde{M}_t)$. This implies that $c_+ \cap (M_t \backslash \tilde{M}_t) = \emptyset$ and $c_- \cap (\tilde{M}_t \backslash M_t) = \emptyset$. Therefore, we have

$$
\begin{aligned}
\boldsymbol{d} \circ \boldsymbol{\chi}_c &= (\boldsymbol{\chi}_{\tilde{M}_t \backslash M_t} - \boldsymbol{\chi}_{M_t \backslash \tilde{M}_t}) \circ (\boldsymbol{\chi}_{c_+} - \boldsymbol{\chi}_{c_-}) \\
&= \boldsymbol{\chi}_{\tilde{M}_t \backslash M_t} \circ \boldsymbol{\chi}_{c_+} + \boldsymbol{\chi}_{M_t \backslash \tilde{M}_t} \circ \boldsymbol{\chi}_{c_-} - \boldsymbol{\chi}_{\tilde{M}_t \backslash M_t} \circ \boldsymbol{\chi}_{c_-} - \boldsymbol{\chi}_{M_t \backslash \tilde{M}_t} \circ \boldsymbol{\chi}_{c_+} \\
&= \boldsymbol{\chi}_{\tilde{M}_t \backslash M_t} \circ \boldsymbol{\chi}_{c_+} + \boldsymbol{\chi}_{M_t \backslash \tilde{M}_t} \circ \boldsymbol{\chi}_{c_-} \\
&= \boldsymbol{\chi}_{c_+} + \boldsymbol{\chi}_{c_-} = |\boldsymbol{\chi}_c|.
\end{aligned}
$$

where the second equality holds since $c_+ \cap (M_t \backslash \tilde{M}_t) = \emptyset$ and $c_- \cap (\tilde{M}_t \backslash M_t) = \emptyset$; and the last equality holds since $c_+ \cap c_- = \emptyset$.

Now, we bound quantity $\langle \mathbf{rad}_t, |\boldsymbol{d}_2| \rangle - \langle \mathbf{rad}_t, |\boldsymbol{d}| \rangle$ as follows

$$
\begin{aligned}
\langle \mathbf{rad}_t, |\boldsymbol{d}_2| \rangle - \langle \mathbf{rad}_t, |\boldsymbol{d}| \rangle &= \langle \mathbf{rad}_t, |\boldsymbol{d}_2| - |\boldsymbol{d}| \rangle = \langle \mathbf{rad}_t, \boldsymbol{d}_2^2 - \boldsymbol{d}^2 \rangle \tag{31} \\
&= \langle \mathbf{rad}_t, (\boldsymbol{d} - \boldsymbol{\chi}_c + \boldsymbol{\chi}_b)^2 - \boldsymbol{d}^2 \rangle
\end{aligned}
$$

$$= \left\langle \mathbf{rad}_t, \boldsymbol{\chi}_b^2 + \boldsymbol{\chi}_c^2 - 2\boldsymbol{\chi}_b \circ \boldsymbol{\chi}_c - 2\boldsymbol{d} \circ \boldsymbol{\chi}_c + 2\boldsymbol{d} \circ \boldsymbol{\chi}_b \right\rangle$$

$$= \left\langle \mathbf{rad}_t, \boldsymbol{\chi}_b^2 - \boldsymbol{\chi}_c^2 + 2\boldsymbol{\chi}_b \circ (\boldsymbol{d} - \boldsymbol{\chi}_c) \right\rangle \tag{32}$$

$$= \left\langle \mathbf{rad}_t, |\boldsymbol{\chi}_b| \right\rangle - \left\langle \mathbf{rad}_t, |\boldsymbol{\chi}_c| \right\rangle - 2 \left\langle \mathbf{rad}_t, \boldsymbol{\chi}_b \circ (\boldsymbol{d} - \boldsymbol{\chi}_c) \right\rangle$$

$$= \left\langle \mathbf{rad}_t, |\boldsymbol{\chi}_b| \right\rangle - \left\langle \mathbf{rad}_t, |\boldsymbol{\chi}_c| \right\rangle - 2 \left\langle \mathbf{rad}_t \circ (\boldsymbol{d} - \boldsymbol{\chi}_c), \boldsymbol{\chi}_b \right\rangle \tag{33}$$

$$\geq \left\langle \mathbf{rad}_t, |\boldsymbol{\chi}_b| \right\rangle - \left\langle \mathbf{rad}_t, |\boldsymbol{\chi}_c| \right\rangle - 2 \left\| \mathbf{rad}_t \circ (\boldsymbol{d} - \boldsymbol{\chi}_c) \right\|_\infty \left\| \boldsymbol{\chi}_b \right\|_1 \tag{34}$$

$$> \left\langle \mathbf{rad}_t, |\boldsymbol{\chi}_b| \right\rangle - \left\langle \mathbf{rad}_t, |\boldsymbol{\chi}_c| \right\rangle - \frac{2\Delta_e}{3 \operatorname{width}(\mathcal{B})} \left\| \boldsymbol{\chi}_b \right\|_1 \tag{35}$$

$$\geq \left\langle \mathbf{rad}_t, |\boldsymbol{\chi}_b| \right\rangle - \left\langle \mathbf{rad}_t, |\boldsymbol{\chi}_c| \right\rangle - \frac{2\Delta_e}{3}, \tag{36}$$

where Eq. (31) holds since $\boldsymbol{d} \in \{-1, 0, 1\}^n$ and $\boldsymbol{d}_2 \in \{-1, 0, 1\}^n$; Eq. (32) follows from the claim that $\boldsymbol{d} \circ \boldsymbol{\chi}_c = |\boldsymbol{\chi}_c| = \boldsymbol{\chi}_c^2$; Eq. (33) and Eq. (34) follow from Lemma 4 and Hölder's inequality; Eq. (35) follows from Eq. (24); and Eq. (36) holds since $b \in \mathcal{B}$ and $\|\boldsymbol{\chi}_b\|_1 = |b_+| + |b_-| \leq \operatorname{width}(\mathcal{B})$.

Applying Lemma 5 by setting $M' = \tilde{M}_t$, we have

$$\left\langle \bar{\boldsymbol{w}}_t, \boldsymbol{d} \right\rangle + \left\langle \mathbf{rad}_t, |\boldsymbol{d}| \right\rangle = \left\langle \bar{\boldsymbol{w}}_t, \boldsymbol{\chi}_{\tilde{M}_t} - \boldsymbol{\chi}_{M_t} \right\rangle + \left\langle \mathbf{rad}_t, |\boldsymbol{\chi}_{\tilde{M}_t} - \boldsymbol{\chi}_{M_t}| \right\rangle$$

$$= \tilde{w}_t(\tilde{M}_t) - \tilde{w}_t(M_t)$$

$$\geq \tilde{w}_t(M'_t \oplus b) - \tilde{w}_t(M_t) \tag{37}$$

$$= \left\langle \bar{\boldsymbol{w}}_t, \boldsymbol{\chi}_{M'_t \oplus b} - \boldsymbol{\chi}_{M_t} \right\rangle + \left\langle \mathbf{rad}_t, |\boldsymbol{\chi}_{M'_t \oplus b} - \boldsymbol{\chi}_{M_t}| \right\rangle$$

$$= \left\langle \bar{\boldsymbol{w}}_t, \boldsymbol{d}_2 \right\rangle + \left\langle \mathbf{rad}_t, |\boldsymbol{d}_2| \right\rangle$$

$$= \left\langle \bar{\boldsymbol{w}}_t, \boldsymbol{d} \right\rangle - \left\langle \bar{\boldsymbol{w}}_t, \boldsymbol{\chi}_c \right\rangle + \left\langle \bar{\boldsymbol{w}}_t, \boldsymbol{\chi}_b \right\rangle + \left\langle \mathbf{rad}_t, |\boldsymbol{d}_2| \right\rangle, \tag{38}$$

where Eq. (37) follows from the fact that $\tilde{w}_t(\tilde{M}_t) = \max_{M \in \mathcal{M}} \tilde{w}_t(M)$; and Eq. (38) follows from the fact that $\boldsymbol{d}_2 = \boldsymbol{d} - \boldsymbol{\chi}_c + \boldsymbol{\chi}_b$. Rearranging the above inequality, we obtain

$$\left\langle \bar{\boldsymbol{w}}_t, \boldsymbol{\chi}_c \right\rangle \geq \left\langle \bar{\boldsymbol{w}}_t, \boldsymbol{\chi}_b \right\rangle + \left\langle \mathbf{rad}_t, |\boldsymbol{d}_2| \right\rangle - \left\langle \mathbf{rad}_t, |\boldsymbol{d}| \right\rangle$$

$$\geq \left\langle \bar{\boldsymbol{w}}_t, \boldsymbol{\chi}_b \right\rangle + \left\langle \mathbf{rad}_t, |\boldsymbol{\chi}_b| \right\rangle - \left\langle \mathbf{rad}_t, |\boldsymbol{\chi}_c| \right\rangle - \frac{2\Delta_e}{3} \tag{39}$$

$$> \left\langle \boldsymbol{w}, \boldsymbol{\chi}_b \right\rangle - \left\langle \mathbf{rad}_t, |\boldsymbol{\chi}_c| \right\rangle - \frac{2\Delta_e}{3} \tag{40}$$

$$> \left\langle \boldsymbol{w}, \boldsymbol{\chi}_b \right\rangle - \frac{\Delta_e}{3} - \frac{2\Delta_e}{3} \tag{41}$$

$$= \left\langle \boldsymbol{w}, \boldsymbol{\chi}_b \right\rangle - \Delta_e \geq 0, \tag{42}$$

where Eq. (39) uses Eq. (36); Eq. (40) follows from the assumption that event $\xi_t$ occurs and Lemma 7; and Eq. (41) holds due to Eq. (30).

We have shown that $\left\langle \bar{\boldsymbol{w}}_t, \boldsymbol{\chi}_c \right\rangle > 0$. Now we can bound $\bar{w}_t(M'_t)$ as follows

$$\bar{w}_t(M'_t) = \left\langle \bar{\boldsymbol{w}}_t, \boldsymbol{\chi}_{M'_t} \right\rangle = \left\langle \bar{\boldsymbol{w}}_t, \boldsymbol{\chi}_{M_t} + \boldsymbol{\chi}_c \right\rangle = \left\langle \bar{\boldsymbol{w}}_t, \boldsymbol{\chi}_{M_t} \right\rangle + \left\langle \bar{\boldsymbol{w}}_t, \boldsymbol{\chi}_c \right\rangle > \left\langle \bar{\boldsymbol{w}}_t, \boldsymbol{\chi}_{M_t} \right\rangle = \bar{w}_t(M_t).$$

However, the definition of $M_t$ ensures that $\bar{w}_t(M_t) = \max_{M \in \mathcal{M}} \bar{w}_t(M)$, which implies that $\bar{w}_t(M_t) \geq \bar{w}_t(M'_t)$. This is a contradiction, and therefore we have $p_t \neq e$ for this case.

**Case (2):** $(e \in M_* \wedge e \in c_-) \vee (e \notin M_* \wedge e \in c_+)$.

First, we claim that $\tilde{M}_t \neq M_*$. Suppose that $e \in M_*$ and $e \in c_-$. Then, we see that $e \notin \tilde{M}_t$, which implies that $\tilde{M}_t \neq M_*$. On the other hand, suppose that $e \notin M_*$ and $e \in c_+$, then $e \in \tilde{M}_t$, which also implies that $\tilde{M}_t \neq M_*$. Therefore we have $\tilde{M}_t \neq M_*$ in either cases.

Hence, by Lemma 2, there exists an exchange set $b = (b_+, b_-) \in \mathcal{B}$ such that $e \in (b_+ \cup b_-)$, $b_- \subseteq (\tilde{M}_t \backslash M_*)$, $b_+ \subseteq (M_* \backslash \tilde{M}_t)$ and $(\tilde{M}_t \oplus b) \in \mathcal{M}$. Lemma 2 also indicates that $\langle \boldsymbol{w}, \boldsymbol{\chi}_b \rangle \geq \Delta_e > 0$. We will use $\tilde{M}_t \oplus b$ as a gadget for this case. Note that the exchange set $b$ defined here is different from the exchange set $b$ used in Case (1).

Next, we define vectors $\boldsymbol{d} = \boldsymbol{\chi}_{\tilde{M}_t} - \boldsymbol{\chi}_{M_t}$ and $\boldsymbol{d}_1 = \boldsymbol{\chi}_{\tilde{M}_t \oplus b} - \boldsymbol{\chi}_{M_t}$. Notice that Lemma 1 gives that $\boldsymbol{d}_1 = \boldsymbol{d} + \boldsymbol{\chi}_b$.

Then, we apply Lemma 3 by setting $M = M_t$ and $M' = \tilde{M}_t$. This shows that

$$\|\mathbf{rad}_t \circ \boldsymbol{d}\|_\infty \leq \max_{i:(\tilde{M}_t \setminus M_t) \cup (M_t \setminus \tilde{M}_t)} \mathrm{rad}_t(i) = \mathrm{rad}_t(e) < \frac{\Delta_e}{3\,\mathrm{width}(\mathcal{B})}, \tag{43}$$

where the last inequality follows from the assumption that $\mathrm{rad}_t(e) < \frac{\Delta_e}{3\,\mathrm{width}(\mathcal{B})}$.

Now, we bound quantity $\langle \bar{\boldsymbol{w}}_t, \boldsymbol{d}_1 \rangle + \langle \mathbf{rad}_t, |\boldsymbol{d}_1| \rangle - \langle \bar{\boldsymbol{w}}_t, \boldsymbol{d} \rangle - \langle \mathbf{rad}_t, |\boldsymbol{d}| \rangle$ as follows

$$\langle \bar{\boldsymbol{w}}_t, \boldsymbol{d}_1 \rangle + \langle \mathbf{rad}_t, |\boldsymbol{d}_1| \rangle - \langle \bar{\boldsymbol{w}}_t, \boldsymbol{d} \rangle - \langle \mathbf{rad}_t, |\boldsymbol{d}| \rangle = \langle \bar{\boldsymbol{w}}_t, \boldsymbol{\chi}_b \rangle + \langle \mathbf{rad}_t, |\boldsymbol{d}_1| - |\boldsymbol{d}| \rangle$$

$$= \langle \bar{\boldsymbol{w}}_t, \boldsymbol{\chi}_b \rangle + \langle \mathbf{rad}_t, \boldsymbol{d}_1^2 - \boldsymbol{d}^2 \rangle \tag{44}$$

$$= \langle \bar{\boldsymbol{w}}_t, \boldsymbol{\chi}_b \rangle + \langle \mathbf{rad}_t, 2\boldsymbol{d} \circ \boldsymbol{\chi}_b + \boldsymbol{\chi}_b^2 \rangle \tag{45}$$

$$= \langle \bar{\boldsymbol{w}}_t, \boldsymbol{\chi}_b \rangle + \langle \mathbf{rad}_t, \boldsymbol{\chi}_b^2 \rangle + 2 \langle \mathbf{rad}_t \circ \boldsymbol{d}, \boldsymbol{\chi}_b \rangle \tag{46}$$

$$\geq \langle \boldsymbol{w}, \boldsymbol{\chi}_b \rangle + 2 \langle \mathbf{rad}_t \circ \boldsymbol{d}, \boldsymbol{\chi}_b \rangle \tag{47}$$

$$\geq \langle \boldsymbol{w}, \boldsymbol{\chi}_b \rangle - 2 \|\mathbf{rad}_t \circ \boldsymbol{d}\|_\infty \|\boldsymbol{\chi}_b\|_1 \tag{48}$$

$$> \langle \boldsymbol{w}, \boldsymbol{\chi}_b \rangle - \frac{2\Delta_e}{3} \tag{49}$$

$$> 0, \tag{50}$$

where Eq. (44) follows from the fact that $\boldsymbol{d}_1 \in \{-1, 0, 1\}^n$ and $\boldsymbol{d} \in \{-1, 0, 1\}^n$; Eq. (45) holds since $\boldsymbol{d}_1 = \boldsymbol{d} + \boldsymbol{\chi}_b$; Eq. (46) follows from Lemma 4; Eq. (47) follows from the assumption that $\xi_t$ occurs and Lemma 7; Eq. (48) follows from Hölder's inequality; Eq. (49) is due to Eq. (43); and Eq. (50) follows from $\langle \boldsymbol{w}, \boldsymbol{\chi}_b \rangle \geq \Delta_e > 0$.

Therefore, we have proven that

$$\langle \bar{\boldsymbol{w}}_t, \boldsymbol{d} \rangle + \langle \mathbf{rad}_t, |\boldsymbol{d}| \rangle < \langle \bar{\boldsymbol{w}}_t, \boldsymbol{d}_1 \rangle + \langle \mathbf{rad}_t, |\boldsymbol{d}_1| \rangle. \tag{51}$$

However, we have

$$\langle \bar{\boldsymbol{w}}_t, \boldsymbol{d} \rangle + \langle \mathbf{rad}_t, |\boldsymbol{d}| \rangle = \langle \bar{\boldsymbol{w}}_t, \boldsymbol{\chi}_{\tilde{M}_t} - \boldsymbol{\chi}_{M_t} \rangle + \langle \mathbf{rad}_t, |\boldsymbol{\chi}_{\tilde{M}_t} - \boldsymbol{\chi}_{M_t}| \rangle$$

$$= \tilde{w}_t(\tilde{M}_t) - \tilde{w}_t(M_t) \tag{52}$$

$$\geq \tilde{w}_t(\tilde{M}_t \oplus b) - \tilde{w}_t(M_t) \tag{53}$$

$$= \langle \bar{\boldsymbol{w}}_t, \boldsymbol{\chi}_{\tilde{M}_t \oplus b} - \boldsymbol{\chi}_{M_t} \rangle + \langle \mathbf{rad}_t, |\boldsymbol{\chi}_{\tilde{M}_t \oplus b} - \boldsymbol{\chi}_{M_t}| \rangle$$

$$= \langle \bar{\boldsymbol{w}}_t, \boldsymbol{d}_1 \rangle + \langle \mathbf{rad}_t, |\boldsymbol{d}_1| \rangle, \tag{54}$$

where Eq. (52) follows from Lemma 5; and Eq. (53) follows from the fact that $\tilde{w}_t(\tilde{M}_t) = \max_{M \in \mathcal{M}} \tilde{w}_t(M)$. This contradicts to Eq. (51) and therefore $p_t \neq e$. □

### A.4   Proof of Theorem 1

Theorem 1 is now a straightforward corollary of Lemma 9 and Lemma 10. For the reader's convenience, we first restate Theorem 1 in the following.

**Theorem 1.** *Given any* $\delta \in (0, 1)$, *any decision class* $\mathcal{M} \subseteq 2^{[n]}$ *and any expected rewards* $\boldsymbol{w} \in \mathbb{R}^n$. *Assume that the reward distribution* $\varphi_e$ *for each arm* $e \in [n]$ *has mean* $w(e)$ *with an* $R$-*sub-Gaussian tail. Let* $M_* = \arg\max_{M \in \mathcal{M}} w(M)$ *denote the optimal set. Set* $\mathrm{rad}_t(e) = R\sqrt{2\log\left(\frac{4nt^3}{\delta}\right)/T_t(e)}$ *for all* $t > 0$ *and* $e \in [n]$. *Then, with probability at least* $1 - \delta$, *the* CLUCB *algorithm (Algorithm 1) returns the optimal set* $\mathsf{Out} = M_*$ *and*

$$T \leq O\left(R^2\,\mathrm{width}(\mathcal{M})^2 \mathbf{H} \log\left(nR^2\mathbf{H}/\delta\right)\right), \tag{5}$$

*where* $T$ *denotes the number of samples used by Algorithm 1,* $\mathbf{H}$ *is defined in Eq. (2) and* $\mathrm{width}(\mathcal{M})$ *is defined in Eq. (4).*

*Proof.* Lemma 8 indicates that the event $\xi \triangleq \bigcap_{t=1}^\infty \xi_t$ occurs with probability at least $1 - \delta$. In the rest of the proof, we shall assume that this event holds.

By Lemma 9 and the assumption on $\xi$, we see that $\mathsf{Out} = M_*$. Next, we focus on bounding the total number $T$ of samples.

Fix any arm $e \in [n]$. Let $T(e)$ denote the total number of pull of arm $e \in [n]$. Let $t_e$ be the last round which arm $e$ is pulled, which means that $p_{t_e} = e$. It is easy to see that $T_{t_e}(e) = T(e) - 1$. By Lemma 10, we see that $\mathrm{rad}_{t_e}(e) \geq \frac{\Delta_e}{3\,\mathrm{width}(\mathcal{M})}$. Using the definition of $\mathrm{rad}_{t_e}$, we have

$$\frac{\Delta_e}{3\,\mathrm{width}(\mathcal{M})} \leq R\sqrt{\frac{2\log\left(4nt_e^3/\delta\right)}{T(e) - 1}} \leq R\sqrt{\frac{2\log\left(4nT^3/\delta\right)}{T(e) - 1}}. \tag{55}$$

By solving Eq. (55) for $T(e)$, we obtain

$$T(e) \leq \frac{18\,\mathrm{width}(\mathcal{M})^2 R^2}{\Delta_e^2}\log(4nT^3/\delta) + 1. \tag{56}$$

Now we define $\tilde{\mathbf{H}} = \max\left\{\,\mathrm{width}(\mathcal{M})^2 R^2 \mathbf{H}, 1\right\}$. In the rest of the proof, we show that

$$T \leq 499\tilde{\mathbf{H}}\log\left(4n\tilde{\mathbf{H}}/\delta\right) + 2n. \tag{57}$$

Notice that the theorem follows immediately from Eq. (57).

If $n \geq \frac{1}{2}T$, then we see that $T \leq 2n$ and therefore Eq. (57) holds immediately. Next we assume that $n < \frac{1}{2}T$. Since $T > n$, we can write

$$T = C\tilde{\mathbf{H}}\log\left(4n\tilde{\mathbf{H}}/\delta\right) + n, \text{ for some } C > 0. \tag{58}$$

If $C \leq 499$, then it is clear that Eq. (57) holds. Suppose, on the contrary, that $C > 499$. Notice that $T = \sum_{e \in [n]} T(e)$. By summing up Eq. (56) for all $e \in [n]$, we have

$$T \leq n + \sum_{e \in [n]} \frac{18\,\mathrm{width}(\mathcal{M})^2 R^2}{\Delta_e^2}\log(4nT^3/\delta)$$

$$\leq n + 18\tilde{\mathbf{H}}\log(4nT^3/\delta)$$

$$= n + 18\tilde{\mathbf{H}}\log(4n/\delta) + 54\tilde{\mathbf{H}}\log(T)$$

$$\leq n + 18\tilde{\mathbf{H}}\log(4n/\delta) + 54\tilde{\mathbf{H}}\log\left(2C\tilde{\mathbf{H}}\log\left(4n\tilde{\mathbf{H}}/\delta\right)\right) \tag{59}$$

$$= n + 18\tilde{\mathbf{H}}\log(4n/\delta) + 54\tilde{\mathbf{H}}\log(2C) + 54\tilde{\mathbf{H}}\log(\tilde{\mathbf{H}}) + 54\tilde{\mathbf{H}}\log\log\left(4n\tilde{\mathbf{H}}/\delta\right)$$

$$\leq n + 18\tilde{\mathbf{H}}\log(4n\tilde{\mathbf{H}}/\delta) + 54\tilde{\mathbf{H}}\log(2C)\log(4n\tilde{\mathbf{H}}/\delta) + 54\tilde{\mathbf{H}}\log(4n\tilde{\mathbf{H}}/\delta) + 54\tilde{\mathbf{H}}\log(4n\tilde{\mathbf{H}}/\delta) \tag{60}$$

$$= n + (126 + 54\log(2C))\tilde{\mathbf{H}}\log(4n\tilde{\mathbf{H}}/\delta)$$

$$< n + C\tilde{\mathbf{H}}\log(4n\tilde{\mathbf{H}}/\delta) \tag{61}$$

$$= T, \tag{62}$$

where Eq. (59) follows from Eq. (58) and the assumption that $n < \frac{1}{2}T$; Eq. (60) follows from the fact that $\tilde{\mathbf{H}} \geq 1$ and $\delta < 1$; Eq. (61) follows since $126 + 54\log(2C) < C$ for all $C > 499$; and Eq. (62) is due to Eq. (58). Now we see that Eq. (62) is a contradiction. Therefore we obtain that $C \leq 499$ and we have proved Eq. (57).

$\square$

# B   Extensions of CLUCB

CLUCB is a general and flexible learning algorithm for the CPE problem. In this section, we present two extensions to CLUCB that allow it to work in the fixed budget setting and PAC learning setting.

## B.1 Fixed Budget Setting

We can extend the CLUCB algorithm to the fixed budget setting using two simple modifications: (1) requiring CLUCB to terminate after $T$ rounds; and (2) using a different construction of confidence intervals. The first modification ensures that CLUCB uses at most $T$ samples, which meets the requirement of the fixed budget setting. And the second modification bounds the probability that the confidence intervals are valid for all arms in $T$ rounds. The following theorem shows that the probability of error of the modified CLUCB is bounded by $O\left(Tn\exp\left(\frac{-T}{\text{width}(\mathcal{M})^2\mathbf{H}}\right)\right)$.

**Theorem 4.** *Use the same notations as in Theorem 1. Given $T > n$ and parameter $\alpha > 0$, set the confidence radius $\text{rad}_t(e) = R\sqrt{\frac{\alpha}{T_t(e)}}$ for all arms $e \in [n]$ and all $t > 0$. Run CLUCB algorithm for at most $T$ rounds. Then, for $0 \le \alpha \le \frac{1}{9}(T-n)\left(R^2\text{width}(\mathcal{M})^2\mathbf{H}\right)^{-1}$, we have*

$$\Pr\left[\text{Out} \ne M_*\right] \le 2Tn\exp\left(-\alpha/2\right). \tag{63}$$

*In particular, the right-hand side of Eq. (63) equals to $O\left(Tn\exp\left(\frac{-T}{\text{width}(\mathcal{M})^2\mathbf{H}}\right)\right)$ when parameter $\alpha = O(T\mathbf{H}^{-1}\text{width}(\mathcal{M})^{-2})$.*

Theorem 4 shows that the modified CLUCB algorithm in the fixed budget setting requires the knowledge of quantity $\mathbf{H}$ in order to achieve the optimal performance. However $\mathbf{H}$ is usually unknown. Therefore, although its probability of error guarantee matches the parameter-free CSAR algorithm up to logarithmic factors, this modified algorithm is considered weaker than CSAR. Nevertheless, Theorem 4 shows that CLUCB can solve CPE in both fixed confidence and fixed budget settings and more importantly this theorem provides additional insights on the behavior CLUCB.

## B.2 PAC Learning

Now we consider a setting where the learner is only required to report an approximately optimal set of arms. More specifically, we consider the notion of $(\epsilon, \delta)$-PAC algorithm. Formally, an algorithm $\mathbb{A}$ is called an $(\epsilon, \delta)$-PAC algorithm if its output $\text{Out} \in \mathcal{M}$ satisfies $\Pr\left[w(M_*) - w(\text{Out}) > \epsilon\right] \le \delta$.

We show that a simple modification on the CLUCB algorithm gives an $(\epsilon, \delta)$-PAC algorithm, with guarantees similar to Theorem 1. In fact, the only modification needed is to change the stopping condition from $\tilde{w}_t(\tilde{M}_t) = \tilde{w}_t(M_t)$ to $\tilde{w}_t(\tilde{M}_t) - \tilde{w}_t(M_t) \le \epsilon$ on line 11 of Algorithm 1. We let CLUCB-PAC denote the modified algorithm. In the following theorem, we show that CLUCB-PAC is indeed an $(\epsilon, \delta)$-PAC algorithm and has a sample complexity similar to CLUCB.

**Theorem 5.** *Use the same notations as in Theorem 1. Fix $\delta \in (0, 1)$ and $\epsilon \ge 0$. Then, with probability at least $1 - \delta$, the output $\text{Out} \in \mathcal{M}$ of CLUCB-PAC satisfies $w(M_*) - w(\text{Out}) \le \epsilon$. In addition, the number of samples $T$ used by the algorithm satisfies*

$$T \le O\left(R^2\sum_{e\in[n]}\min\left\{\frac{\text{width}(\mathcal{M})^2}{\Delta_e^2}, \frac{K^2}{\epsilon^2}\right\}\log\left(\frac{R^2}{\delta}\sum_{e\in[n]}\min\left\{\frac{\text{width}(\mathcal{M})^2}{\Delta_e^2}, \frac{K^2}{\epsilon^2}\right\}\right)\right), \tag{64}$$

*where $K = \max_{M\in\mathcal{M}}|M|$ is the size of the largest member of decision class.*

We see that if $\epsilon = 0$, the sample complexity Eq. (64) of CLUCB-PAC equals to that of CLUCB. Moreover, the sample complexity of CLUCB-PAC decreases when $\epsilon$ increases.

There are several PAC learning algorithms dedicated for the TopK problem in the literature with different guarantees [19, 31, 13]. Zhou et al. [31] proposed an $(\epsilon, \delta)$-PAC algorithm for the TopK problem with a problem-independent sample complexity bound of $O(\frac{K^2n}{\epsilon^2} + \frac{Kn\log(1/\delta)}{\epsilon^2})$.[2] If we ignore logarithmic factors, then the sample complexity bound of CLUCB-PAC for the TopK problem is better than theirs since $\sum_{e\in[n]}\min\{\Delta_e^{-2}, K^2\epsilon^{-2}\} \le nK^2\epsilon^{-2}$. On the other hand, the algorithms

of Kalyanakrishnan et al. [19], Gabillon et al. [13] and Kaufmann and Kalyanakrishnan [20] guarantee to find $K$ arms such that each of them is better than the $K$-th optimal arm within a factor of $\epsilon$ with probability $1 - \delta$. Unless $\epsilon = 0$, their guarantee is different from ours which concerns the optimality of the sum of $K$ arms.

### B.3 Proof of Extension Results

#### B.3.1 Fixed Budget Setting (Theorem 4)

In this part, we analyze the probability of error of the modified CLUCB algorithm in the fixed budget setting and prove Theorem 4. First, we prove a lemma which characterizes the confidence intervals constructed in Theorem 4.

**Lemma 11.** *Fix parameter $\alpha > 0$ and the number of rounds $T > 0$. Assume that the reward distribution $\varphi_e$ is a $R$-sub-Gaussian distribution for all $e \in [n]$. Let the confidence radius $\mathrm{rad}_t(e)$ of arm $e \in [n]$ and round $t > 0$ be $\mathrm{rad}_t(e) = R\sqrt{\frac{\alpha}{T_t(e)}}$. Then, we have*

$$\Pr\left[\bigcap_{t=1}^{T} \xi_t\right] \geq 1 - 2nT \exp\left(-\alpha/2\right),$$

*where $\xi_t$ is the random event defined in Eq. (12).*

*Proof.* For any $t > 0$ and $e \in [n]$, using Hoeffding's inequality, we have

$$\Pr\left[\left|\bar{w}_t(e) - w(e)\right| \geq \mathrm{rad}_t(e)\right] \leq 2\exp(-\alpha/2).$$

By a union bound over all arms $e \in [n]$, we see that $\Pr[\xi_t] \geq 1 - 2n\exp(-\alpha/2)$. The lemma follows immediately by using union bound again over all round $t \in [T]$. ∎

Theorem 4 can be obtained from the key lemmas (Lemma 9 and Lemma 10) and Lemma 11.

*Proof of Theorem 4.* Define random event $\xi = \bigcap_{t=1}^{T} \xi_t$. By Lemma 11, we see that $\Pr[\xi] \geq 1 - 2nT \exp(-\alpha/2)$. In the rest of the proof, we assume that $\xi$ happens.

Let $T^*$ denote the round that the algorithm stops. We claim that the algorithm stops before the budget is exhausted, i.e., $T^* < T$. If the claim is true, then the algorithm stops since it meets the stopping condition on round $T^*$. Hence $\tilde{w}_t(\tilde{M}_{T^*}) = \tilde{w}_t(M_{T^*})$ and $\mathsf{Out} = M_{T^*}$. By assumption on $\xi$ and Lemma 9, we know that $M_{T^*} = M_*$. Therefore the theorem follows immediately from this claim and the bound of $\Pr[\xi]$.

Next, we show that this claim is true. Let $T(e)$ denote the total number of pulls of arm $e \in [n]$. Let $t_e$ be the last round that arm $e$ is pulled. Hence $T_{t_e}(e) = T_e - 1$. By Lemma 10, we see that $\mathrm{rad}_{t_e}(e) \geq \frac{\Delta}{3\,\mathrm{width}(\mathcal{B})}$. Now plugging in the definition of $\mathrm{rad}_{t_e}(e)$, we have

$$\frac{\Delta}{3\,\mathrm{width}(\mathcal{B})} \leq \mathrm{rad}_{t_e}(e)$$

$$= R\sqrt{\frac{\alpha}{T_{t_e}(e)}} = R\sqrt{\frac{\alpha}{T(e) - 1}}.$$

Hence we have

$$T_e \leq \frac{9R^2\,\mathrm{width}(\mathcal{B})^2}{\Delta_e^2} \cdot \alpha + 1. \tag{65}$$

By summing up Eq. (65) for all $e \in [n]$, we have

$$T^* = \sum_{e \in [n]} T_e \leq \alpha \cdot 9R^2\,\mathrm{width}(\mathcal{B})^2 \left(\sum_{e \in [n]} \Delta_e^{-2}\right) + n < T,$$

where we have used the assumption that $\alpha < \frac{1}{9}(T - n) \cdot \left(R^2\,\mathrm{width}(\mathcal{B})^2 \left(\sum_{e \in [n]} \Delta_e^{-2}\right)\right)^{-1}$. ∎

### B.3.2 PAC Learning (Theorem 5)

First, we prove a $(\epsilon, \delta)$-PAC counterpart of Lemma 9.

**Lemma 12.** *If CLUCB-PAC stops on round $t$ and suppose that event $\xi_t$ occurs. Then, we have $w(M_*) - w(\mathsf{Out}) \leq \epsilon$.*

*Proof.* By definition, we know that $\mathsf{Out} = M_t$. Notice that the stopping condition of CLUCB-PAC ensures that $\tilde{w}_t(\tilde{M}_t) - \tilde{w}_t(M_t) \leq \epsilon$. Therefore, we have

$$\epsilon \geq \tilde{w}_t(\tilde{M}_t) - \tilde{w}_t(M_t)$$
$$\geq \tilde{w}_t(M_*) - \tilde{w}_t(M_t) \tag{66}$$
$$= \langle \bar{\boldsymbol{w}}_t, \boldsymbol{\chi}_{M_*} - \boldsymbol{\chi}_{M_t} \rangle + \langle \mathbf{rad}_t, |\boldsymbol{\chi}_{M_*} - \boldsymbol{\chi}_{M_t}| \rangle \tag{67}$$
$$\geq \langle \boldsymbol{w}, \boldsymbol{\chi}_{M_*} - \boldsymbol{\chi}_{M_t} \rangle \tag{68}$$
$$= w(M_*) - w(M_t),$$

where Eq. (66) follows from the definition that $\tilde{w}_t(\tilde{M}_t) = \max_{M \in \mathcal{M}} \tilde{w}_t(M)$; Eq. (67) follows from Lemma 5; Eq. (68) follows from the assumption that $\xi_t$ occurs and Lemma 7. $\qquad \square$

The next lemma generalizes Lemma 10. It shows that on event $\xi_t$ each arm $e \in [n]$ will not be played on round $t$ if $\mathrm{rad}_t(e) < \max\left\{ \frac{\Delta_e}{3 \operatorname{width}(\mathcal{M})}, \frac{\epsilon}{2K} \right\}$.

**Lemma 13.** *Let $K = \max_{M \in \mathcal{M}} |M|$. For any arm $e \in [n]$ and any round $t > n$ after initialization, if $\mathrm{rad}_t(e) < \max\left\{ \frac{\Delta_e}{3 \operatorname{width}(\mathcal{M})}, \frac{\epsilon}{2K} \right\}$ and random event $\xi_t$ occurs, then arm $e$ will not be played on round $t$, i.e., $p_t \neq e$.*

*Proof.* If $\mathrm{rad}_t(e) < \frac{\Delta_e}{3 \operatorname{width}(\mathcal{M})}$, then we can apply Lemma 10 which immediately gives that $p_t \neq e$. Hence, we only need to prove the case that $\frac{\Delta_e}{3 \operatorname{width}(\mathcal{M})} \leq \mathrm{rad}_t(e) < \frac{\epsilon}{2K}$.

Now suppose that $p_t = e$. By the choice of $p_t$, we know that for each $i \in (M_t \backslash \tilde{M}_t) \cup (\tilde{M}_t \backslash M_t)$, we have $\mathrm{rad}_t(i) \leq \mathrm{rad}_t(e) < \frac{\epsilon}{2K}$. By summing up this inequality for all $i \in (M_t \backslash \tilde{M}_t) \cup (\tilde{M}_t \backslash M_t)$, we have

$$\epsilon > \sum_{i \in (M_t \backslash \tilde{M}_t) \cup (\tilde{M}_t \backslash M_t)} \mathrm{rad}_t(i) \tag{69}$$
$$= \langle \mathbf{rad}_t, |\boldsymbol{\chi}_{M_t} - \boldsymbol{\chi}_{\tilde{M}_t}| \rangle, \tag{70}$$

where Eq. (69) follows from the fact that $|(M_t \backslash \tilde{M}_t) \cup (\tilde{M}_t \backslash M_t)| \leq |M_t| + |\tilde{M}_t| \leq 2K$; and Eq. (70) uses the fact that $\boldsymbol{\chi}_{(M_t \backslash \tilde{M}_t) \cup (\tilde{M}_t \backslash M_t)} = |\boldsymbol{\chi}_{M_t} - \boldsymbol{\chi}_{\tilde{M}_t}|$.

Then, we have

$$\tilde{w}_t(\tilde{M}_t) - \tilde{w}_t(M_t) = \langle \bar{\boldsymbol{w}}_t, \boldsymbol{\chi}_{\tilde{M}_t} - \boldsymbol{\chi}_{M_t} \rangle + \langle \mathbf{rad}_t, |\boldsymbol{\chi}_{\tilde{M}_t} - \boldsymbol{\chi}_{M_t}| \rangle \tag{71}$$
$$\leq \langle \bar{\boldsymbol{w}}_t, \boldsymbol{\chi}_{\tilde{M}_t} - \boldsymbol{\chi}_{M_t} \rangle + \epsilon \tag{72}$$
$$= \bar{w}_t(\tilde{M}_t) - \bar{w}_t(M_t) + \epsilon$$
$$\leq \epsilon, \tag{73}$$

where Eq. (71) follows from Lemma 5; Eq. (72) uses Eq. (70); and Eq. (73) follows from $\bar{w}_t(M_t) \geq \bar{w}_t(\tilde{M}_t)$.

Therefore, we see that $\tilde{w}_t(\tilde{M}_t) - \tilde{w}_t(M_t) \leq \epsilon$. By the stopping condition of CLUCB-PAC, the algorithm must terminate on round $t$, before playing any arms. This contradicts to the assumption that $p_t = e$. $\qquad \square$

Using Lemma 13 and Lemma 12, we are ready to prove Theorem 5.

*Proof of Theorem 5.* Similar to the proof of Theorem 1, we appeal to Lemma 8, which shows that the event $\xi \triangleq \bigcap_{t=1}^{\infty} \xi_t$ occurs with probability at least $1 - \delta$. We shall assume that $\xi$ occurs in the rest of the proof.

By the assumption of $\xi$ and Lemma 12, we know that $w(M_*) - w(\mathsf{Out}) \leq \epsilon$. Therefore, we only remain to bound the number of samples $T$.

Consider an arbitrary arm $e \in [n]$. Let $T(e)$ denote the total number of pulls of arm $e \in [n]$. Let $t_e$ be the last round in which arm $e$ is pulled, i.e., $p_{t_e} = e$. Hence $T_{t_e}(e) = T(e) - 1$. By Lemma 13, we see that $\mathrm{rad}_{t_e}(e) \geq \max\{\frac{\Delta_e}{3\,\mathrm{width}(\mathcal{B})}, \frac{\epsilon}{2K}\}$. Then, by the construction of $\mathrm{rad}_{t_e}(e)$, we have

$$\max\left\{\frac{\Delta_e}{3\,\mathrm{width}(\mathcal{B})}, \frac{\epsilon}{2K}\right\} \leq R\sqrt{\frac{2\log\left(4nt_e^3/\delta\right)}{T(e) - 1}} \leq R\sqrt{\frac{2\log\left(4nT^3/\delta\right)}{T(e) - 1}}. \tag{74}$$

Solving Eq. (74) for $T(e)$, we obtain

$$T(e) \leq R^2 \min\left\{\frac{18\,\mathrm{width}(\mathcal{B})^2}{\Delta_e^2}, \frac{16K^2}{\epsilon^2}\right\} \log(4nT^3/\delta) + 1. \tag{75}$$

Notice that $T = \sum_{i \in [n]} T(e)$. Hence the theorem follows by summing up Eq. (75) for all $e \in [n]$ and solving for $T$. $\qquad\square$

## C  Proof of Lower Bound (Theorem 2)

In this section, we prove the problem-dependent lower bound of the general CPE problem (Theorem 2). In addition, we provide evidence on the conjecture that the sample complexity should hinge on the size of exchange sets (Theorem 6), which is relevant for decision classes with non-constant widths.

**Notations.** In this section, we will use the notion of "next-to-optimal set" defined as follows. Fix a decision class $\mathcal{M} \subseteq 2^{[n]}$ and an expected reward vector $\boldsymbol{w} \in \mathbb{R}^n$. Let $M_* = \arg\max_{M \in \mathcal{M}} w(M)$ denote the optimal set. Then, for any $e \in [n]$, we define the next-to-optimal set associated with $e$ as follows

$$M_e = \begin{cases} \arg\max_{M \in \mathcal{M}: e \in M} w(M) & \text{if } e \notin M_*, \\ \arg\max_{M \in \mathcal{M}: e \notin M} w(M) & \text{if } e \in M_*. \end{cases} \tag{76}$$

We note that, by definition of $\Delta_e$ in Eq. (1), we have $w(M_*) - w(M_e) = \Delta_e$.

### C.1  Proof of Theorem 2

For reader's convenience, we restate Theorem 2 in the following.

**Theorem 2.** *Fix any decision class $\mathcal{M} \subseteq 2^{[n]}$ and any vector $\boldsymbol{w} \in \mathbb{R}^n$. Suppose that, for each arm $e \in [n]$, the reward distribution $\varphi_e$ is given by $\varphi_e = \mathcal{N}(w(e), 1)$, where we let $\mathcal{N}(\mu, \sigma^2)$ denote Gaussian distribution with mean $\mu$ and variance $\sigma^2$. Then, for any $\delta \in (0, e^{-16}/4)$ and any $\delta$-correct algorithm $\mathbb{A}$, we have*

$$\mathbb{E}[T] \geq \frac{1}{16}\mathbf{H}\log\left(\frac{1}{4\delta}\right), \tag{6}$$

*where $T$ denote the number of total samples used by algorithm $\mathbb{A}$ and $\mathbf{H}$ is defined in Eq. (2).*

Before stating our proof, we first introduce two technical lemmas. The first lemma is the well-known Kolmogrov's inequality.

**Lemma 14.** (Kolmogrov's inequality [29, Corollary 7.66]) *Let $Z_1, \ldots, Z_n$ be independent zero-mean random variables with $\mathrm{Var}[Z_k] \leq +\infty$ for all $k \in [n]$. Then, for any $\lambda > 0$,*

$$\Pr\left[\max_{1 \leq k \leq n} |S_k| \geq \lambda\right] \leq \frac{1}{\lambda^2} \sum_{i=1}^{n} \mathrm{Var}[Z_k],$$

*where $S_k = X_1 + \ldots + X_k$.*

The second technical lemma shows that the joint likelihood of Gaussian distributions on a sequence of variables does not change much when the mean of the distribution shifts by a sufficiently small value.

**Lemma 15.** *Fix some $d \in \mathbb{R}$ and $\theta \in (0, 1)$. Define $t = \frac{1}{4d^2} \log(1/\theta)$. Given any integer $T \leq 4t$ and any sequence $s_1, \ldots, s_T$. Let $X_1, \ldots, X_T$ be $T$ real numbers which satisfy the following*

$$\left| \sum_{i=1}^{T} X_i - \sum_{i=1}^{T} s_i \right| \leq \sqrt{t \log(1/\theta)}. \tag{77}$$

*Then, we have*

$$\prod_{i=1}^{T} \frac{\mathcal{N}(X_i | s_i + d, 1)}{\mathcal{N}(X_i | s_i, 1)} \geq \theta,$$

*where we let $\mathcal{N}(x | \mu, \sigma^2) = \frac{1}{\sigma\sqrt{2\pi}} \exp\left(-\frac{(x-\mu)^2}{2\sigma^2}\right)$ denote the probability density function of normal distribution with mean $\mu$ and variance $\sigma^2$.*

*Proof.* We define $v_i = X_i - s_i$ for all $i \in [T]$. Then, we have

$$\prod_{i=1}^{T} \frac{\mathcal{N}(X_i | s_i + d, 1)}{\mathcal{N}(X_i | s_i, 1)} = \prod_{i=1}^{T} \exp\left(\frac{-(X_i - s_i - d)^2 + (X_i - s_i)^2}{2}\right)$$

$$= \prod_{i=1}^{T} \exp\left(-v_i d - \frac{1}{2}d^2\right)$$

$$= \exp\left(-\sum_{i=1}^{T} v_i d\right) \exp\left(-\frac{Td^2}{2}\right). \tag{78}$$

We now bound each term on the right-hand side of Eq. (78) as follows

$$\exp\left(-\sum_{i=1}^{T} v_i d\right) \geq \exp\left(-\left|\sum_{i=1}^{T} v_i\right| \cdot |d|\right)$$

$$\geq \exp\left(-\sqrt{t \log(1/\theta)} d\right) \tag{79}$$

$$= \exp\left(-\frac{1}{2} \log(1/\theta)\right) = \theta^{1/2}, \tag{80}$$

where Eq. (79) follows from Eq. (77); and Eq. (80) follows from the fact $t \leq \frac{1}{4d^2} \log(1/\theta)$. Next we have

$$\exp\left(-\frac{Td^2}{2}\right) \geq \exp\left(-2td^2\right) \tag{81}$$

$$= \exp\left(-\frac{1}{2} \log(1/\theta)\right) = \theta^{1/2}, \tag{82}$$

where Eq. (81) follows from $T \leq 4t$ and Eq. (82) follows from the definition of $t$. The lemma follows immediate by combining Eq. (78), Eq. (80) and Eq. (82). □

*Proof of Theorem 2.* Fix $\delta > 0$, $\boldsymbol{w} = \left(w(1), \ldots, w(n)\right)^T$ and a $\delta$-correct algorithm $\mathbb{A}$. For each $e \in [n]$, assume that the reward distribution is given by $\varphi_e = \mathcal{N}(w(e), 1)$. For any $e \in [n]$, let $T_e$ denote the number of trials of arm $e$ used by algorithm $\mathbb{A}$. In the rest of the proof, we will show that for any $e \in [n]$, the number of trials of arm $e$ is lower-bounded by

$$\mathbb{E}[T_e] \geq \frac{1}{16\Delta_e^2} \log(1/4\delta). \tag{83}$$

Notice that the theorem follows immediately by summing up Eq. (83) for all $e \in [n]$.

Now fix an arm $e \in [n]$. We define $\theta = 4\delta$ and $t_e^* = \frac{1}{16\Delta_e^2} \log(1/\theta)$. We prove Eq. (83) by contradiction. Therefore we assume the opposite that $\mathbb{E}[T_e] < t_e^*$ in the rest of the proof.

**Step (1): An alternative hypothesis.** We consider two hypothesis $H_0$ and $H_1$. Under hypothesis $H_0$, all reward distributions are same with our assumption in the theorem as follows

$$H_0 : \varphi_l = \mathcal{N}(w(l), 1) \quad \text{for all } l \in [n].$$

On the other hand, under hypothesis $H_1$, we change the means of reward distributions such that

$$H_1 : \varphi_e = \begin{cases} \mathcal{N}(w(e) - 2\Delta_e, 1) & \text{if } e \in M_* \\ \mathcal{N}(w(e) + 2\Delta_e, 1) & \text{if } e \notin M_* \end{cases} \quad \text{and } \varphi_l = \mathcal{N}(w(l), 1) \quad \text{for all } l \neq e.$$

For $l \in \{0, 1\}$, we use $\mathbb{E}_l$ and $\mathrm{Pr}_l$ to denote the expectation and probability, respectively, under the hypothesis $H_l$.

Now we claim that $M_*$ is no longer the optimal set under hypothesis $H_1$. Let $M_e$ denote the next-to-optimal set defined Eq. (76). By definition of $\Delta_e$ in Eq. (1), we know that $w(M_*) - w(M_e) = \Delta_e$. Let $\boldsymbol{w}_0$ and $\boldsymbol{w}_1$ be expected reward vectors under $H_0$ and $H_1$ respectively. We have

$$w_1(M_*) - w_1(M_e) = w(M_*) - w(M_e) - 2\Delta_e$$
$$= -\Delta_e < 0.$$

This means that under $H_1$, the set $M_*$ is not the optimal set.

**Step (2): Three random events.** Let $X_1, \ldots, X_{T_e}$ denote the sequence of reward outcomes of arm $e$. Now we define three random events $\mathcal{A}, \mathcal{B}$ and $\mathcal{C}$ as follows

$$\mathcal{A} = \{T_e \leq 4t_e^*\}, \ \mathcal{B} = \{\mathsf{Out} = M_*\} \text{ and } \mathcal{C} = \left\{ \max_{1 \leq t \leq 4t_e^*} \left| \sum_{i=1}^t X_t - t \cdot w(e) \right| < \sqrt{t_e^* \log(1/\theta)} \right\},$$

where $\mathsf{Out}$ is the output of algorithm $\mathbb{A}$.

Now we bound the probability of these events under hypothesis $H_0$. First, we show that $\mathrm{Pr}_0[\mathcal{A}] \geq 3/4$. This can be proven by Markov's inequality as follows.

$$\mathrm{Pr}_0[T_e > 4t_e^*] \leq \frac{\mathbb{E}_0[T_e]}{4t_e^*} \leq \frac{t_e^*}{4t_e^*} = \frac{1}{4}.$$

We now show that $\mathrm{Pr}_0[\mathcal{C}] \geq 3/4$. Notice that $\{X_t - w(e)\}_{t=1,\ldots,}$ is a sequence of zero-mean independent random variables under $H_0$. Define $K_t = \sum_{i=1}^t X_t$. Then, by Kolmogorov's inequality (Lemma 14), we have

$$\mathrm{Pr}_0 \left[ \max_{1 \leq t \leq 4t_e^*} |K_t - t \cdot w(e)| \geq \sqrt{t_e^* \log(1/\theta)} \right] \leq \frac{\mathbb{E}_0[(K_{4t_e^*} - 4w(e)t_e^*)^2]}{t_e^* \log(1/\theta)}$$
$$\overset{(a)}{=} \frac{4t_e^*}{t_e^* \log(1/\theta)} \overset{(b)}{<} \frac{1}{4},$$

where (a) follows from the fact that the variance of $\varphi_e$ equals to 1 and therefore $\mathbb{E}_0[(K_{4t_e^*} - 4w(e)t_e^*)^2] = 4t_e^*$; and (b) follows since $\theta < e^{-16}$.

Since the probability of error of algorithm $\mathbb{A}$ is at most $\delta < e^{-16}/4 < 1/4$, we have $\mathrm{Pr}_0[\mathcal{B}] \geq 3/4$. Define random event $\mathcal{S} = \mathcal{A} \cap \mathcal{B} \cap \mathcal{C}$. Then, by union bound, we have $\mathrm{Pr}_0[\mathcal{S}] \geq 1/4$.

**Step (3): The loss of likelihood.** Now, we claim that, under the assumption that $\mathbb{E}_0[T_e] < t_e^*$, one has $\mathrm{Pr}_1[\mathcal{B}] \geq \delta$. Let $W$ be the history of the sampling process until the algorithm stops (including the sequence of arms chosen at each time and the sequence of observed outcomes). Define the likelihood function $L_l$ as

$$L_l(w) = p_l(W = w),$$

where $p_l$ is the probability density function under hypothesis $H_l$.

Now assume that the event $\mathcal{S}$ occurred. We will bound the likelihood ratio $L_1(W)/L_0(W)$ under this assumption. Since $H_1$ and $H_0$ only differs on the reward distribution of arm $e$, we have

$$\frac{L_1(W)}{L_0(W)} = \prod_{i=1}^{T_e} \frac{\mathcal{N}(X_i | w_1(e), 1)}{\mathcal{N}(X_i | w_0(e), 1)}. \tag{84}$$

By definition of $H_1$ and $H_0$, we see that $w_1(e) = w_0(e) \pm 2\Delta_e$ (where the sign depends on whether $e \in M_*$). Therefore, when event $\mathcal{S}$ occurs, it easy to verify that we can apply Lemma 15 (by setting $d = w_1(e) - w_0(e) = \pm 2\Delta_e$, $T = T_e$ and $s_i = w_0(e)$ for all $i$). Hence, by Lemma 15 and Eq. (84), we have

$$\frac{L_1(W)}{L_0(W)} \geq \theta = 4\delta$$

holds if event $\mathcal{S}$ occurs.

Then, define $1_S$ as the indicator variable of event $\mathcal{S}$, i.e., $1_S = 1$ if and only if $\mathcal{S}$ occurs and otherwise $1_S = 0$. Then, we have

$$\frac{L_1(W)}{L_0(W)} 1_S \geq 4\delta 1_S$$

holds regardless the occurrence of event $\mathcal{S}$. Therefore, we can obtain

$$\begin{aligned}
\Pr_1[\mathcal{B}] \geq \Pr_1[\mathcal{S}] &= \mathbb{E}_1[1_S] \\
&= \mathbb{E}_0 \left[ \frac{L_1(W)}{L_0(W)} 1_S \right] \\
&\geq 4\delta \mathbb{E}_0[1_S] \\
&= 4\delta \Pr_0[\mathcal{S}] \geq \delta.
\end{aligned}$$

Now we have proven that, if $\mathbb{E}_0[T_e] < t_e^*$, then $\Pr_1[\mathcal{B}] \geq \delta$. This means that, if $\mathbb{E}_0[T_e] < t_e^*$, algorithm $\mathbb{A}$ will choose $M_*$ as the output with probability at least $\delta$, under hypothesis $H_1$. However, under $H_1$, we have shown that $M_*$ is not the optimal set since $w_1(M_e) > w_1(M_*)$. Therefore, algorithm $\mathbb{A}$ has a probability of error at least $\delta$ under $H_1$. This contradicts to the assumption that algorithm $\mathbb{A}$ is a $\delta$-correct algorithm. Hence, we must have $\mathbb{E}_0[T_e] \geq t_e^* = \frac{1}{16\Delta_e^2} \log(1/4\delta)$.  $\square$

## C.2 Exchange Set Size Dependent Lower Bound

As a supplement to our main lower bound (Theorem 2), we show that, for any arm $e \in [n]$, there exists an exchange set $b = (b_+, b_-)$ which contains $e$ such that a $\delta$-correct algorithm must spend $\tilde{\Omega}\left((|b_+| + |b_-|)^2/\Delta_e^2\right)$ samples on exploring the arms belonging to $b_+ \cup b_-$. Hence, on average, each arm $e \in b_+ \cup b_-$ must be sampled for $\tilde{\Omega}((|b_+| + |b_-|)\Delta_e^{-2})$ times. This is asymptotically stronger than the result of Theorem 2 when the size of corresponding exchange set $|b_+| + |b_-|$ is non-constant. This result is formalized in the following theorem.

**Theorem 6.** *Fix any $\mathcal{M} \subseteq 2^{[n]}$ and any vector $\boldsymbol{w} \in \mathbb{R}^n$. Suppose that, for each arm $e \in [n]$, the reward distribution $\varphi_e$ is given by $\varphi_e = \mathcal{N}(w(e), 1)$, where $\mathcal{N}(\mu, \sigma^2)$ denotes a Gaussian distribution with mean $\mu$ and variance $\sigma^2$. Fix any $\delta \in (0, e^{-16}/4)$ and any $\delta$-correct algorithm $\mathbb{A}$.*

*Then, for any $e \in [n]$, there exists an exchange set $b = (b_+, b_-)$, such that $e \in b_+ \cup b_-$ and*

$$\mathbb{E}\left[ \sum_{i \in b_+ \cup b_-} T_i \right] \geq \frac{(|b_+| + |b_-|)^2}{32\Delta_e^2} \log(1/4\delta),$$

*where $T_i$ is the number of samples of arm $i$.*

The proof is quite similar to that of Theorem 2 except that they use different constructions of alternative hypothesis and consequently this introduces some difference on the details of computations.

*Proof.* Fix $\delta > 0$, $\boldsymbol{w} = \big(w(1), \ldots, w(n)\big)^T$ and a $\delta$-correct algorithm $\mathbb{A}$. For each $i \in [n]$, assume that the reward distribution is given by $\varphi_i = \mathcal{N}(w(i), 1)$. For any $i \in [n]$, let $T_i$ denote the number of trials of arm $i$ used by algorithm $\mathbb{A}$.

**Step (0): Setup.** Fix an arm $e \in [n]$. As the first step, we construct the exchange set $b = (b_+, b_-)$ claimed in the theorem. Let $M_e$ denote the next-to-optimal set as defined in Eq. (76). By definition of $\Delta_e$ in Eq. (1), we know that $w(M_*) - w(M_e) = \Delta_e$. We construct the exchange set $b = (b_+, b_-)$

where $b_+ = M_* \backslash M_e$ and $b_- = M_e \backslash M_*$. It is easy to check that $M_e \oplus b = M_*$ and $\langle \boldsymbol{w}, \boldsymbol{\chi}_b \rangle = \Delta_e > 0$.

We have now constructed the exchange set. We define $T_{b_-} = \sum_{i \in b_-} T_i$ and $T_{b_+} = \sum_{i \in b_+} T_i$. Now we claim that

$$\textbf{(a)} \ \ \mathbb{E}\left[T_{b_-}\right] \geq \frac{|b_-|^2}{16\Delta_e^2} \log(1/4\delta) \quad \text{and} \quad \textbf{(b)} \ \ \mathbb{E}\left[T_{b_+}\right] \geq \frac{|b_+|^2}{16\Delta_e^2} \log(1/4\delta). \tag{85}$$

It is easy to check that theorem follows immediately from claims **(a)** and **(b)**. In the rest of the proof, we focus on claim **(a)**; the claim **(b)** can be proven using an almost identical argument.

Now we define $\theta = 4\delta$ and $t_{b_-}^* = \frac{|b_-|^2}{16\Delta_e^2} \log(1/\theta)$. We prove claim **(a)** by contradiction, that is to assume the opposite that $\mathbb{E}[T_{b_-}] < t_{b_-}^*$.

**Step (1): An alternative hypothesis.** We define two hypotheses $H_0$ and $H_1$. Under hypothesis $H_0$, the reward distribution

$$H_0 : \varphi_l = \mathcal{N}(w(l), 1) \quad \text{for all } l \in [n].$$

Under hypothesis $H_1$, the mean reward of each arm is given by

$$H_1 : \varphi_i = \begin{cases} \mathcal{N}\left(w(i) + \frac{2\Delta_e}{|b_-|}, 1\right) & \text{if } i \in b_-, \\ \mathcal{N}(w(i), 1) & \text{if } i \notin b_-. \end{cases}$$

Similar to the proof of Theorem 2, we let $\boldsymbol{w}_0$ and $\boldsymbol{w}_1$ denote the expected reward vectors under $H_0$ and $H_1$ respectively. One can verify that $w_1(M_*) - w_1(M_e) = -\Delta_e < 0$. This means that under $H_1$, the set $M_*$ is not the optimal set.

**Step (2): Three random events.** First we consider the complete sequence of sampling process by algorithm $\mathbb{A}$. Formally, let $W = \{(\tilde{I}_1, \tilde{X}_1), \ldots, (\tilde{I}_T, \tilde{X}_T)\}$ be the sequence of all trials by algorithm $\mathbb{A}$, where $\tilde{I}_i$ denotes the arm played in $i$-th trial and $\tilde{X}_i$ be the reward outcome of $i$-th trial. Then, consider the subsequence $W_1$ of $W$ which consists of all the trials of arms in $b_-$. Specifically, we write $W = \{(I_1, X_1), \ldots, (I_{T_{b_-}}, X_{T_{b_-}})\}$ such that $W_1$ is a subsequence of $W$ and $I_i \in b_-$ for all $i$.

Now we define three random events $\mathcal{A}$, $\mathcal{B}$ and $\mathcal{C}$ as follows

$$\mathcal{A} = \{T_{b_-} \leq 4t_{b_-}^*\}, \ \mathcal{B} = \{\mathsf{Out} = M_*\} \ \text{and} \ \mathcal{C} = \left\{ \max_{1 \leq t \leq 4t_{b_-}^*} \left| \sum_{i=1}^t X_i - \sum_{i=1}^t w(I_i) \right| < \sqrt{t_{b_-}^* \log(1/\theta)} \right\},$$

where $\mathsf{Out}$ is the output of algorithm $\mathbb{A}$. We now bound the probability of each event. First, by Markov's inequality, we have

$$\mathrm{Pr}_0[T_{b_-} > 4t_{b_-}^*] \leq \frac{\mathbb{E}_0[T_{b_-}]}{4t_{b_-}^*} = \frac{t_{b_-}^*}{4t_{b_-}^*} = \frac{1}{4}.$$

Next, using Kolmogrov's inequality (Lemma 14), we obtain

$$\mathrm{Pr}_0\left[ \max_{1 \leq t \leq 4t_{b_-}^*} \left| \sum_{i=1}^t X_i - \sum_{i=1}^t w(I_i) \right| \geq \sqrt{t_e^* \log(1/\theta)} \right] \leq \frac{\mathbb{E}_0\left[ \left( \sum_{i=1}^{4t_{b_-}^*} X_i - \sum_{i=1}^{4t_{b_-}^*} w(I_i) \right)^2 \right]}{t_e^* \log(1/\theta)}$$

$$\overset{(a)}{=} \frac{4t_{b_-}^*}{t_{b_-}^* \log(1/\theta)} \overset{(b)}{<} \frac{1}{4},$$

where (a) follows from the fact that all reward distributions have unit variance; and (b) follows since $\theta < e^{-16}$.

Since $\mathbb{A}$ is a $\delta$-correct algorithm and $\delta < 1/4$, we have $\mathrm{Pr}_0[\mathcal{B}] \geq 3/4$. Therefore, we have that the random event $\mathcal{S} = \mathcal{A} \cap \mathcal{B} \cap \mathcal{C}$ occurs with probability at least $1/4$ under $H_0$.

**Step (3): The loss of likelihood.** Similar to the proof of Theorem 2, we let $L_l$ denote the likelihood function under hypothesis $H_l$ for $l \in \{0, 1\}$. Since the difference of $H_0$ and $H_1$ only lies in the reward distributions of arms belonging to $b_-$, we have

$$\frac{L_1(W)}{L_0(W)} = \prod_{i=1}^{T_{b_-}} \frac{\mathcal{N}(X_i | w_1(I_i), 1)}{\mathcal{N}(X_i | w_0(I_i), 1)},$$

where $X_i$ and $I_i$ is as defined in Step (2). Assume that $\mathcal{S}$ occurs. Since, for all $i \in [T_{b_-}]$, we have $w_1(I_i) - w_0(I_i) = \frac{2\Delta_e}{|b_-|}$, we can apply Lemma 15 here (by setting $d = \frac{2\Delta_e}{|b_-|}$). Therefore, on event $\mathcal{S}$, we have

$$\frac{L_1(W)}{L_0(W)} \geq \theta.$$

The rest of the proof is identical to Step (3) in the proof of Theorem 2, and one can show that $\Pr_1[\mathcal{B}] \geq \delta$ under the assumption that $\mathbb{E}[T_{b_-}] < t_{b_-}^*$. This means the probability of error of algorithm $\mathbb{A}$ is at least $\delta$. This contradicts to the assumption of $\mathbb{A}$. Therefore we have $\mathbb{E}[T_{b_-}] \geq t_{b_-}^*$ which proves claim **(a)** in Eq. (85). $\qquad\square$

# D   Analysis of CSAR (Theorem 3)

**Notations.** Let $\boldsymbol{w} \in \mathbb{R}^n$ be the vector of the expected rewards of arms. Let $M_* = \arg\max_{M \in \mathcal{M}} w(M)$ be the optimal set. Let $T$ be the budget of samples. We will also use the following additional notations in the rest of this section. Let $M \subseteq [n]$ be a set, we denote $\neg M$ to be the complement of $M$. Let $\Delta_{(1)}, \ldots, \Delta_{(n)}$ be a permutation of $\Delta_1, \ldots, \Delta_n$ such that $\Delta_{(1)} \leq \ldots \ldots \Delta_{(n)}$. Let $A_1, \ldots, A_n$ and $B_1, \ldots, B_n$ be the two sequences of sets which are defined in Algorithm 2. We will also continue to use the notations of incidence vectors of sets and exchange sets, which are defined in Appendix A.

## D.1   Confidence Intervals

First we establish the confidence bounds used for the analysis of CSAR.

**Lemma 16.** *Given a phase $t \in [n]$, we define random event $\tau_t$ as follows*

$$\tau_t = \left\{ \forall i \in [n] \backslash (A_t \cup B_t) \quad \left|\bar{w}_t(i) - w(i)\right| < \frac{\Delta_{(n-t+1)}}{3 \operatorname{width}(\mathcal{M})} \right\}. \tag{86}$$

*Then, we have*

$$\Pr\left[\bigcap_{t=1}^n \tau_t\right] \geq 1 - n^2 \exp\left(-\frac{(T-n)}{18 R^2 \tilde{\log}(n) \operatorname{width}(\mathcal{M})^2 \mathbf{H}_2}\right). \tag{87}$$

*Proof.* Fix some $t \in [n]$ and fix some active arm $i \in [n] \backslash (A_t \cup B_t)$ of phase $t$.

Notice that the arm $i$ has been pulled for $\tilde{T}_t$ times during the first $t$ phases. Therefore, by Hoeffding's inequality (Lemma 6), we have

$$\Pr\left[\left|\bar{w}_t(i) - w(i)\right| \geq \frac{\Delta_{(n-t+1)}}{3 \operatorname{width}(\mathcal{M})}\right] \leq 2 \exp\left(-\frac{\tilde{T}_t \Delta_{(n-t+1)}^2}{18 R^2 \operatorname{width}(\mathcal{M})^2}\right). \tag{88}$$

By plugging the definition of $\tilde{T}_t$, the quantity $\tilde{T}_t \Delta_{(n-t+1)}^2$ on the right-hand side of Eq. (88) can be further bounded by

$$\tilde{T}_t \Delta_{(n-t+1)}^2 \geq \frac{T-n}{\tilde{\log}(n)(n-t+1)} \Delta_{(n-t+1)}^2$$

$$\geq \frac{T-n}{\tilde{\log}(n) \mathbf{H}_2},$$

where the last inequality follows from the definition of $\mathbf{H}_2 = \max_{i \in n} i\Delta_{(i)}^{-2}$. By plugging the last inequality into Eq. (88), we have

$$\Pr\left[\left|\bar{w}_t(i) - w(i)\right| \geq \frac{\Delta_{(n-t+1)}}{3\,\mathrm{width}(\mathcal{M})}\right] \leq 2\exp\left(-\frac{(T-n)}{18R^2\tilde{\log}(n)\,\mathrm{width}(\mathcal{M})^2\mathbf{H}_2}\right). \quad (89)$$

Now using Eq. (89) and a union bound for all $t \in [n]$ and all $i \in [n]\backslash(A_t \cup B_t)$, we have

$$\Pr\left[\bigcap_{t=1}^n \tau_t\right] \geq 1 - 2\sum_{t=1}^n (n-t+1)\exp\left(-\frac{(T-n)}{18R^2\tilde{\log}(n)\,\mathrm{width}(\mathcal{M})^2\mathbf{H}_2}\right)$$

$$\geq 1 - n^2\exp\left(-\frac{(T-n)}{18R^2\tilde{\log}(n)\,\mathrm{width}(\mathcal{M})^2\mathbf{H}_2}\right).$$

$\square$

Readers may notice that the right-hand side of Eq. (87) equals to the probability of error of CSAR claimed in Theorem 3. Indeed, we will show that the CSAR algorithm will not make any mistakes if the random event $\bigcap_{t=1}^n \tau_t$ occurs.

The following lemma builds the confidence bound of inner products.

**Lemma 17.** *Fix a phase $t \in [n]$, suppose that random event $\tau_t$ occurs. For any vector $\boldsymbol{a} \in \mathbb{R}^n$, suppose that $\mathrm{supp}(\boldsymbol{a}) \cap (A_t \cup B_t) = \emptyset$, where $\mathrm{supp}(\boldsymbol{a}) \triangleq \{i \mid a(i) \neq 0\}$ is the support of vector $\boldsymbol{a}$. Then, we have*

$$\left|\langle\bar{\boldsymbol{w}}_t, \boldsymbol{a}\rangle - \langle\boldsymbol{w}, \boldsymbol{a}\rangle\right| < \frac{\Delta_{(n-t+1)}}{3\,\mathrm{width}(\mathcal{M})}\,\|\boldsymbol{a}\|_1.$$

*Proof.* Suppose that $\tau_t$ occurs. Then, similar to the proof of Lemma 7, we have

$$\left|\langle\bar{\boldsymbol{w}}_t, \boldsymbol{a}\rangle - \langle\boldsymbol{w}, \boldsymbol{a}\rangle\right| = \left|\langle\bar{\boldsymbol{w}}_t - \boldsymbol{w}, \boldsymbol{a}\rangle\right|$$

$$= \left|\sum_{i=1}^n \left(\bar{w}_t(i) - w(i)\right)a(i)\right|$$

$$\leq \left|\sum_{i \in [n]\backslash(A_t \cup B_t)} \left(\bar{w}_t(i) - w(i)\right)a(i)\right| \quad (90)$$

$$\leq \sum_{i \in [n]\backslash(A_t \cup B_t)} \left|\left(\bar{w}_t(i) - w(i)\right)a(i)\right|$$

$$\leq \sum_{i \in [n]\backslash(A_t \cup B_t)} \left|\bar{w}_t(i) - w(i)\right|\left|a(i)\right|$$

$$< \frac{\Delta_{(n-t+1)}}{3\,\mathrm{width}(\mathcal{M})} \sum_{i \in [n]\backslash(A_t \cup B_t)} \left|a(i)\right| \quad (91)$$

$$= \frac{\Delta_{(n-t+1)}}{3\,\mathrm{width}(\mathcal{M})}\,\|\boldsymbol{a}\|_1,$$

where Eq. (90) follows from the assumption that $\boldsymbol{a}$ is supported on $[n]\backslash(A_t \cup B_t)$; Eq. (91) follows from the definition of $\tau_t$ (Eq. (86)). $\square$

## D.2 Main Lemmas

We begin with a technical lemma which characterizes several useful properties of $A_t$ and $B_t$.

**Lemma 18.** *Fix a phase $t \in [n]$. Suppose that $A_t \subseteq M_*$ and $B_t \cap M_* = \emptyset$. Let $M$ be a set such that $A_t \subseteq M$ and $B_t \cap M = \emptyset$. Let $a$ and $b$ be two sets satisfying that $a \subseteq M\backslash M_*$, $b \subseteq M_*\backslash M$ and $a \cap b = \emptyset$. Then, we have*

$$A_t \subseteq (M\backslash a \cup b) \quad \text{and} \quad B_t \cap (M\backslash a \cup b) = \emptyset \quad \text{and} \quad (a \cup b) \cap (A_t \cup B_t) = \emptyset.$$

*Proof.* We prove the first part as follows

$$A_t \cap (M \backslash a \cup b) = (A_t \cap (M \backslash a)) \cup (A_t \cap b)$$
$$= A_t \cap (M \backslash a) \tag{92}$$
$$= (A_t \cap M) \backslash a$$
$$= A_t \backslash a \tag{93}$$
$$= A_t, \tag{94}$$

where Eq. (92) holds since we have $A_t \cap b \subseteq A_t \cap (M_* \backslash M) \subseteq M \cap (M_* \backslash M) = \emptyset$; Eq. (93) follows from $A_t \subseteq M$; and Eq. (94) follows from $a \subseteq M \backslash M_*$ and $A_t \subseteq M_*$ which imply that $a \cap A_t = \emptyset$. Notice that Eq. (94) is equivalent to $A_t \subseteq (M \backslash a \cup b)$.

Then, we proceed to prove the second part in the following

$$B_t \cap (M \backslash a \cup b) = (B_t \cap (M \backslash a)) \cup (B_t \cap b)$$
$$= B_t \cap (M \backslash a) \tag{95}$$
$$= (B_t \cap M) \backslash a$$
$$= \emptyset \backslash a = \emptyset, \tag{96}$$

where Eq. (95) follows from the fact that $B_t \cap b \subseteq B_t \cap (M_* \backslash M) \subseteq \neg M_* \cap (M_* \backslash M) = \emptyset$; and Eq. (96) follows from the fact that $B_t \cap M = \emptyset$.

Last, we prove the third part. By combining the assumptions that $A_t \subseteq M_*$ and $A_t \subseteq M$, we see that $A_t \subseteq M \cap M_*$. Also note that $a \subseteq M \backslash M_*$ and $b \subseteq M_* \backslash M$, we have

$$(a \cap A_t) \cup (b \cap A_t) \subseteq ((M \backslash M_*) \cap (M \cap M_*)) \cup ((M_* \backslash M) \cap (M \cap M_*)) = \emptyset. \tag{97}$$

Similarly, we have $B_t \subseteq \neg M \cap \neg M_*$. Hence, we derive

$$(a \cap B_t) \cup (b \cap B_t) \subseteq ((M \backslash M_*) \cap (\neg M \cap \neg M_*)) \cup ((M_* \backslash M) \cap (\neg M \cap \neg M_*)) = \emptyset. \tag{98}$$

By combining Eq. (97) and Eq. (98), we obtain

$$(a \cup b) \cap (A_t \cup B_t) = (a \cap A_t) \cup (b \cap A_t) \cup (a \cap B_t) \cup (b \cap B_t) = \emptyset.$$

$\square$

The next lemma provides an important insight on the correctness of CSAR. Informally speaking, suppose that the algorithm does not make an error before phase $t$. Then, we show that, suppose arm $e$ has a gap $\Delta_e$ larger than the "reference" gap $\Delta_{(n-t+1)}$ of phase $t$, then arm $e$ must be correctly classified by $M_t$, i.e., $e \in M_t$ if and only if $e \in M_*$.

**Lemma 19.** *Fix any phase $t > 0$. Suppose that event $\tau_t$ occurs. Also assume that $A_t \subseteq M_*$ and $B_t \cap M_* = \emptyset$. Let $e \in [n] \backslash (A_t \cup B_t)$ be an active arm. Suppose that $\Delta_{(t-n+1)} \le \Delta_e$. Then, we have $e \in (M_* \cap M_t) \cup (\neg M_* \cap \neg M_t)$.*

*Proof.* Fix an exchange class $\mathcal{B} \in \arg\min_{\mathcal{B}' \in \text{Exchange}(\mathcal{M})} \text{width}(\mathcal{B}')$. Suppose that $e \notin (M_* \cap M_t) \cup (\neg M_* \cap \neg M_t)$. This is equivalent to the following

$$e \in (M_* \cap \neg M_t) \cup (\neg M_* \cap M_t). \tag{99}$$

Eq. (99) can be further rewritten as

$$e \in (M_* \backslash M_t) \cup (M_t \backslash M_*).$$

From this assumption, it is easy to see that $M_t \ne M_*$. Therefore we can apply Lemma 2. Then we know that there exists $b = (b_+, b_-) \in \mathcal{B}$ such that $e \in b_- \cup b_+$, $b_- \subseteq M_t \backslash M_*$, $b_+ \subseteq M_* \backslash M_t$, $M_t \oplus b \in \mathcal{M}$ and $\langle \boldsymbol{w}, \boldsymbol{\chi}_b \rangle \ge \Delta_e > 0$.

Using Lemma 18, we see that $(M_t \oplus b) \cap B_t = \emptyset$, $A_t \subseteq (M_t \oplus b)$ and $(b_+ \cup b_-) \cap (A_t \cup B_t) = \emptyset$. Now recall the definition $M_t \in \arg\max_{M \in \mathcal{M}, A_t \subseteq M, B_t \cap M = \emptyset} \bar{w}_t(M)$ and also recall that $M_t \oplus b \in \mathcal{M}$. Therefore, we obtain that

$$\bar{w}_t(M_t) \ge \bar{w}_t(M_t \oplus b). \tag{100}$$

On the other hand, we have

$$\bar{w}_t(M_t \oplus b) = \left\langle \bar{\boldsymbol{w}}_t, \boldsymbol{\chi}_{M_t} + \boldsymbol{\chi}_b \right\rangle \tag{101}$$

$$= \left\langle \bar{\boldsymbol{w}}_t, \boldsymbol{\chi}_{M_t} \right\rangle + \left\langle \bar{\boldsymbol{w}}_t, \boldsymbol{\chi}_b \right\rangle$$

$$> \left\langle \bar{\boldsymbol{w}}_t, \boldsymbol{\chi}_{M_t} \right\rangle + \left\langle \boldsymbol{w}, \boldsymbol{\chi}_b \right\rangle - \frac{\Delta_{(n-t+1)}}{3\,\mathrm{width}(\mathcal{M})} \left\| \boldsymbol{\chi}_b \right\|_1 \tag{102}$$

$$\geq \left\langle \bar{\boldsymbol{w}}_t, \boldsymbol{\chi}_{M_t} \right\rangle + \left\langle \boldsymbol{w}, \boldsymbol{\chi}_b \right\rangle - \frac{\Delta_e}{3\,\mathrm{width}(\mathcal{M})} \left\| \boldsymbol{\chi}_b \right\|_1$$

$$\geq \left\langle \bar{\boldsymbol{w}}_t, \boldsymbol{\chi}_{M_t} \right\rangle + \left\langle \boldsymbol{w}, \boldsymbol{\chi}_b \right\rangle - \frac{\Delta_e}{3} \tag{103}$$

$$\geq \left\langle \bar{\boldsymbol{w}}_t, \boldsymbol{\chi}_{M_t} \right\rangle + \frac{2}{3}\Delta_e \tag{104}$$

$$\geq \left\langle \bar{\boldsymbol{w}}_t, \boldsymbol{\chi}_{M_t} \right\rangle = \bar{w}_t(M_t), \tag{105}$$

where Eq. (101) follows from Lemma 1; Eq. (102) follows from Lemma 17 and the fact that $(b_+ \cup b_-) \cap (A_t \cup B_t) = \emptyset$; Eq. (103) holds since $b \in \mathcal{B}$ which implies that $\|\boldsymbol{\chi}_b\|_1 = |b_+| + |b_-| \leq \mathrm{width}(\mathcal{B}) = \mathrm{width}(\mathcal{M})$; and Eq. (104) and Eq. (105) hold since we have shown that $\langle \boldsymbol{w}, \boldsymbol{\chi}_b \rangle \geq \Delta_e \geq 0$.

This means that $\bar{w}_t(M_t \oplus b) > \bar{w}_t(M_t)$. This contradicts to Eq. (100). Therefore we have $e \in (M_* \cap M_t) \cup (\neg M_* \cap \neg M_t)$. □

The next lemma takes a step further. It shows that if $\Delta_e \geq \Delta_{(n-t+1)}$ for some arm $e$, then the empirical gap of arm $e$, $\bar{w}_t(M_t) - \bar{w}_t(\tilde{M}_{t,e})$, is greater than $\frac{2}{3}\Delta_{(n-t+1)}$.

**Lemma 20.** *Fix any phase $t > 0$. Suppose that event $\tau_t$ occurs. Also assume that $A_t \subseteq M_*$ and $B_t \cap M_* = \emptyset$. Let $e \in [n] \backslash (A_t \cup B_t)$ be an active arm such that $\Delta_{(t-n+1)} \leq \Delta_e$. Then, we have*

$$\bar{w}_t(M_t) - \bar{w}_t(\tilde{M}_{t,e}) > \frac{2}{3}\Delta_{(t-n+1)}.$$

*Proof.* By Lemma 19, we see that

$$e \in (M_* \cap M_t) \cup (\neg M_* \cap \neg M_t). \tag{106}$$

We claim that $e \in (\tilde{M}_{t,e} \backslash M_*) \cup (M_* \backslash \tilde{M}_{t,e})$ and therefore $M_* \neq \tilde{M}_{t,e}$. Recall the definition of $\tilde{M}_{t,e}$, which ensures that $e \in \tilde{M}_{t,e}$ if and only if $e \notin M_t$. By Eq. (106), we see that either $e \in (M_* \cap M_t)$ or $e \in (\neg M_* \cap \neg M_t)$. First let us assume that $e \in M_* \cap M_t$. Then, by definition of $\tilde{M}_{t,e}$, we see that $e \notin \tilde{M}_{t,e}$. Therefore $e \in M_* \backslash \tilde{M}_{t,e}$. On the other hand, suppose that $e \in \neg M_* \cap \neg M_t$. Then, we see that $e \in \tilde{M}_{t,e}$. This means that $e \in \tilde{M}_{t,e} \backslash M_*$.

Hence we can apply Lemma 2. Then we obtain that there exists $b = (b_+, b_-) \in \mathcal{B}$ such that $e \in b_+ \cup b_-$, $b_+ \subseteq M_* \backslash \tilde{M}_{t,e}$, $b_- \subseteq \tilde{M}_{t,e} \backslash M_*$, $\tilde{M}_{t,e} \oplus b \in \mathcal{M}$ and $\langle \boldsymbol{w}, \boldsymbol{\chi}_b \rangle \geq \Delta_e$.

Define $M'_{t,e} \triangleq \tilde{M}_{t,e} \oplus b$. Using Lemma 18, we have $A_t \subseteq M'_{t,e}$, $B_t \cap M'_{t,e} = \emptyset$ and $(b_+ \cup b_-) \cap (A_t \cup B_t) = \emptyset$. Since $M'_{t,e} \in \mathcal{M}$ and by definition $\bar{w}_t(M_t) = \max_{M \in \mathcal{M}, A_t \subseteq M, B_t \cap M = \emptyset} \bar{w}_t(M)$, we have

$$\bar{w}_t(M_t) \geq \bar{w}_t(M'_{t,e}). \tag{107}$$

Hence, we have

$$\bar{w}_t(M_t) - \bar{w}_t(\tilde{M}_{t,e}) \geq \bar{w}_t(M'_{t,e}) - \bar{w}_t(\tilde{M}_{t,e})$$

$$= \bar{w}_t(\tilde{M}_{t,e} \oplus b) - \bar{w}_t(\tilde{M}_{t,e})$$

$$= \left\langle \bar{\boldsymbol{w}}_t, \boldsymbol{\chi}_{\tilde{M}_{t,e}} + \boldsymbol{\chi}_b \right\rangle - \left\langle \bar{\boldsymbol{w}}_t, \boldsymbol{\chi}_{\tilde{M}_{t,e}} \right\rangle \tag{108}$$

$$= \left\langle \bar{\boldsymbol{w}}_t, \boldsymbol{\chi}_b \right\rangle$$

$$> \left\langle \boldsymbol{w}, \boldsymbol{\chi}_b \right\rangle - \frac{\Delta_{(n-t+1)}}{3\,\mathrm{width}(\mathcal{B})} \left\| \boldsymbol{\chi}_b \right\|_1 \tag{109}$$

$$\geq \langle \boldsymbol{w}, \boldsymbol{\chi}_b \rangle - \frac{\Delta_e}{3\,\mathrm{width}(\mathcal{B})} \|\boldsymbol{\chi}_b\|_1 \tag{110}$$

$$\geq \langle \boldsymbol{w}, \boldsymbol{\chi}_b \rangle - \frac{\Delta_e}{3} \tag{111}$$

$$\geq \frac{2}{3}\Delta_e \geq \frac{2}{3}\Delta_{(n-t+1)}, \tag{112}$$

where Eq. (108) follows from Lemma 1; Eq. (109) follows from Lemma 17, the assumption on event $\tau_t$ and the fact $(b_+ \cup b_-) \cap (A_t \cup B_t) = \emptyset$; Eq. (110) follows from the assumption that $\Delta_e \geq \Delta_{(n-t+1)}$; Eq. (111) holds since $b \in \mathcal{B}$ and therefore $\|\boldsymbol{\chi}_b\|_1 \leq \mathrm{width}(\mathcal{M})$; and Eq. (112) follows from the fact that $\langle \boldsymbol{w}, \boldsymbol{\chi}_b \rangle \geq \Delta_e$. $\qquad \square$

The next lemma shows that, during phase $t$, if $\Delta_e \leq \Delta_{(n-t+1)}$ for some arm $e$, then the empirical gap of arm $e$ is smaller than $\frac{1}{3}\Delta_{(n-t+1)}$.

**Lemma 21.** *Fix any phase $t > 0$. Suppose that event $\tau_t$ occurs. Also assume that $A_t \subseteq M_*$ and $B_t \cap M_* = \emptyset$. Suppose an active arm $e \in [n]\backslash(A_t \cup B_t)$ satisfies that $e \in (M_* \cap \neg M_t) \cup (\neg M_* \cap M_t)$. Then, we have*

$$\bar{w}_t(M_t) - \bar{w}_t(\tilde{M}_{t,e}) \leq \frac{1}{3}\Delta_{(n-t+1)}.$$

*Proof.* Fix an exchange class $\mathcal{B} \in \arg\min_{\mathcal{B}' \in \mathrm{Exchange}(\mathcal{M})} \mathrm{width}(\mathcal{B}')$.

The assumption that $e \in (M_* \cap \neg M_t) \cup (\neg M_* \cap M_t)$ can be rewritten as $e \in (M_*\backslash M_t) \cup (M_t\backslash M_*)$. This shows that $M_t \neq M_*$, hence Lemma 2 applies here. Therefore we know that there exists $b = (b_+, b_-) \in \mathcal{B}$ such that $e \in (b_+ \cup b_-)$, $b_+ \subseteq M_*\backslash M_t$, $b_- \subseteq M_t\backslash M_*$, $M_t \oplus b \in \mathcal{M}$ and $\langle \boldsymbol{w}, \boldsymbol{\chi}_b \rangle \geq \Delta_e > 0$.

Define $M'_{t,e} \triangleq M_t \oplus b$. We claim that

$$\bar{w}_t(\tilde{M}_{t,e}) \geq \bar{w}_t(M'_{t,e}). \tag{113}$$

From the definition of $\tilde{M}_{t,e}$ in Algorithm 2, we only need to show that **(a):** $e \in (M'_{t,e}\backslash M_t) \cup (M_t\backslash M'_{t,e})$ and **(b):** $A_t \subseteq M'_{t,e}$ and $B_t \cap M'_{t,e} = \emptyset$. First we prove **(a)**. Notice that $b_+ \cap b_- = \emptyset$ and $b_- \subseteq M_t$. Hence we see that $M'_{t,e}\backslash M_t = (M_t\backslash b_- \cup b_+)\backslash M_t = b_+$ and $M_t\backslash M'_{t,e} = M_t\backslash(M_t\backslash b_- \cup b_+) = b_-$. In addition, we have that $e \in (b_- \cup b_+) = (M'_{t,e}\backslash M_t) \cup (M_t\backslash M'_{t,e})$, therefore we see that **(a)** holds. Next, we notice that **(b)** follows directly from Lemma 18 by setting $M = M_t$. Hence we have shown that Eq. (113) holds.

Hence, we have

$$\begin{aligned} \bar{w}_t(M_t) - \bar{w}_t(\tilde{M}_{t,e}) &\leq \bar{w}_t(M_t) - \bar{w}_t(M'_{t,e}) \\ &= \langle \bar{\boldsymbol{w}}_t, \boldsymbol{\chi}_{M_t} \rangle - \langle \bar{\boldsymbol{w}}_t, \boldsymbol{\chi}_{M_t} + \boldsymbol{\chi}_b \rangle \\ &= -\langle \bar{\boldsymbol{w}}_t, \boldsymbol{\chi}_b \rangle \end{aligned} \tag{114}$$

$$\leq -\langle \boldsymbol{w}, \boldsymbol{\chi}_b \rangle + \frac{\Delta_{(n-t+1)}}{3\,\mathrm{width}(\mathcal{M})}\|\boldsymbol{\chi}_b\|_1 \tag{115}$$

$$\leq \frac{\Delta_{(n-t+1)}}{3\,\mathrm{width}(\mathcal{M})}\|\boldsymbol{\chi}_b\|_1 \leq \frac{\Delta_{(n-t+1)}}{3}, \tag{116}$$

where Eq. (114) follows from Lemma 1; Eq. (115) follows from Lemma 17, the assumption on $\tau_t$ and $(b_+ \cup b_-) \cap (A_t \cup B_t) = \emptyset$ (by Lemma 18); and Eq. (116) follows from the fact $\|\boldsymbol{\chi}_b\|_1 \leq \mathrm{width}(\mathcal{M})$ (since $b \in \mathcal{B}$) and that $\langle \boldsymbol{w}, \boldsymbol{\chi}_b \rangle \geq \Delta_e \geq 0$. $\qquad \square$

### D.3 Proof of Theorem 3

Using these technical lemmas, we are now ready to prove Theorem 3. For reader's convenience, we first restate Theorem 3 as follows.

**Theorem 3.** *Given any $T > n$, any decision class $\mathcal{M} \subseteq 2^{[n]}$ and any expected rewards $\boldsymbol{w} \in \mathbb{R}^n$. Assume that the reward distribution $\varphi_e$ for each arm $e \in [n]$ has mean $w(e)$ with an $R$-sub-Gaussian tail. Let $\Delta_{(1)}, \ldots, \Delta_{(n)}$ be a permutation of $\Delta_1, \ldots, \Delta_n$ (defined in Eq. (1)) such that $\Delta_{(1)} \leq \ldots \ldots \Delta_{(n)}$. Define $\mathbf{H}_2 \triangleq \max_{i \in [n]} i \Delta_{(i)}^{-2}$. Then, the CSAR algorithm uses at most $T$ samples and outputs a solution $\mathsf{Out} \in \mathcal{M} \cup \{\bot\}$ such that*

$$\Pr[\mathsf{Out} \neq M_*] \leq n^2 \exp\left(-\frac{(T-n)}{18R^2 \tilde{\log}(n)\, \mathrm{width}(\mathcal{M})^2 \mathbf{H}_2}\right), \tag{8}$$

*where $\tilde{\log}(n) \triangleq \sum_{i=1}^n i^{-1}$, $M_* = \arg\max_{M \in \mathcal{M}} w(M)$ and $\mathrm{width}(\mathcal{M})$ is defined in Eq. (4).*

*Proof.* First, we show that the algorithm takes at most $T$ samples. It is easy to see that exactly one arm is pulled for $\tilde{T}_1$ times, one arm is pulled for $\tilde{T}_2$ times, ..., and one arm is pulled for $\tilde{T}_n$ times. Therefore, the total number of samples used by the algorithm is bounded by

$$\sum_{t=1}^n \tilde{T}_t \leq \sum_{t=1}^n \left(\frac{T-n}{\tilde{\log}(n)(n-t+1)} + 1\right)$$
$$= \frac{T-n}{\tilde{\log}(n)} \tilde{\log}(n) + n = T.$$

By Lemma 16, we know that the event $\tau \triangleq \bigcap_{t=1}^T \tau_t$ occurs with probability at least $1 - n^2 \exp\left(-\frac{(T-n)}{18R^2 \tilde{\log}(n)\, \mathrm{width}(\mathcal{M})^2 \mathbf{H}_2}\right)$. Therefore, we only need to prove that, under event $\tau$, the algorithm outputs $M_*$. We will assume that event $\tau$ occurs in the rest of the proof.

We prove by induction. Fix a phase $t \in [t]$. Suppose that the algorithm does not make any error before phase $t$, i.e., $A_t \subseteq M_*$ and $B_t \cap M_* = \emptyset$. We show that the algorithm does not err at phase $t$.

At the beginning of phase $t$, there are exactly $t-1$ inactive arms $|A_t \cup B_t| = t-1$. Therefore there must exists an active arm $e_t \in [n] \backslash (A_t \cup B_t)$ such that $\Delta_{e_t} \geq \Delta_{(n-t+1)}$. Hence, by Lemma 20, we have

$$\bar{w}_t(M_t) - \bar{w}_t(M_{t,e_t}) \geq \frac{2}{3}\Delta_{(n-t+1)}. \tag{117}$$

Notice that the algorithm makes an error in phase $t$ if and only if it accepts an arm $p_t \notin M_*$ or rejects an arm $p_t \in M_*$. On the other hand, by design, arm $p_t$ is accepted when $p_t \in M_t$ and is rejected when $p_t \notin M_t$. Therefore, we see that the algorithm makes an error in phase $t$ if and only if $p_t \in (M_* \cap \neg M_t) \cup (\neg M_* \cap M_t)$.

Suppose that $p_t \in (M_* \cap \neg M_t) \cup (\neg M_* \cap M_t)$. Now appeal to Lemma 21, we see that

$$\bar{w}_t(M_t) - \bar{w}_t(\tilde{M}_{t,p_t}) \leq \frac{1}{3}\Delta_{(n-t+1)}. \tag{118}$$

By combining Eq. (117) and Eq. (118), we see that

$$\bar{w}_t(M_t) - \bar{w}_t(\tilde{M}_{t,p_t}) \leq \frac{1}{3}\Delta_{(n-t+1)} < \frac{2}{3}\Delta_{(n-t+1)} \leq \bar{w}_t(M_t) - \bar{w}_t(M_{t,e_t}). \tag{119}$$

However Eq. (119) is contradictory to the definition of $p_t \triangleq \arg\max_{e \in [n] \backslash (A_t \cup B_t)} \bar{w}_t(M_t) - \bar{w}_t(\tilde{M}_{t,e})$. Therefore we have proven that $p_t \notin (M_* \cap \neg M_t) \cup (\neg M_* \cap M_t)$. This means that the algorithm does not err at phase $t$, or equivalently $A_{t+1} \subseteq M_*$ and $B_{t+1} \cap M_* = \emptyset$. By induction, we have proven that the algorithm does not err at any phase $t \in [n]$.

Hence we have $A_{n+1} \subseteq M_*$ and $B_{n+1} \subseteq \neg M_*$ in the final phase. Notice that $|A_{n+1}| + |B_{n+1}| = n$ and $A_{n+1} \cap B_{n+1} = \emptyset$. This means that $A_{n+1} = M_*$ and $B_{n+1} = \neg M_*$. Therefore the algorithm outputs $\mathsf{Out} = A_{n+1} = M_*$ after phase $n$. $\square$

## E  Analysis of the Uniform Allocation Algorithm

In this section, we analyze the performance of a simple benchmark strategy UNI which plays each arm for a equal number of times and then calls a maximization oracle using the empirical means of arms as input. The pseudo-code of the UNI algorithm is listed in Algorithm 3.

---
**Algorithm 3** UNI: Uniform Allocation
---
**Require:** Budget: $T > 0$; Maximization oracle: $\text{Oracle} : \mathbb{R}^n \to \mathcal{M}$.
1:   Pull each arm $e \in [n]$ for $\lfloor T/n \rfloor$ times.
2:   Compute the empirical means $\bar{\boldsymbol{w}} \in \mathbb{R}^n$ of each arm.
3:   $\text{Out} \leftarrow \text{Oracle}(\bar{\boldsymbol{w}})$
4:   **return:** Out
---

The next theorem upper bounds the probability of error of UNI.

**Theorem 7.** *Given any $T > n$, any decision class $\mathcal{M} \subseteq 2^{[n]}$ and any expected rewards $\boldsymbol{w} \in \mathbb{R}^n$. Assume that the reward distribution $\varphi_e$ for each arm $e \in [n]$ has mean $w(e)$ with an R-sub-Gaussian tail. Also assume without loss of generality that $T$ is a multiple of $n$. Define $\Delta_{(1)} = \min_{i \in [n]} \Delta_i$ and $\mathbf{H}_3 = n\Delta_{(1)}^{-2}$. Then, the output $\text{Out}$ of the UNI algorithm satisfies*

$$\Pr[\text{Out} \neq M_*] \leq 2n \exp \left( -\frac{T}{18R^2 \operatorname{width}(\mathcal{M})^2 \mathbf{H}_3} \right), \tag{120}$$

*where $M_* = \arg\max_{M \in \mathcal{M}} w(M)$.*

From Theorem 7, we see that the UNI algorithm could be significantly worse than CLUCB and CSAR, since it is clear that $\mathbf{H}_3 \geq \mathbf{H} \geq \mathbf{H}_2$ and potentially one has $\mathbf{H}_3 \gg \mathbf{H} \geq \mathbf{H}_2$ for a large number of arms with heterogeneous gaps.

Now we prove Theorem 7. The proof is straightforward using tools of exchange classes.

*Proof.* Define $\Delta_{(1)} = \min_{i \in [n]} \Delta_i$. Define random event $\xi$ as follows

$$\xi = \left\{ \forall i \in [n], \quad |\bar{w}(i) - w(i)| < \frac{\Delta_{(1)}}{3 \operatorname{width}(\mathcal{M})} \right\}.$$

Notice that each arm is sampled for $\lfloor \frac{T}{n} \rfloor$ times. Therefore, using Hoeffding's inequality (Lemma 6) and union bound, we can bound $\Pr[\xi]$ as follows. Fix any $i \in [n]$, by Hoeffding's inequality, we have

$$\Pr \left[ |\bar{w}(i) - w(i)| \geq \frac{\Delta_{(1)}}{3 \operatorname{width}(\mathcal{M})} \right] \leq 2\exp \left( -\frac{T\Delta_{(1)}^2}{18R^2 n \operatorname{width}(\mathcal{M})^2} \right).$$

Then, using a union bound, we obtain

$$\Pr\left[ \xi \right] \geq 1 - 2n \exp \left( -\frac{T\Delta_{(1)}^2}{18nR^2 \operatorname{width}(\mathcal{M})^2} \right).$$

In addition, using an argument very similar to Lemma 17, one can show that, on event $\xi$, for any vector $\boldsymbol{a} \in \mathbb{R}^n$, it holds that

$$|\langle \bar{\boldsymbol{w}}, \boldsymbol{a} \rangle - \langle \boldsymbol{w}, \boldsymbol{a} \rangle| < \frac{\Delta_{(1)}}{3 \operatorname{width}(\mathcal{M})} \|\boldsymbol{a}\|_1 . \tag{121}$$

Now we claim that, on the event $\xi$, we have $\text{Out} = M_*$. Note that theorem follows immediately from the claim. Next, we prove this claim.

Suppose that, on the contrary, $\text{Out} \neq M_*$. In this case, let us write $M = \text{Out}$. We also fix $\mathcal{B} \in \arg\min_{\mathcal{B}' \in \text{Exchange}(\mathcal{M})} \operatorname{width}(\mathcal{B})$. Notice that by definition $\operatorname{width}(\mathcal{B}) = \operatorname{width}(\mathcal{M})$.

Since $M \neq M_*$, we see that there exists $e \in (M \backslash M_*) \cup (M_* \backslash M)$. Now, by Lemma 2, we obtain that there exists $b = (b_+, b_-) \in \mathcal{B}$ such that $e \in b_+ \cup b_-$, $b_- \subseteq M \backslash M_*$, $b_+ \subseteq M_* \backslash M$, $M \oplus b \in \mathcal{M}$ and $\langle \boldsymbol{w}, \boldsymbol{\chi}_b \rangle \geq \Delta_e$. Also notice that $\Delta_e \geq \Delta_{(1)}$. Therefore $\langle \boldsymbol{w}, \boldsymbol{\chi}_b \rangle \geq \Delta_{(1)}$.

Consider $M' \triangleq M \oplus b$. We have

$$\begin{aligned} \bar{w}(M') - \bar{w}(M) &= \langle \bar{\boldsymbol{w}}, \boldsymbol{\chi}_{M'} \rangle - \langle \bar{\boldsymbol{w}}, \boldsymbol{\chi}_M \rangle \\ &= \langle \bar{\boldsymbol{w}}, \boldsymbol{\chi}_b \rangle \end{aligned} \tag{122}$$

$$> \langle \boldsymbol{w}, \boldsymbol{\chi}_b \rangle - \frac{\Delta_{(1)}}{3 \, \mathrm{width}(\mathcal{M})} \|\boldsymbol{\chi}_b\|_1 \tag{123}$$

$$\geq \Delta_{(1)} - \frac{\Delta_{(1)}}{3} \tag{124}$$

$$= \frac{2}{3}\Delta_{(1)} > 0, \tag{125}$$

where Eq. (122) follows from Lemma 1; Eq. (123) follows from Eq. (121); and Eq. (124) follows from the fact that $b \in \mathcal{B}$ and hence $\|\boldsymbol{\chi}_b\|_1 = |b_+| + |b_-| \leq \mathrm{width}(\mathcal{B}) = \mathrm{width}(\mathcal{M})$.

Hence, we have shown that $\bar{w}(M') > \bar{w}(M)$. However this contradicts to the fact that $\bar{w}(M) = \max_{M_1 \in \mathcal{M}} \bar{w}(M_1)$ (by the definition of maximization oracle). Hence, by contradiction, we have proven that $\mathsf{Out} = M_*$. $\qquad\square$

## F  Exchange Classes for Example Decision Classes

In this section, we give formal constructions of the decision classes discussed in Example 1, 2 and 3. Further, we bound the width of exchange classes for different examples. These bounds are proven using concrete constructions of exchange classes (Fact 1 through 5). The constructed exchange classes embody natural combinatorial structures. We illustrate the constructed exchange classes in Figure 3.

(a) An exchange set from $\mathcal{B}_{\mathrm{MATROID}(n)}$ (TOPK: Fact 2; each cylinder represents an arm).

(b) An exchange set from $\mathcal{B}_{\mathrm{MATROID}(n)}$ (Spanning trees: Fact 1; each edge corresponds to an arm).

(c) An exchange set from $\mathcal{B}_{\mathrm{MATCH}(G)}$ (Matchings: Fact 4; each edge corresponds to an arm)

(d) An exchange set from $\mathcal{B}_{\mathrm{PATH}(G)}$ (Paths: Fact 5; each edge corresponds to an arm).

Figure 3: Examples of exchange sets belonging to the exchange classes $\mathcal{B}_{\mathrm{MATROID}(n)}$ (one for TOPK and one for spanning tree), $\mathcal{B}_{\mathrm{MATCH}(G)}$ and $\mathcal{B}_{\mathrm{PATH}(G)}$: green-solid elements constitute the set $b_+$, red-dotted elements constitute the set $b_-$ and an example exchange set is given by $b = (b_+, b_-)$. In Figure 3a, we use TOPK as a specific instance of matroid decision class. In Figure 3b, we use spanning tree as a specific instance of matroid decision class.

**Notation.** We need one extra notation. Let $\sigma : E \to [n]$ be a bijection from some set $E$ with $n$ elements to $[n]$. Let $A \subseteq E$ be an arbitrary set, we define $\sigma(A) \triangleq \{\sigma(a) \mid a \in A\}$. Conversely, for all $M \subseteq [n]$, we define $\sigma^{-1}(M) \triangleq \{\sigma^{-1}(e) \mid e \in M\}$.

**Fact 1** (Matroid). *Let $T = (E, \mathcal{I})$ be an arbitrary matroid, where $E$ is the ground set of $n$ elements and $\mathcal{I}$ is the family of subsets of $E$ called in the independent sets which satisfy the axioms of matroids*

[3]. *Let $\sigma\colon E \to [n]$ be a bijection from $E$ to $[n]$. Let $\mathcal{M}_{\mathrm{MATROID}(T)}$ correspond to the collection of all bases of matroid $T$ and formally we define*

$$\mathcal{M}_{\mathrm{MATROID}(T)} = \left\{ M \subseteq [n] \mid \sigma^{-1}(M) \text{ is a basis of } T \right\}. \tag{126}$$

*Define the exchange class*

$$\mathcal{B}_{\mathrm{MATROID}(n)} = \left\{ (\{i\}, \{j\}) \mid \forall i \in [n], j \in [n] \right\}. \tag{127}$$

*Then we have $\mathcal{B}_{\mathrm{MATROID}(n)} \in \mathrm{Exchange}(\mathcal{M}_{\mathrm{MATROID}(T)})$. In addition, we have $\mathrm{width}(\mathcal{B}_{\mathrm{MATROID}(n)}) = 2$, which implies that $\mathrm{width}(\mathcal{M}_{\mathrm{MATROID}(T)}) \leq 2$.*

To prove Fact 1, we first recall a well-known result from matroid theory which is referred as the strong basis exchange property.

**Lemma 22** (Strong basis exchange [26])**.** *Let $\mathcal{A}$ be the set of all bases of a matroid $T = (E, \mathcal{I})$. Let $A_1, A_2 \in \mathcal{A}$ be two bases. Then for all $x \in A_1 \backslash A_2$, there exists $y \in A_2 \backslash A_1$ such that $A_1 \backslash \{x\} \cup \{y\} \in \mathcal{A}$ and $A_2 \backslash \{y\} \cup \{x\} \in \mathcal{A}$.*

Using Lemma 22, we are ready to prove Fact 1.

*Proof of Fact 1.* Fix a matroid $T = (E, \mathcal{I})$ where $|E| = n$ and fix the bijection $\sigma\colon E \to [n]$. Let $\mathcal{M}_{\mathrm{MATROID}(T)}$ be defined as in Eq. (126) and let $\mathcal{B}_{\mathrm{MATROID}(n)}$ be defined as in Eq. (127). Let $\mathcal{A}$ denote the set of all bases of $T$. By definition, we have $\mathcal{M}_{\mathrm{MATROID}(T)} = \{\sigma(A) \mid A \in \mathcal{A}\}$.

Now we show that $\mathcal{B}_{\mathrm{MATROID}(n)}$ is an exchange class for $\mathcal{M}_{\mathrm{MATROID}(T)}$. Let $M, M'$ be two different elements of $\mathcal{M}_{\mathrm{MATROID}(T)}$. By definition, we see that $\sigma^{-1}(M)$ and $\sigma^{-1}(M')$ are two bases of $T$. Consider any $e \in M \backslash M'$. Let $x = \sigma^{-1}(e)$. We see that $x \in \sigma^{-1}(M) \backslash \sigma^{-1}(M')$.

By Lemma 22, we see that there exists $y \in \sigma^{-1}(M') \backslash \sigma^{-1}(M)$ such that

$$\sigma^{-1}(M) \backslash \{x\} \cup \{y\} \in \mathcal{A} \quad \text{and} \quad \sigma^{-1}(M') \backslash \{y\} \cup \{x\} \in \mathcal{A}. \tag{128}$$

Now we define exchange set $b = (b_+, b_-)$ where $b_+ = \{\sigma(y)\}$ and $b_- = \{\sigma(x)\}$. By Eq. (128) and the fact that $\sigma$ is a bijection, we see that $M \oplus b \in \mathcal{M}_{\mathrm{MATROID}(T)}$ and $M' \ominus b \in \mathcal{M}_{\mathrm{MATROID}(T)}$. We also have $b \in \mathcal{B}_{\mathrm{MATROID}(n)}$. Due to the fact that $M, M'$ and $e$ are chosen arbitrarily, we have verified that $\mathcal{B}_{\mathrm{MATROID}(n)}$ is an exchange class for $\mathcal{M}_{\mathrm{MATROID}(T)}$.

To conclude, we observe that $\mathrm{width}(\mathcal{B}_{\mathrm{MATROID}(n)}) = 2$. $\qquad\square$

Now we show that TOPK and MB are special cases of the family of decision classes of derived from bases of matroids. This enable us to apply Fact 1 to construct exchange classes and bound the widths of these decision classes. We also note that one may use a more direct way to construct the exchange classes for these two problems without appealing to matroids.

**Fact 2** (TOPK)**.** *For all $K \in [n]$, let $\mathcal{M}_{\mathrm{TOPK}(K)} = \{M \subseteq [n] \mid |M| = K\}$ be the collection of all subsets of size $K$. Then we have $\mathcal{B}_{\mathrm{MATROID}(n)} \in \mathrm{Exchange}(\mathcal{M}_{\mathrm{TOPK}(K)})$ and $\mathrm{width}(\mathcal{M}_{\mathrm{TOPK}(K)}) \leq 2$.*

*Proof.* Let $U_n^K = ([n], \mathcal{I}_K)$ where $\mathcal{I}_K$ is given by

$$\mathcal{I}_K = \left\{ M \subseteq [n] \mid |M| \leq K \right\}.$$

Recall that $U_n^K$ is a matroid (in particular, a uniform matroid of rank $K$) [26]. We know that a subset $M$ of $[n]$ is basis of $U_n^K$ if and only if $|M| = K$. Therefore, we have $\mathcal{M}_{\mathrm{TOPK}(K)} = \mathcal{M}_{\mathrm{MATROID}(U_n^K)}$. Then we can conclude immediately by using Fact 1. $\qquad\square$

**Fact 3** (MB)**.** *For any partition $\mathcal{A} = \{A_1, \ldots, A_m\}$ of $[n]$, we define*

$$\mathcal{M}_{\mathrm{MB}(\mathcal{A})} = \left\{ M \subseteq [n] \mid \forall i \in [m] \quad |M \cap A_i| = 1 \right\}.$$

*Then we have $\mathcal{B}_{\mathrm{MATROID}(n)} \in \mathrm{Exchange}(\mathcal{M}_{\mathrm{TOPK}(K)})$ and $\mathrm{width}(\mathcal{M}_{\mathrm{MB}(\mathcal{A})}) \leq 2$.*

*Proof.* Let $P_{\mathcal{A}} = ([n], \mathcal{I}_{\mathcal{A}})$ where $\mathcal{I}_{\mathcal{A}}$ is given by

$$\mathcal{I}_{\mathcal{A}} = \big\{ M \subseteq [n] \mid \forall i \in [m] \quad |M \cap A_i| \leq 1 \big\}.$$

It can be shown that $P_{\mathcal{A}}$ is a matroid (known as partition matroid [26]) and each basis $M$ of $P_{\mathcal{A}}$ satisfies $|M \cap A_i| = 1$ for all $i \in [m]$. Therefore we have $\mathcal{M}_{\mathrm{MB}(\mathcal{A})} = \mathcal{M}_{\mathrm{MATROID}(P_{\mathcal{A}})}$. Then the conclusion follows immediately from Fact 1. $\qquad\square$

**Fact 4** (Matching). *Let $G(V, E)$ be a bipartite graph with $n$ edges. Let $\sigma \colon E \to [n]$ be a bijection. Let $\mathcal{A}$ be the set of all valid matchings in $G$. We define $\mathcal{M}_{\mathrm{MATCH}(G)}$ as follows*

$$\mathcal{M}_{\mathrm{MATCH}(G)} = \big\{ \sigma(A) \mid A \in \mathcal{A} \big\}.$$

*Define the exchange class*

$$\mathcal{B}_{\mathrm{MATCH}(G)} = \Big\{ (\sigma(c_+), \sigma(c_-)) \mid \exists c \in \mathcal{C} \cup \mathcal{P}, \text{ the edges of } c \text{ alternate between } c_+, c_- \Big\},$$

*where $\mathcal{C}$ is the set of all cycles in $G$ and $\mathcal{P}$ is the set of all paths in $G$. Then we have $\mathcal{B}_{\mathrm{MATCH}(G)} \in \mathrm{Exchange}(\mathcal{M}_{\mathrm{MATCH}(G)})$. In addition, we have $\mathrm{width}(\mathcal{B}_{\mathrm{MATCH}(G)}) \leq |V|$, which implies that $\mathrm{width}(\mathcal{M}_{\mathrm{MATROID}(T)}) \leq |V|$.*

To prove Fact 4, we recall a classical result on graph matching which characterizes the properties of augmenting cycles and augmenting paths [4].

**Lemma 23.** *Let $G(V, E)$ be a bipartite graph. Let $M$ and $M'$ be two different matchings of $G$. Then the induced graph $G'$ from the symmetric difference $(M \backslash M') \cup (M' \backslash M)$ consists of connected components that are one of the following*

- *An even cycle whose edges alternate between $M$ and $M'$.*

- *A simple path whose edges alternate between $M$ and $M'$.*

*Proof of Fact 4.* Fix a bipartite graph $G(V, E)$ and a bijection $\sigma \colon E \to [n]$. Let $M, M' \in \mathcal{M}_{\mathrm{MATCH}(G)}$ be two different elements of $\mathcal{M}_{\mathrm{MATCH}(G)}$ and consider an arbitrary $e \in M \backslash M'$. On a high level perspective, we construct an exchange class which contains all augmenting cycles and paths of $G$. We know that the symmetric difference between $M$ and $M'$ can be decomposed into a collection of disjoint augmenting cycles and paths. And $e$ must be on one of the augmenting cycle or path. Then, since "applying" this augmenting cycle/path on $M$ will yield another valid matching which does not contains $e$. We see that this meets the requirements of an exchange class. In the rest of the proof, we carry out the technical details of this argument.

Define $A = \sigma^{-1}(M)$ and $A' = \sigma^{-1}(M')$. Let $a = \sigma^{-1}(e)$. Then $A, A'$ are two matchings of $G$. Let $G'$ be the induced graph from the symmetric difference $(A \backslash A') \cup (A' \backslash A)$. Let $C$ be the connected component of $G'$ which contains the edge $a$. Therefore, by Lemma 23, we see that $C$ is either an even cycle or a simple path with edges alternating between $A$ and $A'$. Let $C_+$ contains the edges of $C$ that belongs to $A' \backslash A$. Similarly, let $C_-$ contains the edges of $C$ that belongs to $A \backslash A'$. Define $b_+ = \sigma(C_+)$ and $b_- = \sigma(C_-)$. Let $b = (b_+, b_-)$ be an exchange set.

Since $b$ corresponds to either an augmenting path or an augmenting cycle, we see that $b \in \mathcal{B}_{\mathrm{MATCH}(G)}$. Since $a \in C_-$, we obtain that $e \in b_-$. In addition, note that $C_+ \subseteq A' \backslash A$ and $C_- \subseteq A \backslash A'$. Therefore we have $b_+ \subseteq M' \backslash M$ and $b_- \subseteq M \backslash M'$.

Since $C$ is an $A$-augmenting path/cycle, therefore it immediately holds that $A \backslash C_- \cup C_+$ is a valid matching. Therefore, we have $M \backslash b_- \cup b_+ \in \mathcal{M}_{\mathrm{MATCH}(G)}$. Similarly, one can show that $M' \backslash b_+ \cup b_- \in \mathcal{M}_{\mathrm{MATCH}(G)}$. Hence we have shown that $\mathcal{B}_{\mathrm{MATCH}(G)}$ is an exchange class for $\mathcal{M}_{\mathrm{MATCH}(G)}$. $\qquad\square$

**Fact 5** (Path). *Let $G(V, E)$ be a directed acyclic graph with $n$ edges. Let $s, t \in V$ be two different vertices. Let $\sigma \colon E \to [n]$ be a bijection. Let $\mathcal{A}(s, t)$ be the set of all valid paths from $s$ to $t$ in $G$. We define $\mathcal{M}_{\mathrm{PATH}(G, s, t)}$ as follows*

$$\mathcal{M}_{\mathrm{PATH}(G, s, t)} = \big\{ \sigma(A) \mid A \in \mathcal{A}(s, t) \big\}.$$

*Define exchange class*

$$\mathcal{B}_{\mathrm{PATH}(G)} = \{ (\sigma^{-1}(p), \sigma^{-1}(q)) \mid p, q \text{ are the arcs of two disjoint paths of } G \text{ with same endpoints} \}.$$

*Then, we have $\mathcal{B}_{\mathrm{PATH}(G)} \in \mathrm{Exchange}(\mathcal{M}_{\mathrm{PATH}(G, s, t)})$. In addition, we have $\mathrm{width}(\mathcal{B}_{\mathrm{PATH}(G)}) \leq |V|$ and therefore $\mathrm{width}(\mathcal{M}_{\mathrm{PATH}(G, s, t)}) \leq |V|$.*

*Proof.* Fix a directed acyclic graph $G(V,E)$ and a bijection $\sigma\colon E \to [n]$. Fix two vertices $s,t \in V$.

We prove that $\mathcal{B}_{\text{PATH}(G)}$ is an exchange class for $\mathcal{M}_{\text{PATH}(G,s,t)}$. Let $M, M' \in \mathcal{M}_{\text{PATH}(G,s,t)}$ be two different sets. Then $\sigma^{-1}(M), \sigma^{-1}(M')$ are two sets of arcs corresponding to two different paths from $s$ to $t$. Let $P = (v_1, \ldots, v_{n_1}), P' = (v'_1, \ldots, v'_{n_2})$ denote the two paths, respectively. Notice that $s = v_1 = v'_1$ and $t = v_{n_1} = v'_{n_2}$. We also denote $E(P) = \sigma^{-1}(M)$ and $E(P') = \sigma^{-1}(M')$.

Fix some $e \in M\backslash M'$ and define $a = \sigma^{-1}(e)$. Suppose that $a$ is an arc from $u$ to $v$. Since $a$ is on path $P$, there exists $i$ such that $v_i = u$ and $v_{i+1} = v$. Now we define $j_1 = \arg\max_{j \leq i, v_j \in P'} j$ and $j_2 = \arg\min_{j \geq i+1, v_j \in P'} j$. Notice that $j_1$ and $j_2$ are well-defined since $P$ and $P'$ intersects on at least two vertices ($s$ and $t$). Let $v'_{k_1} = v_{j_1}$ and $v'_{k_2} = v_{j_2}$ be the corresponding indices in $P'$. Then, we see that $Q_1 = (v_{j_1}, v_{j_1+1}, \ldots, v_{j_2})$ and $Q_2 = (v'_{k_1}, v'_{k_1+1}, \ldots, v'_{k_2})$ are two different paths from $v_{j_1}$ to $v_{j_2}$. Denote the sets of arcs of $Q_1$ and $Q_2$ as $E(Q_1)$ and $E(Q_2)$.

Let $b = (b_+, b_-)$, where $b_+ = \sigma(E(Q_2)), b_- = \sigma(E(Q_1))$. We see that $b \in \mathcal{B}_{\text{PATH}(G)}$. It is clear that $a \in E(Q_1)$, $E(Q_1) \subseteq E(P)\backslash E(P')$ and $E(Q_2) \subseteq E(P')\backslash E(P)$. Therefore $e \in b_-$, $b_- \subseteq M\backslash M'$ and $b_+ \subseteq M'\backslash M$.

Now it is easy to check that $E(P_1)\backslash E(Q_1) \cup E(Q_2)$ equals the set of arcs of path $(v_1, \ldots, v_{j_1}, v'_{k_1+1}, \ldots, v'_{k_2-1}, v_{j_2}, \ldots, v_{n_1})$ (recall that $v_{j_1} = v'_{k_1}$ and $v_{j_2} = v'_{k_2}$). This means that $E(P_1)\backslash E(Q_1) \cup E(Q_2) \in \mathcal{A}(s,t)$ and therefore $M\backslash b_- \cup b_+ \in \mathcal{M}_{\text{PATH}(G,s,t)}$. Using a similar argument, one can show that $M'\backslash b_+ \cup b_- \in \mathcal{M}_{\text{PATH}(G,s,t)}$ and hence we have verified that $\mathcal{B}_{\text{PATH}(G)} \in \text{Exchange}(\mathcal{M}_{\text{PATH}(G,s,t)})$. $\qquad\square$

# G   Equivalence Between Constrained Oracles and Maximization Oracles

In this section, we present a general method to implement constrained oracles using maximization oracles. The idea of the reduction is simple: one can impose the negative constrains $B$ by setting the corresponding weights to be sufficiently small; and one can impose the positive constrains $A$ by setting the corresponding weights to be sufficiently large. The reduction method is shown in Algorithm 4. The correctness of the reduction is proven in Lemma 24. Furthermore, it is trivial to reduce from maximization oracles to constrained oracles. Therefore, Lemma 24 shows that maximization oracles are equivalent to constrained oracles up to a transformation on the weight vector.

---

**Algorithm 4** COracle($\boldsymbol{w}, A, B$)

**Require:** $\boldsymbol{w} \in \mathbb{R}^n$, $A \subseteq [n]$, $B \subseteq [n]$; Maximization oracle Oracle $: \mathbb{R}^n \to \mathcal{M}$
1:   $L_1 \leftarrow \|\boldsymbol{w}\|_1$
2:   **for** $i = 1, \ldots, n$ **do**
3:      **if** $i \in A$ **then**
4:         $w_1(i) \leftarrow 3L_1$
5:      **else**
6:         $w_1(i) \leftarrow w(i)$
7:   $L_2 \leftarrow \|\boldsymbol{w}_1\|_1$
8:   **for** $i = 1, \ldots, n$ **do**
9:      **if** $i \in B$ **then**
10:        $w_2(i) \leftarrow -3L_2$
11:     **else**
12:        $w_2(i) \leftarrow w_1(i)$
13:  $M \leftarrow \text{Oracle}(\boldsymbol{w}_2)$
14:  **if** $B \cap M = \emptyset$ and $A \subseteq M$ **then**
15:     Out $= M$
16:  **else**
17:     Out $= \perp$
18:  **return:** Out

---

**Lemma 24.** *Given $\mathcal{M} \subseteq 2^{[n]}$, $\boldsymbol{w} \in \mathbb{R}^n$, $A \subseteq [n]$ and $B \subseteq [n]$, suppose that $A \cap B = \emptyset$. Then the output* Out *of Algorithm 4 satisfies* Out $\in \arg\max_{M \in \mathcal{M}, A \subseteq M, B \cap M = \emptyset} w(M)$ *where we use the convention that the* $\arg\max$ *of an empty set is $\perp$. Therefore Algorithm 4 is a valid constrained oracle.*

*Proof.* Let $\boldsymbol{w}_1$ and $\boldsymbol{w}_2$ be defined as in Algoritm 4. Let $M = \text{Oracle}(\boldsymbol{w}_2)$. Let $\mathcal{M}_{A,B} = \{M \in \mathcal{M} \mid A \subseteq M, B \cap M = \emptyset\}$ be the subset of $\mathcal{M}$ which satisfies the constraints. If $\mathcal{M}_{A,B} = \emptyset$, then it is clear $M$ cannot satisfy both of the constraints $A \subseteq M$ and $B \cap M = \emptyset$. Therefore Algorithm 4 returns $\perp$ in this case.

In the rest of the proof, we assume that $\mathcal{M}_{A,B} \neq \emptyset$. Since $\mathcal{M}_{A,B}$ is non-empty, we can fix an arbitrary $M_0 \in \mathcal{M}_{A,B}$, which will be used later in the proof. We will also frequently use the fact that, for all $\boldsymbol{v} \in \mathbb{R}^n$ and all $S \subseteq [n]$, we have

$$- \|\boldsymbol{v}\|_1 \leq v(S) \leq \|\boldsymbol{v}\|_1. \tag{129}$$

First we claim that $B \cap M = \emptyset$. Suppose that $B \cap M \neq \emptyset$. Then there exists $i \in B \cap M$ and we fix such an $i$. Then we have

$$w_2(M) = w_2(M \backslash \{i\}) + w_2(i)$$
$$\leq w_2(M \backslash B) + w_2(i) \tag{130}$$
$$= w_1(M \backslash B) + w_2(i) \tag{131}$$
$$\leq L_2 - 3L_2 = -2L_2, \tag{132}$$

where Eq. (130) follows from the fact that $w_2(j) = -L_2 \leq 0$ for all $j \in B \backslash \{i\}$; Eq. (131) holds since $\boldsymbol{w}_1$ and $\boldsymbol{w}_2$ coincide on all entries of $M \backslash B$; and Eq. (132) follows from the definition $L_2 = \|\boldsymbol{w}_1\|_1$ and Eq. (129).

On the other hand, observing that $B \cap M_0 = \emptyset$, we can bound $w_2(M_0)$ as follows

$$w_2(M_0) = w_1(M_0) \geq -L_2.$$

Therefore we see that $w_2(M_0) > w_2(M)$. However, this contradicts to the definition of $M$ since $M \in \arg\max_{M' \in \mathcal{M}} w_2(M')$. Therefore our claim $B \cap M = \emptyset$ is true. By this claim and since $\boldsymbol{w}_2$ and $\boldsymbol{w}_1$ coincide on entries of $[n] \backslash B$, we have

$$w_2(M) = w_1(M). \tag{133}$$

Next we claim that $A \subseteq M$. Suppose that $A \not\subseteq M$. Then we have

$$w_2(M) = w_1(M) = w_1(M \cap A) + w_1(M \backslash A)$$
$$= 3|M \cap A|L_1 + w(M \backslash A) \tag{134}$$
$$\leq (3|A| - 3)L_1 + L_1 \tag{135}$$
$$= (3|A| - 2)L_1, \tag{136}$$

where Eq. (134) follows from the definition of $\boldsymbol{w}_1$; and Eq. (135) follows from the assumption that $A \not\subseteq M$ and therefore $|M \cap A| \leq |A| - 1$.

On the other hand, using the fact that $A \subseteq M_0$ (since $M_0 \in \mathcal{M}_{A,B}$), we have

$$w_2(M_0) = w_1(M_0) = w_1(A) + w_1(M_0 \backslash A) \tag{137}$$
$$= 3|A|L_1 + w(M_0 \backslash A) \tag{138}$$
$$\geq 3|A|L_1 - L_1 \tag{139}$$
$$= (3|A| - 1)L_1, \tag{140}$$

where Eq. (137) follows from the fact that $M_0 \cap B = \emptyset$ and $A \subseteq M_0$; Eq. (138) follows from the definition of $\boldsymbol{w}_1$, which ensures that $\boldsymbol{w}_1$ and $\boldsymbol{w}$ coincide on $M_0 \backslash A$; and Eq. (139) follows from Eq. (129).

Therefore, by combining Eq. (136) and Eq. (140), we see that $w_2(M_0) > w_2(M)$. Again this contradicts to the definition of $M$, which proves the claim that $A \subseteq M$.

Now we see that $A \subseteq M$ and $B \cap M = \emptyset$, which means that $M \in \mathcal{M}_{A,B}$. Therefore, we remain to verify that $w(M) = \max_{M' \in \mathcal{M}_{A,B}} w(M')$. Suppose that there exists $M_1 \in \mathcal{M}_{A,B}$ such that $w(M_1) > w(M)$. Notice that $B \cap M_1 = \emptyset$ and $A \subseteq M_1$, we have

$$w_2(M_1) = w_1(M_1) = w_1(M_1 \backslash A) + w_1(B) = w(M_1 \backslash A) + 3|A|L_1 = w(M_1) + 3|A|L_1 - w(A).$$

Similarly, one can show that $w_2(M) = w(M) + 3|A|L_1 - w(A)$. By combining with the assumption that $w(M_1) > w(M)$ we see that $w_2(M_1) > w_2(M)$, which contradicts to the definition of $M$. Hence we have verified that $w(M) = \max_{M' \in \mathcal{M}_{A,B}} w(M')$. $\square$

# H    Preliminary Experiments: Identifying the Minimum Spanning Tree

In this section, we present some preliminary experimental results of our algorithms CLUCB and CSAR. We conduct experiments on a real-world dataset with decision classes corresponding to collections of spanning trees. We compare our algorithms with the uniform allocation benchmark UNI discussed in Appendix E. The experiment results show that the proposed algorithms are considerably more sample efficient than the UNI algorithm, which agrees with our theoretical analysis.

(a) Network 1755            (b) Network 3257            (c) Network 3967

Figure 4: Comparison of empirical probability of errors with respect to $\mathbf{H}$.

(a) Network 1755            (b) Network 3257            (c) Network 3967

Figure 5: Empirical sample complexity of CLUCB with respect to $\mathbf{H}$.

(a) Network 1755 ($\mathbf{H} = 117.6$)    (b) Network 3257 ($\mathbf{H} = 181.5$)    (c) Network 3967 ($\mathbf{H} = 108.7$)

Figure 6: Empirical probability of error of CSAR and UNI with respect to budget size $T$.

**Setup.** Our task is to identify the optimal routing tree from a networking system which has the lowest expected latency in an exploration procedure, where one can obtain noisy measurements of latencies between different nodes. We model this problem as a CPE problem where the arms correspond to edges and the decision class corresponds to the set of spanning trees (which is a special case of matroids, as we have discussed in Example 3). We use a real-world dataset called RocketFuel [30], which contains several ISP networks with routing information such as average latencies between nodes pairs. We select three medium-sized ISP networks with numbers of edges ranging from 161 to 328. For each network, we model the latency $X(e)$ of edge $e$ as the sum of the given average latency $l(e)$ and an additive random noise $\mathcal{N}(0,1)$. Then we model the reward of edge $e$ as the

negative latency $-X(e)$ and therefore the expected reward of $e$ is given by $w(e) = -l(e)$. Notice that we now need to find the spanning tree that maximizes the expected reward, which is exactly an instance of CPE.

Since the ground-truth of expected reward $w$ is known, we can compute the ground-truth of the optimal set $M_*$ and the hardness measures $\mathbf{H}$. Furthermore, in order to investigate the relationship between $\mathbf{H}$ and sample complexity empirically, we generate a number of instances with different $\mathbf{H}$ by adjusting the expected reward of each arm $e \in M_*$ with a same additive quantity $c_0$ while not changing the optimality of $M_*$. By definition of $\mathbf{H}$, we see that $\mathbf{H}$ decreases when $c_0$ increases.

Recall that the sampling process of CSAR is divided into $n$ phases and, in the end of each phase, the algorithm either accepts or rejects an arm. In practice, for some decision classes, there may exists a phase $t < n$ such that there is only a unique set of arms which satisfies all constraints of phase $t$, i.e. $|\mathcal{M}_{A_t,B_t}| = 1$ (see Section 5 for the definitions of $A_t$, $B_t$ and $\mathcal{M}_{A_t,B_t}$). By the design of CSAR, one can see that this unique element of $\mathcal{M}_{A_t,B_t}$ will be the output of algorithm. In this case, the remaining budget for phases $t+1,\ldots,n$ are wasted in the sense that they do not affect the output. Therefore, in our experiment, we modify CSAR to utilize these remaining budget in this case by using the following heuristic. If $|\mathcal{M}_{A_t,B_t}| = 1$ for some $t < n$, we stop the algorithm and then re-run the algorithm using the remaining budget. During re-running the algorithm, the algorithm computes the empirical means of arms by using all samples including the samples from all previous runs of the algorithm. This process continues until the the algorithm terminates normally or the remaining budget is less than 30% of the original budget.

**Evaluation method.** We use the following evaluation procedure to compare the sample efficiency among CLUCB, CSAR and UNI. Since CSAR and UNI are both learning algorithms in the fixed budget setting, the comparison among them is straightforward: for each given budget, we run both algorithms with this budget independently for 1000 times and compare their empirical probability of errors (the fraction of runs where a tested algorithm fails to report the ground-truth optimal set $M_*$). On the other hand, we use the following procedure to compare CLUCB with other fixed budget algorithms. For each instance of ISP network, we run CLUCB independently for 1000 times. Suppose that the $i$-th run of CLUCB uses $T_i$ samples, we also run UNI and CSAR with budget $T_i$. Then we compare the empirical probability of errors of the tested algorithms after the 1000 runs are completed. In this way, we see that the compared algorithms use an equal number of samples in each run, which allows us to compare their sample efficiency. Finally, we set $\delta = 0.3$ for CLUCB throughout the experiments.

**Experimental results.** We test all competing algorithms using the aforementioned evaluation method. The experimental results are shown in Figure 4, Figure 5 and Figure 6. From the results (Figure 4 and Figure 6), we see that both CLUCB and CSAR are consistently more sample efficient than UNI by a large margin, i.e., they incur a smaller empirical probability of error than UNI when using a same number of samples. This matches our theoretical analyses of these algorithms. We also see that the probability of error of CLUCB is always smaller than the guarantee $\delta = 0.3$ (Figure 4) and the sample complexity of CLUCB is approximately linear in $\mathbf{H}$ (Figure 5), which agrees with our theory that the sample complexity bound for the spanning tree decision class is $\tilde{O}(\mathbf{H})$ (see Example 3).