[Reviews · NeurIPS 2014]

Submitted by Assigned_Reviewer_42

The paper deals with a pure exploration problem of combinatorial bandits with the following setup. Each arm has an associated probability distribution and at each round we are allowed to observe a sample from one of these distributions. Eventually our task is to find a subset of arms, out of a pre-defined family of subsets, maximizing the sum of associated rewards. This problem captures for example the problem of finding the best K arms, but also captures more complex problems such as best spanning tree, matching, etc.

The suggested algorithm requires an oracle that given a weight assignment for the arms finds the best valid subset with respect to it, meaning the valid subset whose corresponding sum of weights is maximal. The algorithm finds the best subset for the current hypothesis, then shifts the weights of the arms according to the lower and upper confidence bars so as to make the chosen subset the least attractive, and chooses a new subset accordingly. If the same subset is chosen the algorithm operates, otherwise the queried arm is that with the largest confidence region from the symmetric difference of the two chosen subsets. The proof uses a tool called an exchange class that can be seen a family of edges that can be used draw a path between any pair of valid subsets.

Generally, the paper is well written, including the supplementary material (though I did not go over all of the proofs). The techniques presented are elegant and have the potential to be used for other problems of similar nature.

Comments:
- There may be room for a comparison to the linear bandit setup. This is a more constraint setting where the queries are not to arms but rather to subsets, and the result is the reward for that subset.
- It should be stated more clearly in the introduction that the objective is a linear function over the arms in the subset
- Line 158 “equal to to the lower…”
- In the equations starting at line 788 there are missing parentheses
- In the equation of line 895 w_t is missing a bar

Summary: The paper is well written, its techniques are elegant and have the potential to be used for other problems of similar nature, and the problem being solved is interesting. The paper should definitely be accepted.

Submitted by Assigned_Reviewer_45

Summary: The authors provide a unified framework for the analysis of pure exploration in combinatorial bandit problems. Combinatorial bandit problems are bandit problems whose actions are sets of actions of an underlying MAB-problem. Thus they generalise the classic MAB problem. Pure exploration refers to settings in which algorithms are required only to return a guess of the optimal action; typically either the algorithm is required to return a guess after spending a fixed budget of queries (called "fixed budget"), and its performance is measured by the probability of this guess being correct, or the algorithm is required only to stop eventually, but to guarantee that the guess is accurate with probability (1-\delta) (called "fixed confidence"), and its performance is measured by the number of queriers it requires to achieve this goal. Several works have addressed particular problems within this spectrum, however here the authors succeed in unifying many of these in a single problem family.

Moreover, the authors are able to show that two generalisations of two of the algorithms currently proposed essentially solve pure exploration within this problem family. In particular, the authors give a lower bound for the number of queries needed to solve a fixed confidence problem, and give upper bounds for the performance of the CLUCB and CSAR algorithms (which solve the fixed confidence, and fixed budget problems respectively). Noting a relationship between fixed confidence and for fixed budget problems the authors conclude that the two algorithms essentially solve this problem family.

The authors achieve their results by introducing an object called an exchange class. These exchange classes are sets of ordered pairs of subsets of the set of actions of the underlying MAB problem, and they provide a way of describing differences and interpolating between different actions in the combinatorial bandit problem. In the proofs of their results, the authors use these classes to define an algebra on the actions of the combinatorial bandit, and thus to translate traditional concentration results into concentration results on actions of the combinatorial bandit. A fundamental quantity associated with exchange classes, which the authors call a width, appears in the resulting complexity bounds that they obtain.

Quality and Clarity: The main submission is very clearly written, and of a high quality. The appendix is extremely long, however having made it through the proof of Theorem 1, I can state that this is also clear, and I have found no errors. It is unfortunate that the arguments have to be so long and fiddly, but the resulting Theorems are both interesting and important.

Originality and Significance: I find that this is a highly original piece of work, although I disagree with the authors that their algorithms are novel. The algorithms proposed here are simply generalisations of existing algorithms. However, this is a minor quibble, and I would reiterate that the exchange-class-based insights are highly original (at least I have not seen anything along these lines before, but I have also not heard of the "patches" and "gadgets" that the authors say inspired their approach). It is also a significant piece of work. It establishes important results and brings together work and results in a significant number of previous papers.
Summary: This paper studies fixed budget and fixed confidence, pure exploration, combinatorial bandit problems, providing algorithms with performance bounds, and matching lower bounds for this family of problems.

Submitted by Assigned_Reviewer_46

This paper is (to the best of my knowledge) the first paper that studies best 'configuration' identification (or pure-exploration) in combinatorial multi-armed bandit. Two algorithms are proposed: CLUCB in the fixed-confidence setting and CSAR in the fixed-budget setting, and a lower bound is given in the fixed-confidence setting. The algorithms generalize is a very nice and natural way the LUCB and SAR algorithm designed for K best arms (or TopK) identification. As highlighted by the authors, the main strenght of the algorithms is that they treat in a unified way every combinatorial bandit problem. Their analysis feature a complexity term H can can be particularized to each combinatorial problem and appears natural. However, the paper suffer from two shortcomings: first, there is a small error in the proof of Theorem 1 (that can be corrected, an union bound is missing) and second, it is far too long. A NIPS reviewer will clearly NOT read 40 pages... I would advise you to remove all extensions (Appendix B. Appendix C.2 and Appendix E.) and keep them for an extended paper in a journal (where they will be more likely to be really reviewed). Moreover, the references to the theory of matroids (already in the main text) are a bit difficult to get for a non-initiated reader (like myself), so you should either define matroids in the main text or not mention them.

Theorem 1 is a high probabability bound on the number of samples from the arms needed to identify the best configuration in the fixed-confidence setting. Its proof consists in showing that on the event \xi on which for all t, the true means of all arms belong to their confidence interval, the algorithm outputs the right configuration and the number of sample used are upper bounded deterministically. And it is shown that the confidence intervals are chosen such that the event \xi has probability larger than 1 - \delta. This fact is actually not completely true with your choices of confidence intervals: in the display at line 706, the number of draws of arm e T_t(e) is random, so Hoeffding inequality cannot be applied directly: you should add a union bound (summing over all the possible values for T_t from 1 to t), which would lead to a log(CKt^3/\delta) -with some constant C- in your confidence interval. On the event \xi, the upper bound on the number of samples strongly relies on Lemma 10, which resorts to the width of a class \cM and to the concept of exchange class, which is particular to combinatorial problems. Finally, in the end of the proof of Theorem 1, you say that you get the upper bound "solving for T". Would it be possible that you really give an explicit upper bound on T rather than a big O?

I did not read the proof of the analysis of CSAR, but I guess it uses similar tools on exchange classes pluged into the analysis of SAR (and I hope other reviewers will confirm it is right).

The quality of the paper is good, it is well-structured, except for the presence of "Related works" in Section 6, which I think would be more naturally presented in the introduction. There are some weird formulations and english mistakes at some places, some of them are:
l.24 "we prove problem-dependent upper bounds of our algorithm" -> for our algorithms
l.102 " a probability of error bound of the CSAR algorithm" -> an upper bound on the probability of error of the CSAR algorithm
l.113 "standard assumptions of the stochastic MAB" -> in the stochastic MAB
second paragraph of page 3 : you use too much "the learner" (he or him could be fine at some places)
paragraph 3 of the same page: "two different stopping conditions", I would rather say "constraints" (since in the fixed-budget setting there is no stopping condition)
l.137 -> we present THE CLUCB algorithm
l.187 -> of -> for

It is a bit difficult from the main text to really have an intuition on what the width of an exchange class represents, but I don't really know how to improve that.

There is though a small error in the proof that must be corrected
Summary: This paper provides a complete theory for pure exploration in combinatorial bandit problems (algorithms, and a lower bound), an interesting framework that has never been considered in the literature. The algorithms seem really natural and interesting. The paper suffers from being a bit too long (and some part should be removed for the sake of clarity), and in the proof I read, I found a small mistake related to a missing uniong bound, that should be easily corrected.
Author Feedback
Author rebuttal: We thank all reviewers for their careful reading of the paper and their constructive comments. We will take these comments into serious consideration and make corresponding revisions.

--Reviewer 42--

** linear bandits **
- On line 409, we discussed the setting where queries are subsets of arms, under the name of ``combinatorial bandits’’. As we noted in the paper, most existing work of this category focuses on minimizing regret. On the other hand, to the best of our knowledge, there are no known results on the pure exploration problem of linear bandits. We consider the pure exploration problem of linear bandits as an important future work.

--Reviewer 45--

Thank you very much for your encouraging review!

--Reviewer 46--

** missing union bound **
- We acknowledge that the reviewer’s concern on the mistake in the proof of Theorem 1 (around line 706, Lemma 8) is valid. Just like the reviewer suggested, this can be easily fixed by enumerating over all possible values of T_t(e) (from 1 to t-1). We have already fixed it. This slightly increases the log factor in the confidence interval from log(4nt^2/\delta) to log(4nt^3/\delta).

** explicit sample complexity bound **
- We will give an explicit sample complexity bound in Theorem 1 in our final version.

** matroid **
- A formal definition of matroids appears in Appendix F (page 34, footnote 3). We will move it to the main text.